# Benchmarking test of empirical root water uptake models

Marcos Alex dos Santos[1], Quirijn de Jong van Lier[2], Jos C. van Dam[3], and Andre Herman Freire Bezerra[1]

[1]"Luiz de Queiroz" College of Agriculture, University of São Paulo, Piracicaba (SP), Brazil
[2]Center for Nuclear Energy in Agriculture, University of São Paulo, Piracicaba (SP), Brazil
[3]Department of Environmental Sciences, Wageningen University, The Netherlands

*Correspondence to:* Marcos Alex dos Santos (marcosalex.ma@gmail.com)

**Abstract.** Detailed physical models describing root water uptake (RWU) are an important tool for the prediction of RWU and crop transpiration, but involved hydraulic parameters are hardly-ever available, making them less attractive for many studies. Empirical models are more readily used because of their simplicity and the associated lower data requirements. The purpose of this study is to evaluate the capability of some empirical models to mimic the RWU distribution under varying environmental conditions predicted from numerical simulations with a detailed physical model. A review of some empirical models used as sub-models in ecohydrological models is presented, and alternative empirical RWU models are proposed. All these empirical models are analogous to the standard Feddes model, but differ in how RWU is partitioned over depth or how the transpiration reduction function is defined. The parameters of the empirical models are determined by inverse modelling of simulated depth-dependent RWU. The performance of the empirical models and their optimized empirical parameters depend on the scenario. The standard empirical Feddes model only performs well in scenarios with low root length density $R$, i.e. for scenarios with low RWU "compensation". For medium and high $R$, the Feddes RWU model cannot mimic properly the root uptake dynamics as predicted by the physical model. The Jarvis RWU model in combination to the Feddes reduction function (JMf) only provides good predictions for low and medium $R$ scenarios. For high $R$, it cannot mimic the uptake patterns predicted by the physical model. Incorporating a newly proposed reduction function in the Jarvis model improved RWU predictions. Regarding the ability of the models in predicting plant transpiration, all models accounting for compensation show good performance. The Akaike information criteria (AIC) indicates that the Jarvis (2010) model (JMII), with no empirical parameters to be estimated, is the "best model". The proposed models are better in predicting RWU patterns similar to the physical model. The statistical indices point them as the best alternatives to mimic RWU predictions of the physical model.

# 1 Introduction

The rate at which a crop transpires depends on atmospheric conditions, the shape and properties of the boundary between crop and atmosphere, root system geometry, and crop and soil hydraulic properties. The study and modelling of the involved interactions is motivated by the importance of transpiration for global climate and crop growth (Chahine, 1992) as well as by the role of root water uptake (RWU) in soil water distribution (Yu et al., 2007). The common modelling approach introduced by Gardner (1960), referred to as microcoscopic or mesoscopic (Raats, 2007), is not readily applicable to practical problems due to the difficulty in describing the complex geometrical and operational function of the root system and its complex interactions with soil (Passioura, 1988). However, it gives insight into the process and allows developing upscaled- physical macroscopic models (De Willigen and van Noordwijk, 1987; Heinen, 2001; Raats, 2007; De Jong van Lier et al., 2008, 2013).

In many one- and two-dimensional problems, macroscopic RWU is modelled as a sink term in the Richards equation, whose dependency on water content or pressure head is usually represented by simple empirical functions (ex. Feddes et al. (1976, 1978); Lai and Katul (2000); Li et al. (2001); Vrugt et al. (2001); Li et al. (2006)). Most of these models are derived from the Feddes et al. (1978) model, which consists of partitioning potential transpiration over depth according to root length density and applying a stress reduction function of piecewise linear shape — defined by five threshold empirical parameters — to accounting for local uptake reduction. Results of experimental studies (Arya et al., 1975b; Green and Clothier, 1995, 1999; Vandoorne et al., 2012) and the development of physically based-models (De Jong van Lier et al., 2008; Javaux et al., 2008) increased the understanding of the mechanism of RWU as a non-local process affected by non-uniform soil water distribution in the rhizosphere (Javaux et al., 2013). Accordingly, a plant can increase water uptake in wetter soil layers in order to compensate for uptake reductions in dryer layers to keep transpiration rate at the potential rate or mitigate transpiration reduction. Several empirical approaches have been developed over the years to account for this so-called compensation mechanism (Jarvis, 1989; Li et al., 2002, 2006; Lai and Katul, 2000). These models have been incorporated into larger hydrological models and tested at site-specific environments, showing an improvement of predictive quality for, e.g., soil water content and crop transpiration (ex. Braud et al. (2005); Yadav et al. (2009); Dong et al. (2010)). Comparisons with physically-based models (Jarvis, 2011; de Willigen et al., 2012) that implicitly account for compensation showed that models not including compensation, like Feddes et al. (1978), are less accurate with respect to crop transpiration and soil water content predictions under some circumstances, e.g. at a high root length density.

Recently, De Jong van Lier et al. (2013) developed a mechanistic model for predicting water potentials along the soil-root-leaf pathway, allowing the prediction of RWU and crop transpiration. This model was incorporated in the eco-hydrological model SWAP (Van Dam et al., 2008) by employing a piecewise function between leaf pressure head and relative transpiration, reducing the number of empirical parameters needed when compared to other relations (ex. Fisher et al. (1981)). Besides parameters describing soil hydraulic properties and root geometry, this new model requires information about root radial hydraulic conductivity, xylem axial conductance and a limiting leaf water potential. Although conceptually interesting, the difficulty to obtain the required input parameters makes the model less attractive for routine applications.

Other physical RWU models include the Couvreur et al. (2012), comparable to the De Jong van Lier et al. (2013), as well as more complex three-dimensional models (e.g. Javaux et al. (2013)), which account for the full root architecture, requiring more input parameters and a higher computational effort. Specifically, the De Jong van Lier et al. (2013) model differs from the previously mentioned models by the fact the RWU is based on matric flux potential with an equation derived from the microscopic RWU approach (De Jong van Lier et al., 2008), whereas in other models RWU is based on water pressure head. Scenarios including an osmotic potential can also be simulated by the model (de Jong van Lier et al., 2009).

Empirical RWU models are more readily used because of their relative simplicity and lower data requirements. On the other hand, their empirical parameters do not have a clear physical meaning and cannot be independently measured. Their limitations under varying environmental conditions are not well established. For the case of the Feddes et al. (1978) transpiration reduction function, threshold values are available in literature (Taylor and Ashcroft, 1972; Doorenbos and Kassam, 1986) for some crops and some levels of transpiration demand. Nevertheless, experimental (Denmead and Shaw, 1962; Zur et al., 1982) and theoretical (Gardner, 1960; De Jong Van Lier et al., 2006) studies indicate that these parameters should not depend only on crop type and atmospheric demand, but are also determined by root system parameters and soil hydraulic properties. Furthermore, threshold values are hardly ever validated, and they cannot be used for other models (like the Jarvis (1989) model) due to conceptual differences. Accurate values for crops accounting for more environmental factors are necessary to apply these models for a wider range of scenarios. Due to the great number of models developed over the years, it is paramount to investigate some of these before attempting to determine their parameters.

The general purpose of this study was to evaluate the ability of some empirical models to mimic the dynamics of RWU distribution under varying environmental conditions performed in numerical experiments with a detailed physical model proposed by De Jong van Lier et al. (2013). The detailed physical model accounts for hydraulic resistances from the soil to the leaf. We first review some empirical RWU models that have been employed in ecohydrological models and suggest some alternatives. By determining the parameters of the empirical models using inverse modelling of simulated depth-dependent RWU, it becomes clear to which extent the empirical models can mimic the dynamic patterns of RWU.

## 2  Theory

RWU and crop transpiration are linked through the principle of mass conservation for water flow in the soil-plant-atmosphere pathway:

$$T_a = \int_{z_m} S(z) \mathrm{d}z \tag{1}$$

where $T_a$ (L) is the crop transpiration and $S$ (L$^3$L$^{-3}$T$^{-1}$) is the root water uptake, dependent on crop properties and soil hydraulic conditions, a function of soil depth $z$ (L), and $z_m$ (L) the maximum rooting depth. Eq. 1 neglects the change of water storage in the plant, which is justified for daily scale predictions, assuming that plants rehydrate to the same early morning water potentials on successive days (Taylor and Klepper, 1978).

In a macroscopic modelling approach, RWU is calculated as a sink term $S$ in the Richards equation, which for the vertical coordinate is given by:

$$\frac{\partial \theta}{\partial t} = \frac{\partial}{\partial z}\left[K(\theta)\left(\frac{\partial h}{\partial z}+1\right)\right] - S \tag{2}$$

where $\theta$ (L$^3$ L$^{-3}$) is the soil water content, $h$ (L) the soil water pressure head, $K$ (L T$^{-1}$) the soil hydraulic conductivity, $t$ (T) the time and $z$ (L) the vertical coordinate (positive upward). To apply eq. 2, a functional expression for $S$ is needed. Physical equations in analogy to Ohm's law have been suggested (see the review of Molz (1981) for examples) as well as expressions derived by upscaling microscopic models (De Willigen and van Noordwijk, 1987; Feddes and Raats, 2004; De Jong van Lier et al., 2008, 2013). Alternatively, simple empirical models requiring less information about plant and soil hydraulic properties have also been proposed and are commonly used. Most of these models use the Feddes approach (Feddes et al., 1976, 1978), formulated as:

$$S(z) = S_p(z)\alpha(h[z]) \tag{3}$$

where $\alpha(h)$ is the RWU reduction function, defined by Feddes et al. (1978) as a piecewise linear function of $h$ (Fig. 1). According to this approach, a reduction in $S$ due to $\alpha(h[z]) < 1$ directly implies a transpiration reduction, making $\alpha(h)$ to be called a transpiration reduction function. $S_p$ is the potential RWU, which is determined by partitioning potential transpiration $T_p$ over depth. Several ways to estimate $S_p$ as a function of depth have been proposed (Prasad, 1988; Li et al., 2001; Raats, 1974; Li et al., 2006), the most common being to distribute $T_p$ over depth according to the normalized root length density $\beta$ (L$^{-1}$) defined as a fraction of root length density $R$ (L L$^{-3}$):

$$S_p(z) = \frac{R(z)}{\displaystyle\int_{z_m} R(z)\mathrm{d}z} T_p = \beta(z)T_p \tag{4}$$

Different expressions for $\alpha$ have been suggested, normally considering $\alpha$ a function of $\theta$ (ex. Lai and Katul (2000); Jarvis (1989)), of $h$ (ex. Feddes et al. (1978)) or of a combination of both (Li et al., 2006). Comparing to $\theta$, $h$ seems to be more feasible because of its relation to soil water energy and the fact that obtained parameters of such a function would be more likely applicable to different soils. Some reduction functions, generally associated to reservoir models for soil water balance, correlates RWU to the effective saturation. Regarding the shape of the reduction curve, smooth non-linear functions constrained between wilting point and saturation have been used, as well as piecewise linear functions, but they are all described by at least two empirical parameters. The parameters of the smooth non-linear functions allow easy curve fitting, whereas in the piecewise functions they stand for the threshold at which RWU (or crop transpiration) is reduced due to drought stress, which has been an important parameter in crop water management.

Metselaar and De Jong van Lier (2007) showed for a vertically homogeneous root system that the shape of $\alpha$ is not linearly related to soil water content neither to pressure head. A linear relation between $\alpha$ and matric flux potential, a composite soil hydraulic function defined in eq. 5, is physically more plausible and was experimentally corroborated by Casaroli et al. (2010).

Matric flux potential is defined as

$$M = \int\limits_{h_w}^{h} K(h)\,\mathrm{d}h \tag{5}$$

where $h_w$ is the soil pressure head at wilting point. Accordingly, a more suitable expression for $\alpha$ would be a piecewise linear function of $M$ (Fig. 1). RWU can then be calculated by the Feddes model (eq. 3), replacing its reduction function at the dry
side by the alternative illustrated in Fig. 1.

## 2.1   Physically based root water uptake model

By upscaling earlier findings about water flow towards a single root in the microscopic scale disregarding plant hydraulic resistance (De Jong Van Lier et al., 2006; Metselaar and De Jong van Lier, 2007), De Jong van Lier et al. (2008) derived the following expression for $S$:

$$S(z) = \rho(z)(M_s(z) - M_0(z)) \tag{6}$$

where $M_s$ is the bulk soil matric flux potential, $M_0$ the value of $M$ at root surface and $\rho(z)$ (L$^{-2}$) a composite parameter, depending on $R$ and root radius $r_0$:

$$\rho(z) = \frac{4}{r_0^2 - a^2 r_m^2(z) + 2[r_m^2(z) + r_0^2]\ln[ar_m(z)/r_0]} \tag{7}$$

where $r_m(= \sqrt{1/\pi R})$ (L) is the rhizosphere radius — defined as the half distance between neighbouring roots— and $a$ the
distance relative to $r_m{-}r_0$ where water content equals the average soil water content. In De Jong van Lier et al. (2013), this model is extended by taking into account the hydraulic resistances to water flow within the plant. Dividing water transport within the plant into two physical domains (from root surface to root xylem to leaf), assuming no water changes within the plant tissue and by coupling eq. 6 for water flow within the rhizosphere, they derived the following expression relating water potentials and $T_a$:

$$h_0(z) = h_l + \varphi(M_s(z) - M_0(z)) + \frac{T_a}{L_l} \tag{8}$$

where $h_0$ and $h_l$ (L) are the pressure heads at the root surface and leaf, respectively, $L_l$ (T$^{-1}$) is the overall conductance of the root xylem-to-leaf pathway, and $\varphi$ (T L$^{-1}$) is defined as:

$$\varphi(z) = \frac{\rho r_m^2(z)\ln\dfrac{r_0}{r_x}}{2K_{root}} \tag{9}$$

where $K_{root}$ (L T$^{-1}$) is the radial root tissue conductivity (referring to the pathway from the root surface to the root xylem), and
$r_x$ (L) is the xylem radius. An analytical solution of eq. 8 for $h_0$ or $M_0$ depends on an expression for $M_0(h_0)$. For a particular case of Brooks and Corey (1964) soils a solution is provided by De Jong van Lier et al. (2013). For van Genuchten–Mualem type soils, eq. 8 can be solved numerically or by using a semi-analytical solution of eq. 5 (De Jong van Lier et al., 2009). In

any case, application of eq. 8 requires a mathematical function relating $T_a$ and $h_l$. De Jong van Lier et al. (2013) defined $T_a$ by a piecewise function imposing a limiting value $h_{wl}$ on $h_l$:

$$T_r = \begin{cases} 1 & : h_l > h_{wl} \\ 0 \leq T_r \leq 1 & : h_l = h_{wl} \\ 0 & : h_l < h_{wl} \end{cases} \tag{10}$$

where $T_r$ $(= T_a/T_p)$ is the relative crop transpiration. Because $T_a$ and $h_l$ are unknowns and $T_a$ is undefined when $h_l = h_{wl}$, the
equation system cannot be solved analytically. An iterative solution was provided in De Jong van Lier et al. (2013) by defining a maximum transpiration rate $T_{p,max}$, corresponding to $T_a$ (eq. 8) for $h_l = h_{wl}$. The system of equations is then solved by defining plant stress in terms of $T_{p,max}$, according to the following boundary conditions:

$$\begin{cases} \text{unstressed conditions} : T_{p,max} > T_p & : T_a = T_p, \ h_l > h_{wl} \\ \text{stressed conditions} : \quad T_{p,max} < T_p & : h_l = h_{wl}, \ T_a < T_p \end{cases}$$

In the De Jong van Lier et al. (2013) model, crop water stress, a condition for which $T_a < T_p$, is defined at the crop level
(Tardieu, 1996) and onsets when $h_l = h_w$. $S$ can be calculated using eq. 6 by solving eq. 8, with $h_0$ (so $M_0$) variable over the root zone and controlled by plant hydraulic properties and soil hydraulic conditions.

Fig. 2 shows RWU for several values of $h_l$ and helps to understand how RWU is distributed over depth. $h_l$ can be regarded as a measure of water deficit stress over the whole root zone at crop level: as soil water is depleted, $h_l$ is reduced, thus increasing the driving force for RWU. Fig. 2 shows RWU for several values of $h_l$. As soil pressure head $h_s$ decreases, high uptakes are
only achieved by lower $h_l$. For a given $h_l$ value, RWU is substantially reduced as $h_s$ decreases. If $h_l$ is not reduced while $h_s$ decreases, $S$ becomes negative (although not shown in Fig. 2, negative $S$ is part of the extension of each curve) and water will flow from root to soil, a phenomenon called hydraulic lift or hydraulic re-distribution (Jarvis, 2011). This situation occurs when parts of the root zone are wetter and RWU from these parts satisfies transpiration demand, hence $h_l$ is not reduced.

Fig. 2 also shows that RWU is sensitive to both $R$ and $h_s$, and that it can be locally balanced by $R$ and soil water content.
Under homogeneous soil water distribution, RWU is partitioned proportionally to $R$. For heterogeneous conditions, RWU for lower $R$ and higher $R$ may be the same depending on the stress level (indicated by $h_l$) and the $h_s$ (see Fig. 2). This is in agreement with experimental results reported by several authors (Arya et al., 1975a, b; Green and Clothier, 1995; Verma et al., 2014) who found less densely-rooted but wetter parts of the root zone to correspond to a significant portion of RWU when more densely-rooted parts of the soil were drier, allowing the crop to maintain transpiration at potential rates. Empirical model
concepts that only use $R$ for predicting RWU distribution over depth (under non-stressed conditions) are most common, and therefore these results have been interpreted as due to a mechanism labelled "compensation" by which uptake is "increased" from wetter layers to compensate the "reduction" in the drier layers (Jarvis, 1989; Šimůnek and Hopmans, 2009). It is clear, however, that this compensation concept is found merely on a reference RWU distribution based on $R$ and it only needs to be explicitly addressed in empirical models. In physical models, distinguishing compensation is not necessary since in such
models "compensation" follows implicitly from the RWU mechanism.

In order to account for RWU pattern changes due to heterogeneous soil water distribution (the so-called "compensation"), several empirical models have been developed over the years. These models follow the general framework of the Feddes et al. (1978) model given by eq. 3. Below we review these models and present a new empirical alternative.

## 2.2 Empirical root water uptake models accounting for compensation

### 2.2.1 The Jarvis (1989) model

Jarvis (1989) defined a weighted-stress index $\omega$ ($0 \leq \omega \leq 1$) as

$$\omega = \int_{z_m} \alpha(z)\beta(z)\mathrm{d}z. \tag{11}$$

where, differently from Feddes et al. (1978), $\alpha$ was defined as a function of the effective saturation. Whereas Feddes et al. (1978) assume the RWU reduction directly to reflect in crop transpiration reduction, the Jarvis (1989) approach employs a so-called "whole-plant stress function" given by:

$$\frac{T_a}{T_p} = \min\left\{1, \frac{\omega}{\omega_c}\right\} \tag{12}$$

where $\omega_c$ is a threshold value of $\omega$ for the transpiration reduction. Substituting eq. 3 and 4 into eq. 1 (the mass conservation principle) and combining with eq. 12, results in:

$$S(z) = S_p\alpha(z)\alpha_2, \text{ where } \alpha_2 = \frac{1}{\max\{\omega, \omega_c\}} \tag{13}$$

where $\alpha_2$ is called the compensation factor of RWU, distinct from Feddes' $\alpha$ (eq. 3) and which can be derived by defining $T_a$ by eq. 12. In the Jarvis (1989) model, $\alpha$ accounts for local reduction of RWU and transpiration reduction is computed by eq. 12. When $\omega = 1$, there is no RWU reduction ($\alpha = 1$ throughout the root zone) and the model prediction is equal to the Feddes model. For $\omega_c < \omega < 1$, uptake is reduced in some parts of the root zone (as computed by $\alpha < 1$) but the plant can still achieve potential transpiration rates by increasing RWU over the whole root zone by the factor $\alpha_2$. When $\omega < \omega_c$, even though the uptake is increased by the factor $\alpha_2$, the potential transpiration rate cannot be met. The threshold value $\omega_c$ places a limit on the plant's ability to deal with soil water stress. When $\omega_c$ tends to zero, eq. 12 tends to 1, and the plant can fully compensate uptake and transpire at the potential rate if $\alpha > 0$ at some position within the root zone.

In principle, any definition of $\alpha$ is applicable in eq. 11, and commonly the Feddes et al. (1978) reduction function is used instead of the original Jarvis (1989) reduction function, e.g. in the HYDRUS model (Simunek et al., 2009). This modified version of the Jarvis (1989) model, hereafter referred to as JMf, will be further analysed. Nevertheless, one should be careful in setting up and interpreting the threshold parameters of JMf. The Feddes et al. (1978) model does not account for compensation, and the threshold pressure head value below which RWU is reduced ($h_3$) also represents the value below which transpiration is reduced, making $h_3$ values from literature to refer to this interpretation. Instead, in the JMf the transpiration reduction only takes place when $\omega < \omega_c$, and soil pressure head in some layers is already supposed to be more negative than $h_3$. Therefore, $h_3$ in JMf is less negative than its namesake in the Feddes model. In that sense, $h_3$ for the JMf is hard to determine experimentally. An option to do so would be by inverse modelling, optimizing outcomes of soil water flow models with experimental data.

### Comparison to the De Jong van Lier et al. (2008) model

The physical basis of Jarvis (1989), defined by eq. 11 to 13 with using any $\alpha$, has been questioned (Skaggs et al., 2006; Javaux et al., 2013). However, the Jarvis model has, to some extent, a physical basis, and a comparison with the physically-based model of De Jong van Lier et al. (2008) can be made, as demonstrated in Jarvis (2010, 2011). This is discussed in the following.

De Jong Van Lier et al. (2006) derived eq. 6 for describing RWU. Crop transpiration is obtained by integrating eq. 6 over $z_m$ as defined in eq. 1, leaving two unknowns: $M_0$ and $T_a$. To solve for these, De Jong van Lier et al. (2008) defined $T_a$ as a piecewise function:

$$\frac{T_a}{T_p} = \min\left\{1, \frac{T_{p,max}}{T_p}\right\} \tag{14}$$

where $T_{p,max}$ (L T$^{-1}$), differently from the definition in the De Jong van Lier et al. (2013) model, is the maximum transpiration rate reached when the root surface pressure head is constant over depth and equal to a limiting value $h_w$. For such a condition $M_0 = 0$ and $T_{p,max}$ is given by:

$$T_{p,max} = \int_{z_m} \rho(z) M(z) \, dz. \tag{15}$$

From eq. 14 we see that drought stress occurs when $T_{p,max} < T_p$. At the onset of drought stress, $T_a = T_{p,max}$. Under this condition, $M_0 = 0$ and $S(z)$ becomes:

$$S(z) = \rho(z) M(z). \tag{16}$$

When $T_{p,max} > T_p$, $T_a = T_p$ (no drought stress) and $M_0$ ($> 0$) is given by:

$$M_0 = \frac{\int_{z_m} \rho(z) M(z) dz - T_p}{\int_{z_m} \rho(z) dz} \tag{17}$$

Jarvis (2011) observed the similarities between eq. [14] and [12] of the models, as well as the algebraic similarity between $\omega$ (eq. 11) and $T_{p,max}$ (eq. 15). Thus, Jarvis (2010) showed that both models provide the same results under drought stress if $\alpha$ and $\beta(z)$ are defined as follows:

$$\alpha = \frac{M}{M_{max}} \tag{18}$$

$$\beta = \frac{\rho(z)}{\int_{z_m} \rho(z) dz} \tag{19}$$

where $M_{max}$ is the maximum value of $M$ (i.e., at $h = 0$). By substituting eq. [18] and [19] into eq. 15 and comparing eq. 12 with eq. 14, $\omega_c$ is found to be equal to:

$$\omega_c = \frac{T_p}{M_{max} \int\limits_{z_m} \rho(z)\,\mathrm{d}z} \tag{20}$$

Substitution of eq. [18] to [20] into eq. [12] and [11] results in eq. 16 of the De Jong van Lier et al. (2008) model for stressed

conditions. Consequently, both models provide the same numerical results. For unstressed conditions, analogous substitution results in:

$$S(z) = \rho(z)M(z)\frac{T_p}{T_{p_{max}}} = \frac{\rho(z)M(z)}{\int\limits_{z_m} \rho(z)M(z)\,\mathrm{d}z}T_p \tag{21}$$

Eq. 21 is different from eq. 6 and, therefore, the models cannot be correlated for these conditions. The Jarvis (1989) model predicts RWU by a weighting factor between $\rho$ and $M$ throughout rooting depth. Defining $\alpha$ and $\beta$ by eq. 18 and 19, respec-

tively, allowed to correlate both models only for stressed conditions. These definitions and the resulting model will be further analysed.

### 2.2.2  The Li et al. (2001) model

Li et al. (2001) proposed to distribute potential transpiration over the root zone by a weighted stress index $\zeta$, being a function of both root distribution and soil water availability:

$$\zeta(z) = \frac{\alpha(z)R(z)^{l_m}}{\int\limits_{z_m} \alpha(z)R(z)^{l_m}\mathrm{d}z} \tag{22}$$

where $\alpha$ (-) and $R$ (L L$^{-3}$) were previously defined and the exponent $l_m$ is an empirical factor modifying the shape of RWU distribution over depth. Originally, the $l_m$ values were based on experimental works. For $0 < l_m < 1$, the RWU in sparsely rooted soil layers is increased in the attempt to mimic compensation. For $l_m > 1$, which has no maximum, the uptake from more densely rooted soil layers increases. Thus, $S_p$ is given by:

$$S_p = \zeta(z)T_p \tag{23}$$

and RWU is calculated by substituting eq. 23 into eq. 3, following the Feddes approach.

As an alternative to the Jarvis (1989) model, $S_p$ can be defined as function of root length density and soil water availability distribution. Compensation is directly accounted for by the weighted stress index in eq. 22. However, using $\alpha$ to represent soil water availability in eq. 22 does not mimic properly the compensation mechanism. Compensation may take place before

transpiration reduction. Using $\alpha$ in eq. 22 means that compensation will only take place after the onset of transpiration reduction when $\alpha$ in one or more layers is smaller than 1. The $l_m$ parameter may also be interpreted as to account for compensation under non-stressed condition. However, compensation as well as the shape of the RWU distribution are likely to change when a soil becomes drier, and a constant $l_m$ cannot account for that.

### 2.2.3 The Molz and Remson (1970) and Selim and Iskandar (1978) models

Decades before Li et al. (2001), Molz and Remson (1970) and Selim and Iskandar (1978) already suggested to distribute potential transpiration over depth according to root length density and soil water availability. Instead of using $\alpha$ to account for soil water availability, they used soil hydraulic functions. The weighted stress index was defined as

$$\zeta(z) = \frac{\Gamma(z)R(z)}{\displaystyle\int_{z_m} \Gamma(z)R(z)\mathrm{d}z} \tag{24}$$

where $\Gamma$ is a soil hydraulic function to account for water availability. Molz and Remson (1970) used soil water diffusivity $D$ ($\mathrm{L^2T^{-1}}$), and Selim and Iskandar (1978) used soil hydraulic conductivity $K$ ($\mathrm{LT^{-1}}$) for $\Gamma$ in eq. 24. RWU is then calculated by substituting eq. 24 into eq. 23 and then into eq. 3 following the Feddes approach.

These models may better represent RWU and compensation than the Li et al. (2001) model. The compensation is implicitly accounted for by means of $\Gamma$ in $\zeta$. Since $\Gamma$ decreases as soil water is depleted, in a heterogeneous soil water distribution $\zeta$ in wetter layers is relatively increased because the overall $\int \Gamma R dz$ is reduced due to the reduction of $\Gamma$ in drier, more densely rooted soil layers. Differently from the Li et al. (2001) model, this change in RWU distribution can occur before the onset of transpiration reduction. Heinen (2014) compared different types of $\Gamma$ in eq. 24 such as the relative hydraulic conductivity ($K_r = K/K_{sat}$) and relative matric flux potential ($M_r = M/M_{max}$), among others. He found large differences in predicted RWU patterns for different forms of $\Gamma$, but did not indicate a preference for a specific one.

### 2.2.4 Proposed empirical model

In describing soil water availability, the matric flux potential $M$ may be a better choice than $K$ or $D$, since it integrates $K$ and $h$ or $D$ and $\theta$ (Raats, 1974; De Jong van Lier et al., 2013). We propose a new weighted stress index, defined as:

$$\zeta_m(z) = \frac{R^{l_m}M(h)}{\displaystyle\int_{z_m} R^{l_m}M(h)\mathrm{d}z} \tag{25}$$

The exponent $l_m$ provides additional flexibility on distributing $T_P$ over depth, as also shown by Li et al. (2001). The proposed model differs from Li et al. (2006) only on the hydraulic property to account for soil water availability. The $\alpha$ function used in Li et al. (2006) can only alter RWU distribution after the onset of transpiration reduction, as commented earlier. Contrastingly, $M$ affects RWU distribution before transpiration reduction, integrating the effect of both $K$ and $h$.

The RWU can then be obtained by inserting eq. 25 into eq. 23 ($S_p$) and multiplying by any reduction function, such as the Feddes et al. (1978) and proposed reduction functions. In other words, the model follows the Feddes approach, which computes RWU by the two mentioned steps, differing only with respect to the way $S_p$ is obtained: eq. 25 (multiplied by $T_p$) versus eq. 4.

## 3 Material and Methods

### 3.1 Applied models

Table 1 summarizes the empirical RWU models evaluated in this study. They all follow the original Feddes model (eq. 3), but differing in how RWU is partitioned over rooting depth or how $\alpha$ is defined. For each model, except for Jarvis (2010), we defined a modified version by substituting the Feddes reduction function by the proposed reduction function (Fig. 1b), and these modified versions were also evaluated. The threshold values of the Feddes et al. (1978) reduction function for anoxic conditions ($h_1$ and $h_2$) were set to zero. The value of the parameter $h_4$ was set to $-150$ m. The other parameters of the models were obtained by optimization as described in section 3.3.

All these models were embedded as sub-models into the ecohydrological model SWAP (Van Dam et al., 2008), allowing to to solve eq. 2 and to apply it different scenarios of root length density, atmospheric demand and soil type (described in section 3.2) to analyse the behaviour and sensitivity of the models. Simulation results of SWAP in combination with each of the RWU models were compared to the SWAP predictions in combination with the physical RWU model developed by De Jong van Lier et al. (2013).

The values of the De Jong van Lier et al. (2013) model parameters used in the simulations are listed in Table 2. The values of $K_{root}$ and $L_l$ are within the range reported by De Jong van Lier et al. (2013).

### 3.2 Simulation scenarios

#### 3.2.1 Drying-out simulation

Boundary conditions for drying-out simulations were no rain/irrigation and a constant atmospheric demand (potential transpiration) over time. The simulation continued until simulated crop transpiration by the physical RWU model approached zero. Soil evaporation was set to zero making soil water to depleted only due to RWU or bottom drainage. Free drainage (unit hydraulic gradient) at the maximum rooting depth was the bottom boundary condition. The soil was initially at hydrostatic equilibrium with a water table located at 1 m depth. We performed simulations for two levels of atmospheric demand given by potential transpiration ($T_p$) of 1 and 5 mm d$^{-1}$. We also considered three soil types and three levels of root length density, as described in the following.

#### 3.2.2 Soil type

Soil data for three top soils from the Dutch Staring series (Wösten et al., 1999) were used. The physical properties of these soils are described by the Mualem-van Genuchten functions (Mualem, 1976; Van Genuchten, 1980) for the $K - \theta - h$ relations:

$$\Theta = [1 + |\alpha h|^n]^{(1/n)-1} \tag{26}$$

$$K = K_{sat}\Theta^\lambda[1 - (1 - \Theta^{n/(n-1)})^{1-(1/n)}]^2 \tag{27}$$

where $\Theta = (\theta - \theta_r)/(\theta_s - \theta_s)$; $\theta, \theta_r$ and $\theta_s$ are water content, residual water content and saturated water content ($L^3\ L^{-3}$), respectively; $h$ is pressure head (L); $K$ and $K_{sat}$ are hydraulic conductivity and saturated hydraulic conductivity, respectively ($L\ T^{-1}$); and $\alpha$ ($L^{-1}$), $n$, and $\lambda$ are empirical parameters. The parameter values for the three soils are listed in Table 3. These soils are identified in this text as clay, loam and sand.

### 3.2.3  Root length density distribution

Three levels of root length density were used, according to the range of values normally found in the literature. We considered low, medium and high root length density for average crop values equal to 0.01, 0.1 and 1.0 cm $cm^{-3}$, respectively. For all cases, we set the maximum rooting depth $z_{max}$ equal to 0.5 m. Root length density over depth $z$ was described by the exponential function:

$$R(z_r) = R_0(1 - z_r)\exp^{-bz_r} \tag{28}$$

where $R_0$ ($L\ L^{-3}$) is the root length density at the soil surface, $b$ (-) is a shape-factor parameter and $z_r$ ($= z/z_{max}$) is the relative soil root depth. The term $(1 - z_r)$ in eq. 28 guarantees that root length density is zero at the maximum rooting depth. The parameter $R_0$ is hardly ever determined, whereas the average root length density of crops $R_{avg}$ is usually reported in the literature. Assuming $R$ of such a crop given by eq. 28, it can be shown that:

$$\int_0^1 R_0(1 - z_r)\exp^{-bz_r}\ \mathrm{d}z_r = R_{avg} \tag{29}$$

Solving eq. 29 for $R_0$ and substituting into eq. 28 gives:

$$R(z_r) = \frac{b^2 R_{avg}}{b + \exp^{-b} - 1}(1 - z_r)\exp^{-bz_r} \qquad (b > 0) \tag{30}$$

Fig. 3 shows $R(z_r)$ calculated from eq. 30 for different values of $b$ and $R_{avg} = 1$ cm $cm^{-3}$. As $b$ approaches zero, eq. 30 tends to become linear, however it is not defined for $b = 0$. In our simulations $b$ was arbitrarily set equal to 2.0.

### 3.3  Optimization

The parameters of the empirical RWU models were estimated by solving the following constrained optimization problem:

$$\text{minimize} \qquad \Phi(\mathbf{p}) = \sum_{i=1}^{n}\sum_{j=1}^{m}[S_{i,j}^* - S_{i,j}(\mathbf{p})]^2 \tag{31}$$

$$\text{subject to} \qquad \mathbf{p} \in \Omega$$

where $\Phi(\mathbf{p})$ is the objective function to be minimized, $S_{i,j}^*$ is the RWU simulated by SWAP model together with the De Jong van Lier et al. (2013) model at time $i$ (time interval of one day) and depth $j$ (of each soil layer) and $S_{i,j}(\mathbf{p})$ is the corresponding RWU predicted by SWAP in combination with one of the empirical models shown in Table 1. $\mathbf{p}$ is the model parameter vector to be optimized, constrained in the domain $\Omega$. Both $\mathbf{p}$ and $\Omega$ vary depending on the empirical RWU model used. Table 4 shows

the parameters of each empirical RWU model that were optimized and their respective constraints $\Omega$. $m$ is the number of soil layers (50 soil layers of 1 cm thickness) and $n$ is the duration, in days, of the simulation. The Jarvis (2010) model has no empirical parameters and therefore requires no optimization.

Eq. 31 was solved using the PEST (Parameter ESTimation) tool (Doherty et al., 2005) coupled to the adapted version of SWAP. PEST is a non-linear parameter estimation program that solves eq. 31 by the Gauss-Levenberg-Marquardt (GLM) algorithm, searching for the deviation, initially along the steepest gradient of the objective function and switching gradually the search to Gauss-Newton algorithm as the minimum of the objective function is approached. Upon setting PEST parameters, we made reference runs of SWAP with each empirical model using random values of $\mathbf{p}$ aiming to assess the ability of PEST for retrieving $\mathbf{p}$. These reference runs allowed to properly set up PEST for our case. For highly non-linear problems as in eq. 31, the optimized parameters set depends on the initial values of $\mathbf{b}$. We used five random sets of initial values for $\mathbf{p}$ in order to guarantee that GLM encountered the global minimum and also to check the uniqueness of the solution. Runs led to the same minimum in most cases, but if not, the minimum was compared and a fit run was made again.

The optimizations were performed for the drying-out simulation only. This guaranteed that RWU predictions from SWAP corresponded to the best fit of each empirical model to the De Jong van Lier et al. (2013) model. This analysis aimed to investigate the capacity of the empirical RWU models to mimic the RWU pattern predicted by the De Jong van Lier et al. (2013) model. The optimized parameters were subsequently used to evaluate the models in an independent growing season scenario.

### 3.4 Growing season simulation

In the growing season simulation, all models were evaluated by simulating the transpiration of grass with weather data from the De Bilt weather station, the Netherlands (52°06' N; 5°11 'E), for the year 2006. The same root system distribution as in the drying-out simulations was used, i.e. a crop with roots exponentially distributed over depth as eq. 30 ($b = 2.0$) down to 50 cm below soil surface. We also performed simulations for the same three types of soils and root length densities. In all cases the crop fully covered the soil with a leaf area index of 3.0. Daily reference evapotranspiration $ET_0$ was calculated by SWAP using the FAO Penman-Monteith method (Allen et al., 1998). In SWAP model, a potential crop evapotranspiration $ET_p$ is obtained by multiplying $ET_0$ by a crop factor, which for the grass vegetation was set to 1 (Van Dam et al., 2008). $ET_p$ was partitioned over potential evaporation $E_p$ and $T_p$ using parameter values for common crops given in SWAP model (see Van Dam et al. (2008) for details).

The values of the empirical parameters of each RWU model corresponding to the type of soil and root length density were taken from the optimizations performed in the drying-out experiment. Each parameter was estimated for two levels of $T_p$ (1 and 5 mm d$^{-1}$) and was linearly interpolated for intermediate levels of $T_p$. For $T_p$ higher than 5 mm d$^{-1}$ and $T_p$ lower than 1 mm d$^{-1}$, the values estimated for 5 and 1 mm d$^{-1}$, respectively, were used.

As in the drying-out simulations, the bottom boundary condition was free drainage. Initial pressure heads were obtained by iteratively running SWAP starting with the final pressure heads of the previous simulation until convergence.

## 4 Results and Discussion

### 4.1 Drying-out simulation

#### 4.1.1 Root water uptake pattern: De Jong van Lier et al. (2013) model

In this section we first focus on the behaviour of the De Jong van Lier et al. (2013) model in predicting RWU for the evaluated scenarios in the drying-out experiment. Fig. 4 shows the RWU patterns for the case of the clay soil for the three evaluated root length densities $R$ and the two levels of potential transpiration $T_p$. It can be seen how $R$ and $T_p$ affect RWU distribution and transpiration reduction as the soil dries out. The onset and shape of transpiration reduction is affected by the RWU pattern. For low $R$, the low number of roots in deeper layers is not sufficient to supply high RWU rates. When the upper layers become drier, transpiration reduction follows immediately. Under medium and high $R$, the RWU front moves gradually downward as water from the upper layers is depleted. Comparing from high to medium $R$, the RWU front goes even deeper, and transpiration is maintained at potential rates for a longer time (Fig. 4). Accordingly, the plant exploits the whole root zone and little water is left when transpiration reduction onsets, causing an abrupt drop in transpiration. Regarding $T_p$, the RWU patterns are very similar for both evaluated rates, differing only in time scale: for high $T_p$ the onset of transpiration reduction and the shift in RWU front occur earlier. The uptake patterns for the sand and loam soil (not shown here) are very similar. However, for the sand soil potential transpiration is maintained a little longer and more water is extracted from deeper layers. For the loam soil, the onset of transpiration reduction occurred earlier.

The leaf pressure head $h_l$ over time shown in Fig. 4 illustrates how the model adapts $h_l$ to $R$ and $T_p$ levels in a drying soil. Initially all scenarios have the same water content distribution and lower $h_l$ values are required for low $R$ or high $T_p$ scenarios to supply potential transpiration rates. As soil becomes drier, $h_l$ is decreased to increase the pressure head gradient between bulk soil and root surface, thus maintaining RWU corresponding to the demand. Therefore, uptake in wetter layers becomes more important. Transpiration reduction only onsets when $h_l$ reaches the limiting leaf pressure head $h_{wl}$ ($= -200$ m), after significant changes in the RWU patterns, characterized by increased uptake from deeper layers.

For the high $T_p$–low $R$ scenarios, transpiration reduction starts at the first day of simulation although the soil is relatively wet. This is a case of transpiration reduction under non-limiting soil hydraulic conditions due to high atmospheric demand (Cowan, 1965). For such conditions, the high water flow within the plant required to meet the atmospheric demand cannot be supported by the root system with a low $R$ and hydraulic parameters given in Table 2. Higher atmospheric demand (here represented by $T_p$) leads to faster reduction of $h_l$ caused by the hydraulic resistance to water flow within the plant, and the transpiration rate and RWU are a function of $h_l$. The physical model assumes a parsimonious relationship (eq. 10) between transpiration and $h_l$: transpiration rate is only reduced when $h_l$ reaches a limiting value $h_{wl}$, which corresponds to a maximum transpiration rate $T_{p,max}$ allowed by the plant for the current soil hydraulic and atmospheric conditions. Under non-limiting soil hydraulic conditions, root system properties and plant hydraulic parameters (Table 2) are the major determining factors for $T_{p,max}$, whereas soil hydraulic conditions play a minor role. Fig. 5 shows $T_{p,max}$ as a function of $K_{root}$ for some values of $L_l$ with a constant soil pressure head of -1 m in the root zone for low $R$ in the sandy soil. In this scenario, $K_{root}$ is limiting the

crop transpiration and $L_l$ becomes important only when $K_{root}$ increases. The potential transpiration can be achieved by raising $K_{root}$ up to about $10^{-7}$ m d$^{-1}$. This can also be achieved by decreasing $h_{wl}$ (not shown in Fig. 5).

In the field, transpiration rate and root length density are related to each other: a high transpiration rate only occurs at high leaf area and a high leaf area implies a high root length density. Thus, even in very dry and hot weather conditions, a crop with a

5 low $R$ may not be able to realize high transpiration rate. Furthermore, crop transpiration depends on the stomatal conductance. In the De Jong van Lier et al. (2013) model, this is implicitly taken into account by the simple relationship between $h_l$ and $T_a$. However, stomatal conductance is relatively complex and depends on several environmental factors such as air temperature, solar radiation and $CO_2$ concentration. Therefore, high potential transpiration rates may not be achieved because of the stomatal conductance reduction due to temperature or solar radiation. This behaviour could be simulated by the coupling of the De Jong

van Lier et al. (2013) model to stomatal conductance models, such as the Tuzet et al. (2003) model.

### 4.1.2 Root water uptake pattern predicted by the empirical models

In this section, we evaluate the empirical RWU models (models and their abbreviations are listed in Table 1) based on the comparison of RWU patterns and transpiration reduction over time with the respective predictions from the De Jong van Lier et al. (2013) model (VLM). All empirical model predictions were obtained with respective optimized parameters as shown in

Table 5 and are discussed in section 4.1.4, and therefore represent the best fit with VLM.

The RWU patterns simulated by VLM and the empirical models for the sandy soil and high $R$ scenario are shown in Fig. 6 and 7 for low and high $T_p$, respectively. Both versions of the Feddes model (FM and FMm) predicted enhanced RWU from the upper soil layers. When the soil pressure head ($h_s$) (for FM) or soil matric flux potential ($M_s$) (for FMm) is greater than the threshold value for uptake reduction, these uptake patterns are equivalent to the vertical $R$ distribution. For conditions

drier than the threshold value (when $\alpha_f$ and $\alpha_m$ are less than 1), the predicted RWU patterns by the models become different (Fig. 6 and 7).

When reducing RWU for a period depending on $R$, $T_p$ and $h_3$, RWU from the upper soil layers predicted by FM rapidly decreases to zero. This zero-uptake zone expands downward as soil dries out. On the other hand, the uptake predicted by FMm is substantially reduced right after the onset of transpiration reduction, proceeding at lower rates and a much longer time until

approaching zero. These features become evident by comparing the shape of both reduction functions (Fig. 8). $\alpha_m$ is linear with $M$ after $M > M_c$, but it is concavely-shaped as a function of $h$ — as also shown by Metselaar and De Jong van Lier (2007) and De Jong van Lier et al. (2009). This makes $\alpha_m$ to reduce abruptly for $M > M_c$, causing a substantial decrease in RWU even when $h$ is slightly below the threshold value. Therefore, RWU proceeds at low rates for a longer time. In contrast and due to the linear shape of $\alpha_f$, RWU predicted by FM remains higher for a longer time after $h < h_3$. FM does not predict

an abrupt change in RWU patterns, especially when $T_p$ is low (Fig. 6). When $h$ approaches $h_4$, $\alpha_f$ is still relatively high and RWU makes $h$ to decrease rapidly. Another diverging feature between $\alpha_f$ and $\alpha_m$, also shown in Fig. 8, is that the shape of $\alpha_m$ varies with soil type (regardless the value of its threshold parameter $M_c$), whereas $\alpha_f$ does not. These different features of the reduction functions also affect the matching values of the parameters as discussed below. Although the choice of the reduction function affects transpiration over time only slightly, RWU patterns are strongly affected (Fig. 6 and 7).

The RWU patterns predicted by JMf and JMm models can be very different, as shown by Fig. 6 for the high $R$–low $T_p$ scenario. In this scenario, the JMf model did not predict any compensation because the optimal $\omega_c$ equalled 1 (Table 5) — thus becoming identical to FM — and the optimal $h_3$ for JMf and FM were similar. In Fig. 6, although $h_3$ values for FM and JMf ($\omega_c = 1$) are close to zero, the plant transpiration is near $T_p$ for a prolonged time due to a small reduction of $\alpha$. These high $R$–low $T_p$ scenarios with a high $R$ in deep soil layers allow RWU at higher rates when surface soil layers become drier (as predicted by VLM). Then, the reduction of $\omega_c$, an attempt to numerically predict compensation with JMf, makes the RWU pattern to deviate even more from the VLM pattern. This is illustrated in Fig. 6 and by the optimal $h_3$ and $\omega_c$ values shown in Table 5. In order to mimic the VLM uptake patterns, the value of $h_3$ for all soil types in this scenario was equal or close to zero. Decreasing $h_3$ or $\omega_c$ to simulate compensation makes JMf predicting higher uptake from upper layers, increasing the discrepancy between the models. The optimal $\omega_c$ for all soil types was equal to 1 (in other words: there was no compensation). RWU in the upper layers predicted by VLM is substantially reduced within a few days, whereas reducing $\omega_c$ in the JMf model to predict compensation has the side-effect of causing an increase of uptake from upper layers. The model, therefore, is not able to adequately mimic the scenarios with compensation evaluated here. On the other hand, the JMm model was able to reproduce considerably well the VLM pattern for the evaluated scenarios due to the shape of $\alpha_m$ as discussed above. As soon as $M > M_c$ in the upper layers, RWU decreased at a higher rate, compensated by increasing uptake from the wetter, deeper layers. This agrees more closely to VLM predictions.

For high $T_p$ (Fig. 7), the JMf model can predict compensation ($\omega_c < 1$), however its predicted RWU pattern is quite different from JMm and VLM. JMf predicts a higher longer lasting RWU near the soil surface than the other models that account for compensation. This makes soil water depletion to be more intense and RWU from these layers to cease sooner when $h_s$ becomes lower than $h_4$. At this point, $T_a$ is predicted to continue equal to $T_p$ because of the low optimal $\omega_c$ (= 0.19), which increases RWU from the deeper layers where $h$ is close or equal to $h_4$. JMm performed very differently, predicting uptake over the first few days (when $M_s > M_c$) in accordance with $R$ distribution. After $M < M_c$ in the upper soil layers, the RWU pattern started to change gradually and RWU increased at lower depths.

The proposed models (PM and PMm ) are capable of predicting RWU patterns similar to VLM. For the low $T_p$–high $R$ scenario (Fig. 6), RWU is more uniformly distributed over depth than in the VLM model for the first days and uptake from upper layers is lower than that predicted by VLM model. For high $T_p$ (Fig. 7), these models better represent RWU patterns and, in general, differences in predictions of RWU between the proposed models are small. The shape of the transpiration reduction over time however, is smoother than the VLM model. Concerning the relative transpiration curve, the proposed models appear to be less precise than the other models that account for RWU compensation.

JMII does not mimic well the RWU pattern predicted by VLM for the high $R$–low $T_p$ scenarios. It overestimates uptake from surface layers during the first days. Before the onset of transpiration reduction, uptake from upper layers reaches zero, but it is compensated by a higher uptake from deeper layers. The model is very sensitive to both $R$ and $M$. For the high $R$–high $T_p$ scenarios, JMII provides better uptake pattern predictions (Fig. 7). However, the model does not perform well in the other scenarios with low and medium $R$ (data not shown here).

Comparing RWU predictions from JMf and JMII, it is seen that the Jarvis-type models are affected by the definition of $\alpha$. This becomes clear from Fig. 9 which shows $\alpha$ of JMII (eq. 18) as a function of $h_s$ and $\omega_c$ (eq. 20) for different soil types, expressed by $M_{max}$. The $\alpha$ function shows that even though the soil resistance increases as the soil becomes drier, defining $\alpha$ by eq. 18 does not seem plausible. In this case, $\alpha$ is suddenly reduced when the soil is still near saturation. When $h_s = 1$ m, for instance, $\alpha$ is much lower than 0.5. Such a behaviour is not reasonably expected for the $\alpha$ concept. The $\omega_c$ values are also extremely low. The low $\alpha$ values are, however, balanced by high $\alpha_2$ values (due to low $\omega$ and $\omega_c$ values), leading to suitable values of RWU in a given soil layer. Nevertheless, the magnitude of $\alpha$ and $\omega_c$ are conceptually questionable. Therefore, we conclude that: i) the $\omega_c$ value in Jarvis-type models, which sets the compensation level, depends on the $\alpha$ definition. For instance, for the original Jarvis (1989) model, $\omega_c = 0.5$ corresponds to a moderate level of compensation. Surely, this would not hold if $\alpha$ is defined by eq. 18; ii) Comparing the Jarvis (1989) to the De Jong van Lier et al. (2008) model led to a rather unrealistic $\alpha$ function, and its behaviour does not properly represent the $\alpha$ concept. This is may be caused by the fact that the De Jong van Lier et al. (2008) model does not take into consideration the plant hydraulic resistances. This might explain the rapid decline of $\alpha$ near saturation. The threshold type functions seem to be more feasible.

The fact that JMII is more sensitive to both $R$ and $M$, as stated above, when compared to the other $M$–based models is attributed to the $\alpha$ function and the derived equations to express their parameters (eq. 19 and 20). It can be seen from Fig. 9(c) that $\beta$ defined by eq. 19 ($\beta$ of JMII) tends to be higher when $R$ increases and lower when $R$ decreases compared to $\beta$ of JMf and JMm. Thereby, for the first days of simulations when the soil hydraulic conditions tend to be rather uniform over depth, JMII overestimates RWU compared to VLM predictions. This becomes more important for the high $R$–low $T_p$ scenarios. For such conditions, the RWU over depth predicted by the VLM tends to be more uniform, which seems reasonable as the low transpiration demand can be met by any small $R$ in deeper soil depths. After some time, the discrepancies between VLM and JMII tend to increase, since the higher RWU in the upper layers reduces $h$; thus, because of the $\alpha$ shape of JMII RWU in the upper layers are suddenly reduced towards zero. These are the main reasons why JMII does not predict well in the high $R$–low $T_p$ scenarios.

### 4.1.3 Statistical indices

The performance of the empirical models was analysed by the coefficient of determination $r^2$ and the model efficiency coefficient $E$ (Nash and Sutcliffe, 1970) calculated by comparing to the RWU and relative transpiration predicted by VLM. For the low $R$–high $T_p$ scenarios, the VLM predicts water stress ($T_a < T_p$) from the beginning of the simulation as discussed in section 4.1.1. The empirical models (except for JMf and JMm by setting $\omega_c > 1$) are not able to reproduce these results, thus these scenarios were not considered when analysing the performance of the models.

Statistical indices for the evaluated scenarios of each model are concisely shown by the boxplots in Fig. 10. The width of whiskers indicates the range of the statistical indices for each model used in the evaluated scenarios. The outliers indicate whether a model had different performance at some scenarios than its overall performance. Focusing first on RWU, the figure shows that the better performed better. The performance of PM was just a bit poorer than PMm's, shown by the presence of an outlier and lower median. JMm performed as good as the proposed models, and only in two scenarios it had a bad performance

as shown by the outliers in Fig. 10. The wider whiskers and presence of outliers of the others models confirm their poorer performance.

Among the models that account for RWU compensation, JMf and JMII performed worst, especially in the high $R$–low $T_p$ scenarios. In general their performances were poorer for medium $R$ scenarios, especially for low $T_p$. Thus, the use of $\alpha_m$ in Jarvis-type models promotes substantial improvements, especially from medium to high $R$ scenarios. For low $R$ scenarios all models performed well and the highest values of the boxes in Fig. 10 usually refer to this scenario.

On predicting transpiration all models accounting for compensation performed well, except JMf. It can be noticed that JMII performed much better on predicting transpiration than RWU. As for the RWU, all models performed worst in the high $R$ scenarios.

As the evaluated models differ regarding the number of empirical parameters (from 0 to 2), it is important to use a statistical measure that accounts for this and penalizes the models with more parameters. The Akaike's information criteria (AIC) is a suitable measure for such a model comparison. The selection of the "best" model is determined by an AIC score, defined as (Burnham and Anderson, 2002):

$$AIC = 2K - \log(\mathcal{L}(\hat{\theta}|y)) \tag{32}$$

where $K$ is the number of fitting parameters and $\mathcal{L}(\hat{\theta}|y)$ is the log-likelihood at its maximum point. The "best" model is the one with the lowest $AIC$ score. Table 6 lists the best models for every scenario based on $AIC$ score. Overall, the AIC supports the above descriptive statistical analyses, indicating that the proposed models are the best models in predicting RWU estimated by VLM, especially from medium–high $R$ scenarios. For the low $R$ scenarios JMm is the best model. On predicting $T_r$ by VLM, the above analyses indicated that, in general, most models performed similarly. The AIC indicated comparable results, but overall JMm was the best model. The proposed models (PM or PMm) were the best models for high $R$–low $T_p$ scenarios.

### 4.1.4   Relation of the optimal empirical parameters to $R$ and $T_p$ levels

The optimal values of the empirical parameters of all models (except JMII that has no empirical parameters) for all scenarios but the high $T_p$–low $R$ scenario are shown in Table 5. The threshold reduction transpiration parameters $h_3$ and $M_c$ (for FM and FMm, respectively) stand for the soil hydraulic conditions at which the crop cannot meet its potential transpiration rate. Conceptually, the higher $R$, the lower is $h_3$ or $M_c$ due to the larger root surface area for RWU, i.e. the crop can extract water in drier soil conditions. Similarly, lower $h_3$ and $M_c$ are expected for low $T_p$. This can also be deduced from Fig. 6 and 7 by means of the predictions of relative transpiration and RWU by VLM.

The optimal $h_3$ and $M_c$ values (Table 5) for FM and FMm, respectively, increase as $R$ increases, contradicting their conceptual relation to $R$. For $T_p$, there is no specific relationship for these parameters: whether they increase or decrease with $T_p$ depends on the value of $R$. In drying-out scenarios, soil water from top layers depletes rapidly due to the higher initial uptake. Thus, uptake from these layer starts to decrease whereas RWU in deeper, wetter layers increases. This effect becomes stronger at higher $R$, as seen by the VLM predictions in section 4.1.1. Because FM and FMm do not account for this mechanism, decreasing $h_3$ or $M_c$ in search for conceptually meaningful values would make these models to predict higher RWU from upper

layers (in accordance with $R$ distribution) for a longer period, increasing the discrepancy with VLM predictions. Therefore, their best fitted values are physically without meaning due to the model assumptions.

In order to interpret the parameters in Table 5 for JMf, one should first recall that $\alpha$ in JMf stands for the local RWU reduction due to soil hydraulic resistance. Thus, its $h_3$ parameter refers to the local soil pressure head at which RWU starts to reduce. It may be argued that RWU reduction occurs in drier soil conditions as $R$ increases, i.e., $h_3$ is more negative for higher $R$ (similarly as for FM and FMm). However, since JMf accounts for compensation, RWU is interpreted as a non-local process, and uptake from one layer depends on the water status and root properties from other layers (Javaux et al., 2013). Thus, $h_3$ parameter from JM is affected by other parts of the root zone. Predictions by VLM show that RWU reduction from the upper layers starts at less negative pressure head values as $R$ increases. Therefore, $h_3$ in JMf should increase with increasing $R$. The values of $h_3$ for JMf shown in Table 5 agrees to this conceptual meaning. The $M_c$ parameter from JMm can be interpreted likewise.

Values for $\omega_c$ from JMf for the high $R$–low $T_p$ scenarios equal 1, thus contradicting its conceptual meaning: as in these scenarios the compensation mechanism is more intense, $\omega_c$ should be less than one for the medium and high $R$ scenarios. The reason for $\omega_c = 1$ was discussed in section 4.1.2. Conversely, $\omega_c$ values for JMm follow the conceptual meaning.

The optimal parameters of the proposed models follow their logical relation to $R$ and $T_p$. The $l_m$ values for both models are very close. The optimal $l_m$ values are less sensitive to soil types and more sensitive to $R$.

High correlation parameters might result in uncertainties and nonunique solution of the optimization problem. In general, the correlation parameter coefficients were low, except for some scenarios in which high correlation coefficients between $\omega_c$ and $h_3$ (or $M_c$) were found. These high correlations maybe be due to model structure rather than to the data used for fitting the models, since the correlation for PM and PMm parameters were considerably low (absolute correlation coefficient below 0.53).

### 4.1.5    Optimization using $T_r$

The empirical models fitted only to RWU, since the primary interest is to evaluate the model's capability to predict the RWU patterns under different scenarios. RWU is not easily obtained in real conditions, making the use of RWU physical RWU models a great advantage. On the other hand, plant transpiration, one of the main outputs in RWU models, is more easily measured. Thus, one might consider to fit the models to the temporal course of (relative) plant transpiration or to fit the models simultaneously to both plant transpiration and RWU, for which a rather complicated optimization scheme would be required.

We addressed this issue by fitting the models to the course of relative transpiration for some scenarios. The procedure was the same as explained in Section 3.3, but substituting $S_{i,j}$ in eq 31 by $T_{r_i}$. The results for some models in two contrasting scenarios of $R$ are shown in Fig 11. Models that account for "compensation" can predict $T_r$ quite reasonably even when fitted to RWU only. The models that do not account for "compensation" do not mimic the $T_r$ course over time correctly for the high $R$ scenario predicted by VLM, even when they are fitted to $T_r$, and the predictive quality decreases when fitted to RWU. The most important aspect shown in Fig 11 is that fitting the models to $T_r$ can improve $T_r$ predictions but impairs their RWU predictions considerably, especially in high $R$ scenarios. Conversely, if a model fits well to RWU, it can provide suitable transpiration

predictions. This can also be seen by the analysis of Section 4.1.3, when the proposed models and JMm had good performance in predicting $T_r$ as well.

## 4.2   Growing season simulation

By evaluating the RWU models under real weather conditions during a relatively dry year and considering the same soil types and crop characteristics as for the drying-out experiment, it was possible to use the calibrated parameters for specific soil type and root length density. This evaluation is important to analyse whether our calibration of the empirical models with a single drying-out experiment results in consistent predictions for other circumstances. Models were not evaluated for the low $R$ scenario because the empirical models (except JMf and JMm) were not able to mimic those conditions for high $T_p$ (section 4.1.1)

Fig 12 shows the time course of cumulative actual transpiration simulated by SWAP using all the RWU models, together with rainfall and $T_p$ throughout the growing season period. Following the first dry spell, $T_{ac}$ predicted by FM and FMm, not accounting for "compensation", starts to be lower than predictions from other models. Two or three more dry spells occur in the evaluated period. The magnitude of the underestimation, however, varies with soil type and $R$. For the medium $R$–loam soil scenario, for instance, the $T_{ac}$ for all models are similar. The $T_{ac}$ at the end of the evaluated period predicted by VLM for low $R$ (not shown in Fig. 12) was much lower and approximately equal for the three soil types (40.45, 40.05 and 40.08 cm for clay, loam and sand soil, respectively). In fact, a higher $R$ resulted in an increasing difference of cumulative transpiration between soil types. Most water is extracted from the clay soil, followed by sand and loam. Little difference of cumulative transpiration is found between medium and high $R$: for sand and clay soil, the cumulative transpiration was slightly higher for high $R$; for the loam soil it was and practically identical.

Comparing cumulative $T_a$ predicted by the empirical models with VLM predictions shows that the models that do not account for compensation underestimate cumulative $T_a$ from 2.0 % (medium $R$ –sand soil scenario) to 13.9 % (high $R$–clay soil scenario). Overall, the highest underestimates occurred for high $R$. All other models predict similar values. Therefore, for total actual transpiration prediction, any of the evaluated models accounting for compensation might be suitable after calibration.

An overall analysis of model performance is shown in Fig. 13 and a list of the "best" model for each scenario based on AIC is shown in Table 7. The best performances are from the models that account for compensation. An improvement of JMf by using the proposed reduction function can be observed. Among the models that account for compensation, JMf had the worst performance. JMII also was poor in predicting RWU, but showed good performance in estimating plant transpiration. Overall, the best performances were also obtained by the proposed models (PM and PMm) and by the modified Jarvis (1989) model (JMm) in predicting RWU. These results also indicate that the strategy of designing a single drying-out experiment to calibrate an empirical model is successful.

According to the AIC, PM, PMm and JMm are best in predicting RWU. Regarding $T_r$ predictions, Fig. 13 shows considerably high statistical indices ($E$ and $r^2$) for all models that account for "compensation". However, the AIC, which penalizes the models with more parameters, indicates that JMII was the "best" model for most of the scenarios.

In general, the proposed models as well as JMm showed better performance than the other empirical models. It should be noted, however, that these models are based on $M$, making them closer to the physical De Jong van Lier et al. (2013) model. In this regard, it is important to separately compare JMf and JMm and PM and PMm. The only difference between JMf and JMm is the $\alpha$ reduction, which resulted in considerable improvements as discussed. In the proposed models, $M$ is included in $S_p(z)$ to distribute $T_p$ over depth. In PMm, $\alpha_m$ is used instead of the Feddes reduction function (used in PM). These simple modifications were sufficient to allow these empirical models to be fitted too mimic the predictions made by the more complex physical model.

## 5  Conclusions

Several simple RWU models have been developed over the years and we outlined some of these models and also proposed alternatives. Some of these models were embedded as sub-models into the eco-hydrological model SWAP (Van Dam et al., 2008) and their evaluation was based on the comparison with RWU predictions performed by the physical De Jong van Lier et al. (2013) model (also embedded into the SWAP model) for two numerical experiments with several scenarios of soil type, root length density and potential transpiration rates. The parameters of the empirical models were determined by inverse modelling of simulated RWU. The simulated scenarios also allowed insight into the behaviour of the De Jong van Lier et al. (2013) model, especially under wet soil conditions and high potential transpiration. In such scenarios and with a low $R$, the De Jong van Lier et al. (2013) model predicts crop transpiration reduction, as the maximum crop transpiration rate becomes dependent on crop hydraulic parameters, especially on the radial root hydraulic conductivity. More insight into these results may be obtained by coupling the De Jong van Lier et al. (2013) physical model to a stomatal conductance model. Regarding the performance of the empirical models we conclude:

- The widely-used Feddes et al. (1978) empirical RWU model only performs well under circumstances of low root length density $R$, in other words, when root water "compensation" is low. From medium to high $R$, the model cannot mimic properly the RWU dynamics as predicted by the De Jong van Lier et al. (2013) model, resulting in a poor performance. Moreover, the optimized $h_3$ values are counterintuitive when interpreting their conceptual meaning. Employing the proposed RWU reduction function (the FMm model) does not improve performance with this respect.

- The JMf model provides good predictions only for low and medium $R$ scenarios. For high $R$, the model cannot mimic the RWU patterns predicted by the De Jong van Lier et al. (2013) model. Using the proposed JMm reduction function helps to improve RWU predictions. Similarly, the JMII model does not perform well for high $R$–low $T_p$ scenarios, as explained in Section 4.1.2.

- The proposed models can predict RWU patterns similar to those obtained by the De Jong van Lier et al. (2013) model. The statistical indices point them as the best alternatives to mimic RWU predictions by the De Jong van Lier et al. (2013) model.

- Regarding the ability of the models in predicting plant transpiration, all models accounting for compensation have good performance. The AIC indicates that JMII is the "best model". This model is also more suitable for blind predictions, as no empirical parameters need to be estimated.

• The simulations of a growing season with grass confirmed these findings, suggesting that an experiment of soil drying-out for two levels of potential transpiration, as performed, is adequate to analyse the performance of RWU models and retrieve their empirical parameters by defining the objective function in terms of RWU.

It should be noticed that the predictions from the De Jong van Lier et al. (2013) physical model do not represent a real system. However, they show to be consistent with observed behaviour and have adequate sensitivity to variables and system boundaries. It is a common practice users to refer to the parameter compilation made by Taylor and Ashcroft (1972), which does not account for the dependence of the parameters on soil type. Moreover, these parameters depend on type of transpiration reduction function; although not explicit in the Taylor and Ashcroft (1972) compilation, it is usually considered to refer to the Feddes model. The best empirical models in predicting RWU, based on the comparison with the De Jong van Lier et al. (2013) physical model (the proposed models and JMm), have one additional parameter, also dependent on soil type, root length density and potential transpiration. Although the parameters for three soil types, root length density and potential transpiration are provided in this study, a more robust and complete calibration may be necessary, mainly because general values of plant hydraulic resistances were used. Due to the dependence of the empirical parameters on soil type and potential transpiration, parameterizing the selected empirical models for a specific crop might require more effort than when using the physical model with parameters that can be determined independently. The use of the physical model predictions, as in this study, seems a good strategy to calibrate the empirical models. Ultimately, the option for the empirical or physical model will be based on the desired complexity and understanding of the system, and on the availability of parameter values.

*Acknowledgements.* The first author thanks CAPES (The Capes Foundation, Ministry of Education of Brazil) and CNPq (National Council of Technological and Scientific Development, Brazil) for the PhD scholarship. The authors are grateful to Dr. N. Jarvis who acted as a reviewer and who, together with two anonymous reviewers, gave valuable comments and suggestions during the reviewing process.

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

## List of tables

**Table 1.** Summary of empirical models used in this study. $\alpha_f$ and $\alpha_m$ are the Feddes et al. (1978) (Fig. 1a) and proposed reduction functions (Fig. 1b), $S_p$ (eq. 4) is the potential root water uptake, $\omega$ (eq. 11) and $\omega_c$ are the weighted stress index and threshold value in Jarvis (1989) model and $\zeta_m$ (eq. 25) is the weighted stress index in the proposed models.

| Model | Acronym | Equation |
|---|---|---|
| Feddes et al. (1978) model | FM | $S(z) = S_p \alpha_f$ |
| Modified Feddes et al. (1978) model | FMm | $S(z) = S_p \alpha_m$ |
| Jarvis (1989) model | JMf | $S(z) = S_p \dfrac{\alpha_f}{\max\{\omega, \omega_c\}}$ |
| Modified Jarvis (1989) model | JMm | $S(z) = S_p \dfrac{\alpha_m}{\max\{\omega, \omega_c\}}$ |
| Jarvis (2010) model | JMII | Eqs. 11 to 13 with parameters given by eqs. 18 to 20 |
| proposed model I | PM | $S(z) = \zeta_m T_p \alpha_f$ |
| proposed model II | PMm | $S(z) = \zeta_m T_p \alpha_m$ |

**Table 2.** Values of the parameters of the De Jong van Lier et al. (2013) model used in the simulations. $h_{ws}$ is the limiting value $h_w$ in eq. 5 for the empirical models.

| Parameter | Value | Unit |
|---|---|---|
| $r_0$ | 0.5 | mm |
| $r_x$ | 0.2 | mm |
| $K_{root}$ | $3.5 \cdot 10^{-8}$ | m d$^{-1}$ |
| $L_l$ | $1 \cdot 10^{-6}$ | d$^{-1}$ |
| $h_{ws}$ | -150 | m |
| $h_{wl}$ | -200 | m |

**Table 3.** Mualem-van Genuchten parameters for three soils of the Dutch Staring series (Wösten et al., 1999) used in simulations. $\theta_s$ and $\theta_r$ are the saturated and residual water content, respectively; $K_s$ is saturated hydraulic conductivity and $\alpha$, $\lambda$ and $n$ are fitting parameters.

| Staring soil ID | Textural class | Reference in this paper | $\theta_r$ | $\theta_r$ | $K_s$ | $\alpha$ | $\lambda$ | $n$ |
|---|---|---|---|---|---|---|---|---|
| | | | m m$^{-3}$ | m m$^{-3}$ | m d$^{-1}$ | m$^{-1}$ | - | - |
| B3 | Loamy sand | Sand | 0.02 | 0.46 | 0.1542 | 1.44 | -0.215 | 1.534 |
| B11 | Heavy Clay | Clay | 0.01 | 0.59 | 0.0453 | 1.95 | -5.901 | 1.109 |
| B13 | Sand Loam | Loam | 0.01 | 0.42 | 0.1298 | 0.84 | -1.497 | 1.441 |

**Table 4.** Parameters of the root water uptake models estimated by optimization and their respective constraints $\Omega$.

| Model | Parameter | $\Omega$ | Unit |
|-------|-----------|----------|------|
| FM | $h_3$ | $-150 < h_3 < 0$ | m |
| FMm | $M_c$ | $0 < M_c < M_{max}$ | $m^2\ d^{-1}$ |
| JMf | $h_3$ | $-150 < h_3 < 0$ | m |
| | $\omega_c$ | $0 < \omega_c \leq 1$ | - |
| JMm | $M_c$ | $0 < M_c < M_{max}$ | $m^2\ d^{-1}$ |
| | $\omega_c$ | $0 < \omega_c \leq 1$ | - |
| PM | $h_3$ | $-150 < h_3 < 0$ | m |
| | $l_m$ | $0 < l_m \leq 3$ | - |
| PMm | $M_c$ | $0 < M_c < M_{max}$ | $m^2\ d^{-1}$ |
| | $l_m$ | $0 < l_m \leq 3$ | - |

**Table 5.** Optimal parameters of each empirical model for all scenarios in the drying-out experiment

| | | | FM | FMm | JMf | | JMm | | PM | | PMm | |
|---|---|---|---|---|---|---|---|---|---|---|---|---|
| Soil | Tp | $R$ | $h_3$ | $M_c$ | $h_3$ | $\omega_c$ | $M_c$ | $\omega_c$ | $h_3$ | $l_m$ | $M_c$ | $l_m$ |
| | mm d$^{-1}$ | cm cm$^{-3}$ | cm | cm$^2$ d$^{-1}$ | cm | - | cm$^2$ d$^{-1}$ | - | cm | - | cm$^2$ d$^{-1}$ | - |
| clay | 1 | 0.01 | -1968.7 | 0.213 | -284.5 | 0.711 | 0.366 | 0.494 | -1615.7 | 1.322 | 0.227 | 1.290 |
| clay | 1 | 0.10 | -1211.0 | 0.329 | -132.4 | 0.196 | 0.944 | 0.024 | -7579.9 | 0.869 | 0.076 | 0.884 |
| clay | 1 | 1.00 | -1.7 | 0.950 | -0.0 | 1.000 | 5.971 | 0.004 | -10673.7 | 0.354 | 0.022 | 0.342 |
| loam | 1 | 0.01 | -7588.1 | 0.334 | -5.0 | 0.457 | 22.483 | 0.016 | -6927.6 | 1.086 | 0.408 | 1.084 |
| loam | 1 | 0.10 | -6085.6 | 0.487 | -93.9 | 0.126 | 25.721 | 0.002 | -11795.6 | 0.911 | 0.113 | 0.917 |
| loam | 1 | 1.00 | -17.0 | 5.014 | -48.0 | 1.000 | 106.223 | 0.000 | -10878.8 | 0.561 | 0.058 | 0.553 |
| sand | 1 | 0.01 | -1014.0 | 0.146 | -291.6 | 0.942 | 0.288 | 0.436 | -621.2 | 1.262 | 0.149 | 1.252 |
| sand | 1 | 0.10 | -1122.6 | 0.115 | -113.6 | 0.407 | 1.925 | 0.005 | -2351.3 | 1.179 | 0.024 | 1.159 |
| sand | 1 | 1.00 | -3.9 | 0.338 | -0.0 | 1.000 | 25.887 | 0.000 | -3158.0 | 0.717 | 0.005 | 0.706 |
| clay | 5 | 0.10 | -1397.7 | 0.334 | -218.4 | 0.325 | 0.395 | 0.271 | -5537.2 | 1.512 | 0.196 | 1.449 |
| clay | 5 | 1.00 | -260.6 | 0.792 | -135.3 | 0.148 | 1.212 | 0.013 | -6745.0 | 0.672 | 0.088 | 0.687 |
| loam | 5 | 0.10 | -5236.5 | 0.784 | -0.0 | 0.277 | 2.306 | 0.100 | -8322.9 | 1.165 | 0.488 | 1.157 |
| loam | 5 | 1.00 | -1249.5 | 2.563 | -292.9 | 0.161 | 28.143 | 0.001 | -8630.0 | 0.833 | 0.224 | 0.838 |
| sand | 5 | 0.10 | -918.0 | 0.190 | -556.2 | 0.432 | 4.154 | 0.018 | -1273.9 | 1.612 | 0.083 | 1.510 |
| sand | 5 | 1.00 | -582.3 | 0.533 | -342.5 | 0.193 | 4.888 | 0.001 | -3582.3 | 1.272 | 0.012 | 1.240 |

**Table 6.** Best models for the evaluated scenarios (root length density $R$, soil type and potential transpiration $T_p$) based on Akaike's information criteria AIC through comparison of root water uptake (RWU) and relative transpiration ($T_r$) predicted by De Jong van Lier et al. (2013) physical model in the drying-out experiment.

|  | | Low $T_p$ | | | High $T_p$ | | |
|---|---|---|---|---|---|---|---|
|  | $R$ | Clay | Loam | Sand | Clay | Loam | Sand |
| RWU | Low | JMm | JMf | JMm | JMm | JMm | JMm |
| | Medium | PMm | PMm | JMII | JMm | PM | PMm |
| | High | PMm | PMm | PM | PM | PMm | PM |
| $T_r$ | Low | JMm | JMm | JMm | JMm | JMm | JMm |
| | Medium | JMm | JMm | JMII | JMm | PM | JMf |
| | High | PMm | PMm | PMm | JMII | JMm | JMm |

**Table 7.** Best models for the evaluated scenarios (root length density $R$ and soil type) based on Akaike's information criteria AIC through comparison of root water uptake (RWU) and relative transpiration ($T_r$) predicted by De Jong van Lier et al. (2013) physical model in the growing season experiment.

| | Clay | | Loam | | Sand | |
|---|---|---|---|---|---|---|
| | Medium R | High R | Medium R | High R | Medium R | High R |
| RWU | JMm | PM | PM | PMm | JMm | JMm |
| $T_r$ | JMII | JMII | JMf | JMm | JMII | JMII |

**List of figures**

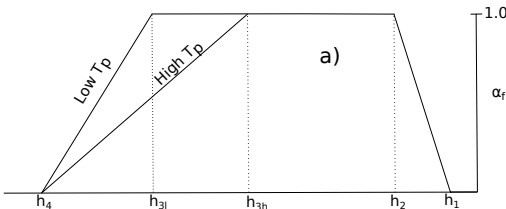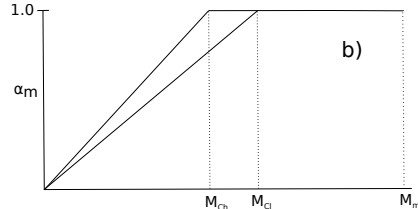

**Figure 1.** a) Feddes et al. (1978) root water uptake reduction function. $h_2$ and $h_3$ are the threshold parameters for reduction in root water uptake due to oxygen deficit and water deficit, respectively. The subscripts $l$ and $h$ stands for low and high potential transpiration $T_p$. $h_1$ and $h_4$ are the soil pressure head values above and below which root water uptake is zero due to oxygen and water deficit, respectively. b) Root water uptake reduction function $\alpha_m$ as a function of matric flux potential $M$; $M_{ch}$ and $M_{cl}$ are the critical values of $M$ for high and low $T_p$, respectively, from which the uptake is reduced and $M_{max}$ is the maximum value of $M$, dependent on soil type.

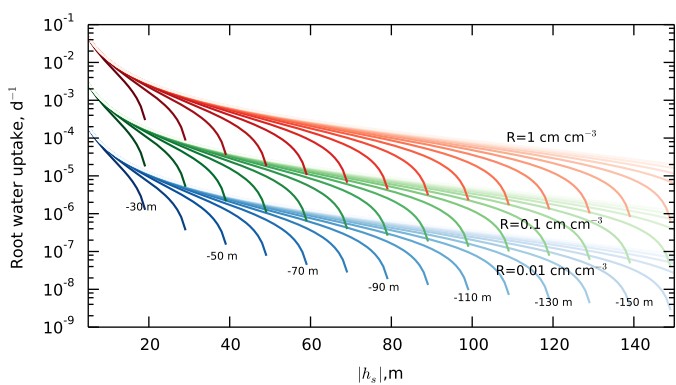

**Figure 2.** Root water uptake (RWU) as a function of soil pressure head $h_s$ for three values of root length density (0.01, 0.1 and 1.0 cm cm$^{-3}$) and leaf pressure head values ranging from -30 to -200 m by -10 m interval shown by colour gradient (lighter colours indicate lower values, some values are indicated in the plot). Results were obtained by the analytical solution of eq. 8 given by De Jong van Lier et al. (2013) for a special case of Brooks and Corey (1964) soil. Plant transpiration was set to 1 mm d$^{-1}$ and the soil and plant hydraulic parameters were taken from De Jong van Lier et al. (2013).

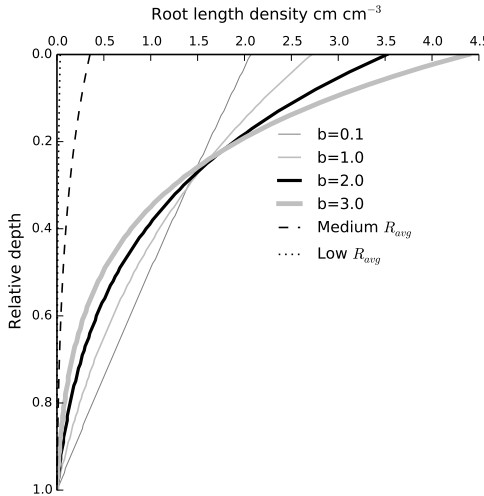

**Figure 3.** Root length density distribution over depth calculated by eq. 30 for several values of $b$ and $R_{avg}$ = 1.0 cm cm$^{-3}$ and for low and medium $R_{avg}$ with $b = 2$.

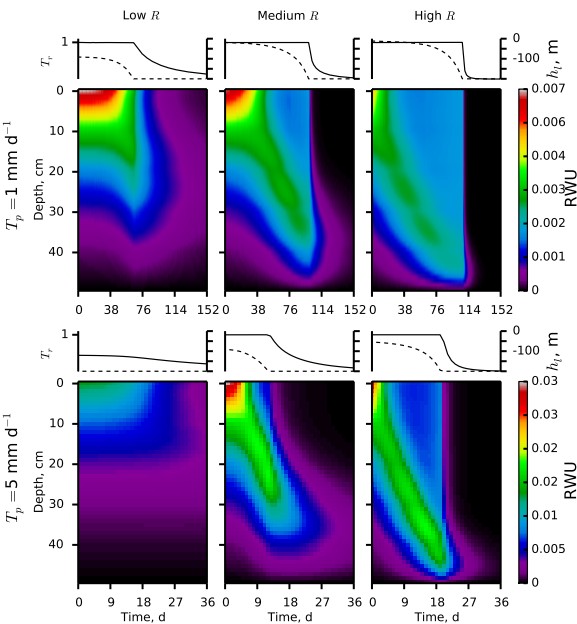

**Figure 4.** Time-depth root water uptake (RWU, $d^{-1}$) pattern, leaf pressure head ($h_l$, dashed line) and relative transpiration ($T_r$, continuous line) simulated by SWAP model together with the De Jong van Lier et al. (2013) model for clay soil, two levels of potential transpiration $T_p$: 1 and 5 mm $d^{-1}$ (first and second line of plots, respectively) and three levels of root length density $R$: low, medium and high (indicated at the top of the figure).

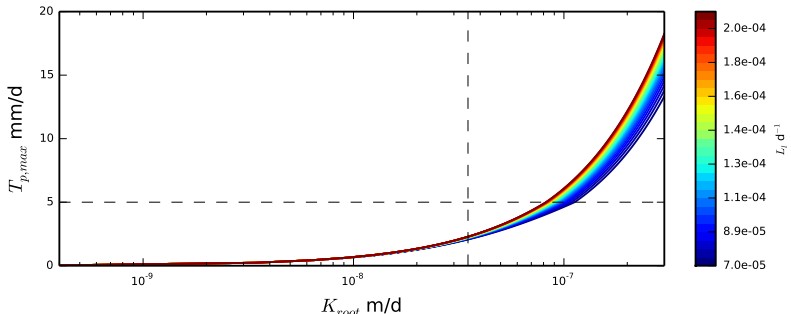

**Figure 5.** Maximum possible transpiration $T_{p,max}$ as a function of root hydraulic conductivity $K_{root}$ for some values of the overall conductance over the root-to-leaf pathway $L_l$ computed by De Jong van Lier et al. (2013) model for rooting depth of 0.5 m, low root length density and constant soil pressure head over depth equals to -1 m for sandy soil. The dashed vertical line highlights the value of $K_{root} = 3.5\ 10^{-8}$ m d$^{-1}$ that was used in our simulations. Horizontal dashed line highlights the value of potential transpiration.

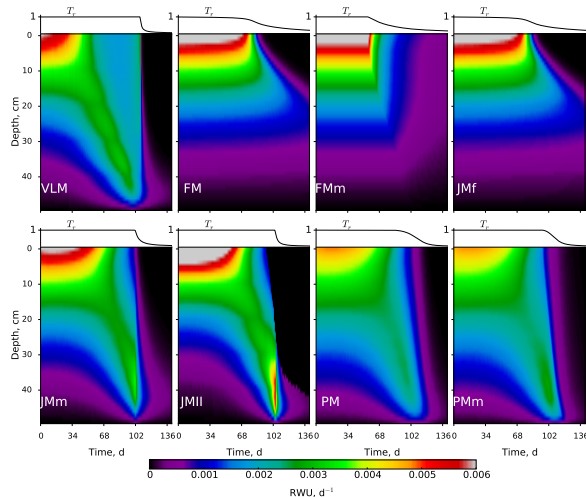

**Figure 6.** Time-depth root water uptake (RWU) pattern and relative transpiration ($T_r$) simulated by SWAP model in combination to the De Jong van Lier et al. (2013) reduction function and the empirical models, for the sand soil at high root length density and $T_p = 1$ mm d$^{-1}$.

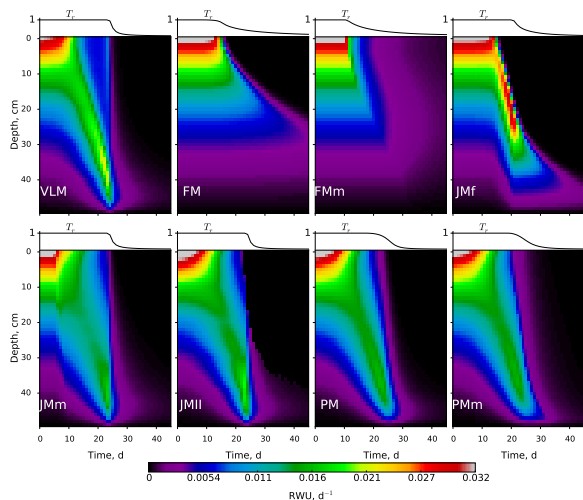

**Figure 7.** Time-depth root water uptake (RWU) pattern and relative transpiration ($T_r$) simulated by SWAP model in combination to the De Jong van Lier et al. (2013) reduction function and the empirical models, for the sand soil at high root length density and $T_p = 5$ mm d$^{-1}$.

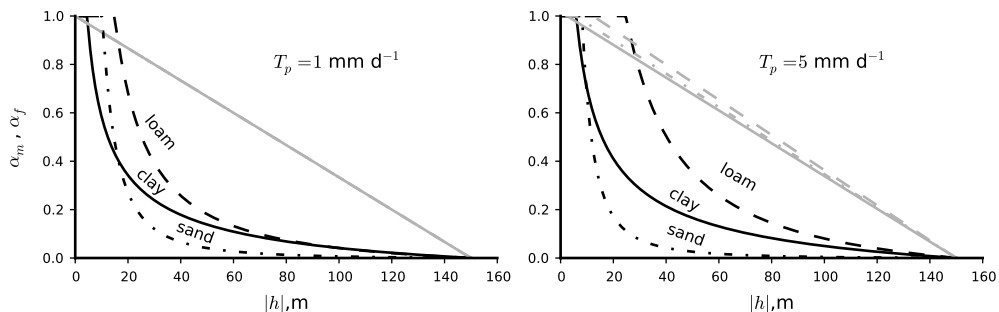

**Figure 8.** Feddes et al. (1978) ($\alpha_f$, gray lines) and proposed ($\alpha_m$, black lines) water uptake reduction functions as a function of soil pressure head $h$ using their respective optimized parameters for the scenario of high root length density, three types of soil and two potential transpiration levels.

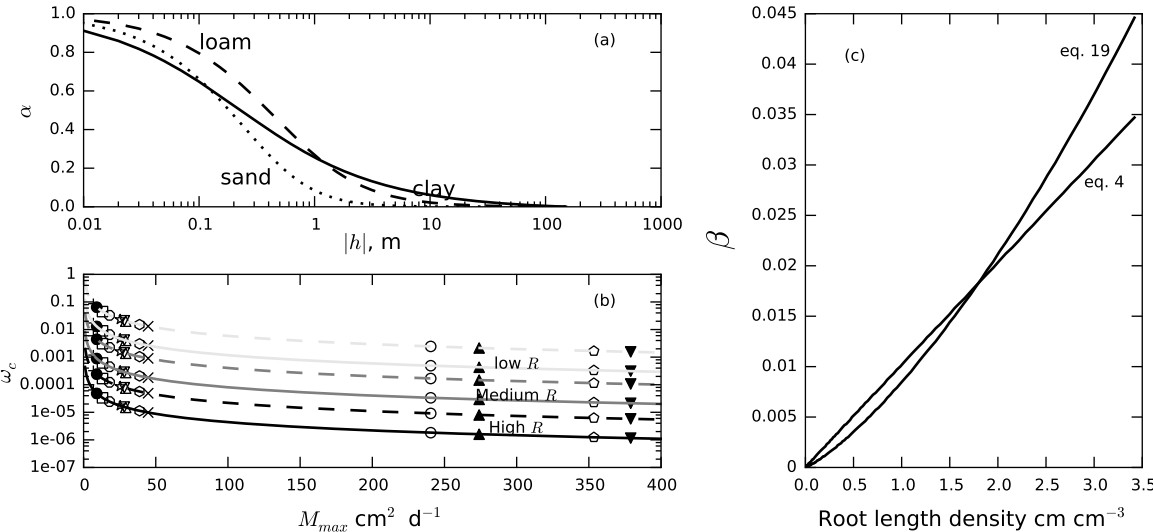

**Figure 9.** (a) $\alpha$ of JMII model (eq. 18) as function of soil pressure head $h_s$, (b) $\omega_c$ parameter (eq. 20) for different soil types (the three soil types used in the simulations and more soils from Wösten et al. (1999) ), expressed by $M_{max}$ and (c) the normalized root length density $\beta$ computed by eqs. 4 (JMf) and 19 (JMII) as function of root length density $R$, given by eq. 30 with $R_{avg} = 1.0$ cm cm$^{-3}$ and $b = 2$.

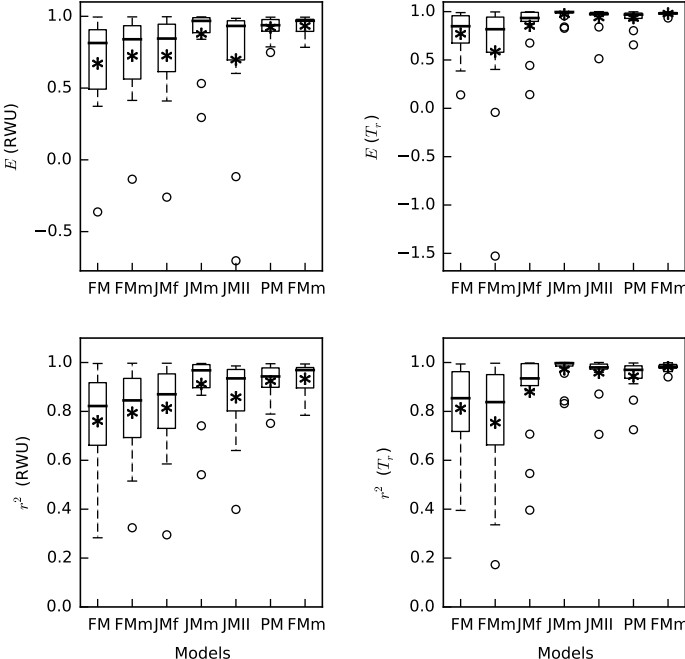

**Figure 10.** Box plot of the coefficient of determination $r^2$ and model efficiency coefficient $E$ for the comparison of root water uptake (RWU) and actual transpiration ($T_a$) predicted by the empirical models and the De Jong van Lier et al. (2013) model for the drying-out simulations at three levels of root length density for three types of soil and two potential transpiration levels. The symbols $*$ and $\circ$ represent the average and outliers, respectively.

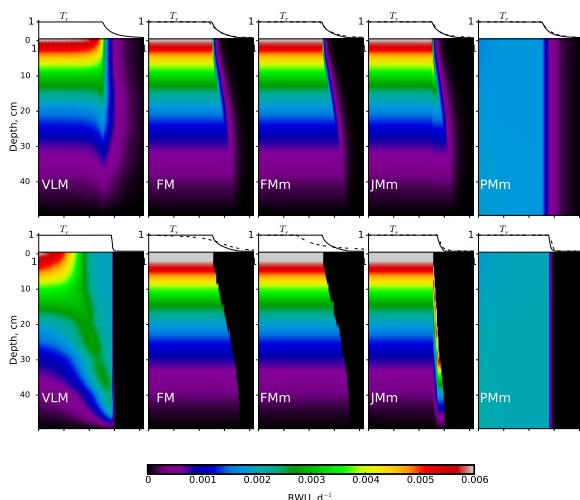

**Figure 11.** Time-depth root water uptake (RWU) pattern and relative transpiration ($T_r$) simulated by SWAP model in combination to the De Jong van Lier et al. (2013) reduction function and evaluated empirical models optimized performed $T_r$ instead of RWU for loam soil, low (first line of plots) and high (second line of plots) root length density and $T_p = 1$ mm d$^{-1}$. The dashed lines indicate $T_r$ when the models were optimized with RWU.

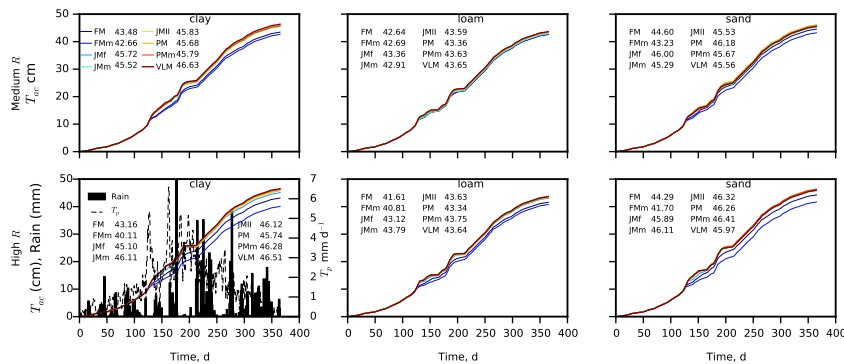

**Figure 12.** Time course of actual cumulative plant transpiration $T_{ac}$ predicted by the De Jong van Lier et al. (2013) model and empirical models for three soils (clay, loam and sand) and two levels of root length density (medium and high), together with rainfall and potential transpiration $T_p$ for the growing season experiment. The total $T_{ac}$ values predicted by each model for the whole period are shown in the plot aside the model names.

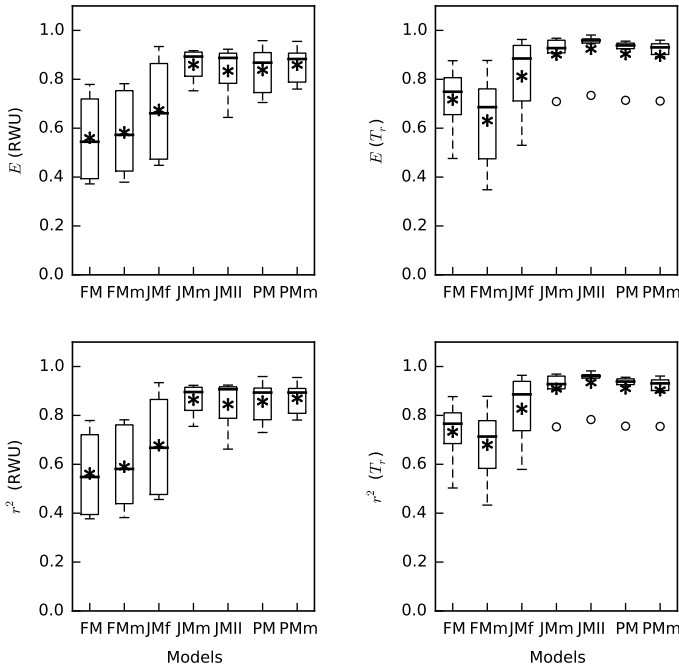

**Figure 13.** Box plot of the coefficient of determination $r^2$ and model efficiency coefficient $E$ for the comparison of root water uptake (RWU) and actual transpiration ($T_a$) predicted by empirical models compared to De Jong van Lier et al. (2013) model predictions, for the growing season experiment, two levels of root length density and three soils. The symbols $*$ and $\circ$ represent the average and outliers, respectively.