# Peer review of "Benchmarking test of empirical root water uptake models"

_Hydrology and Earth System Sciences, 2016_

## Referee Comment (RC1) · N. Jarvis (Referee) · 4 Mar 2016

This paper presents the results of a comparative modelling exercise: the performances of several empirical root water uptake models are compared against a common benchmark, a physics-based model of water uptake and transpiration for the complete soil-plant-atmosphere system. I like the approach taken, because I think the best way to evaluate models is in a comparative test framework.

However, the results of this kind of exercise are sometimes tricky to interpret and I do have a few concerns in this respect. The conclusion that modified versions of the Li model are recommended seems quite shaky, since for some scenarios the derived value for the lambda parameter seems to take non-physical values (point 16). The model works well here because you are calibrating against a physical model, but how

much confidence can you have in blind predictions with this model, which only has a weak physical basis? I also miss a discussion and explanation of why this model performs best even though it lacks a sound physical basis (see points 7 and 10). I would also like to see a fuller discussion of why the empirical models JMm and PMm give simulations that better match the complete physical model of the soil-plant system, VLM, than the model JMII, which has a physics-based treatment of the soil that is, in principle, identical to that of VLM, but which excludes plant resistances (see points 13 and 19). Is it because introducing a threshold function of matric flux potential (with Mc as the critical value) for local uptake in the empirical models mimics the fact that a constant plant resistance dominates the overall resistance to water uptake in the early stages of soil drying? When M becomes less than Mc, then the soil resistances start to play a more important role. This may be the same reason why the original Jarvis (1989) model, which gives the local resistance as a threshold function of saturation, also seems to work quite well in practice. But there may be other reasons (one other likely candidate is mentioned in comment 13). I also wonder if it is fair to compare an uncalibrated model (JMII) with calibrated models in this way. Perhaps the number of calibrated parameters should be considered in a comparative assessment of model performance? For example, would it be better to assess model performance using the Akaike information criterion, which penalizes models with more parameters? All these aspects should be addressed.

Another less critical question I have is that you are treating the soil hydraulic parameters as fixed and known. This is OK for your particular modelling exercise, but in reality this will not be the case. Uncertainty in the hydraulic parameters is likely to affect the outcome of the comparison of different root water uptake models. Their performance may become indistinguishable even with only moderate uncertainty in these (and other) model parameters. This could be briefly discussed.

I am not so keen on the title of the paper, as it doesn't really reflect the contents so well: two alternative suggestions are: "Comparative test of empirical root water uptake

models" and "Benchmarking test of empirical root water uptake models".

The specific comments in the attached file should also be addressed.

Please also note the supplement to this comment:
http://www.hydrol-earth-syst-sci-discuss.net/hess-2016-59/hess-2016-59-RC1-supplement.pdf

**Supplement:**

Scientific/technical issues

1. Page 2, lines 24-25 (and line 3 in the abstract): I am not so convinced of this. I would prefer to use a physics-based model even if it did have two or three more parameters, as long as they were, in principle, measurable. The limiting leaf water potential is quite well known, at least.

2. Page 4, lines 6-8: the sentence starting … "Using h seems …." is wrong, as the authors know well enough. It is immediately contradicted by the text at lines 14-21 on the same page (and by the results shown later in the paper). This sentence should be deleted. The remaining text just says that various forms have been proposed for the $\alpha$ function, but that making $\alpha$ depend on M is physically the most plausible. This is quite sufficient.

3. Page 5, equation 8: ho(z) is not defined, as far as I can see?

4. Page 5, lines 20-22: yes, it would be good if you mentioned this phenomenon by its name: hydraulic lift or hydraulic re-distribution. You could also cite Jarvis (2011) here, since he discussed and clarified the relationship between water uptake compensation and hydraulic lift in some detail (see the text in relation to equations 13 to 15 in the final version of this paper, not the HESS discussion paper that you cited: see point 4. under "Presentation")

5. Page 6, lines 1-3: I know what you are trying to say here, but it is not so well expressed. You could replace i.) " …. is only relevant" by "…. it only needs to be explicitly addressed …." and ii.) "… becomes less important" by "… is not necessary". This would help, but you could also add a sentence at the end of this saying that the effects of compensation can nevertheless be explicitly discriminated and identified in physics-based models. This is demonstrated in Jarvis (2011) in the text related to equations 13 and 14 in that paper (again, in the final version)

6. Page 6, lines 11-12: "In principle, any definition of $\alpha$ is applicable…". Yes, perhaps, but it does make a difference to the results of course, as you demonstrate very well later in the paper! But what is definitely not debatable is that Jarvis (1989) used a threshold type function for $\alpha$ based on water content (degree of saturation). The reason for adopting this approach was discussed by Jarvis (1989) in relation to the experimental evidence available at that time and no other type of function was considered. The fact that you adopt a Feddes-type function means that in the rest of the paper you cannot refer to this model as the Jarvis (1989) model. It is a modified Jarvis (1989) model, in exactly the same way that JMm is also a modified Jarvis (1989) model, where the threshold water content function is replaced by a threshold function of matric flux potential: in other words, you investigated two different modified Jarvis models and you should refer to them as such, both in table 1 and throughout the rest of the paper, including the abstract (perhaps you could call them JMm1 and JMm2?).

7. Page 6, line 28 to page 7, line 4: this is a little vague. You followed quite closely what Skaggs et al. (2006) wrote in this section, but since they wrote their paper ten years ago, it is now much better established exactly how the original Jarvis (1989) model departs from physicality. This was clarified in the papers by Jarvis (2010, 2011), which you also discuss in the following section. There are two aspects to this:

i.) the choice of function for $\alpha$. The threshold function chosen by Jarvis (1989) doesn't make complete physical sense, as the local resistance to uptake should in principle increase continuously as the soil dries (e.g. like equation 18). Jarvis (1989) discussed this choice in terms of the overall resistance to uptake being dominated by an air gap between soil and root which might only develop after a certain critical water deficit was reached: this choice was strongly influenced by experimental studies which showed such an effect. Also, at high soil water contents, the overall resistance to uptake in the soil-plant system would be dominated by plant resistances, which may be more or less constant. Thus, a threshold function might be a good choice from an empirical point of view. In this respect, it can also be pointed out here that the authors also adopt a threshold $\alpha$ function in the PMm model. This model is the one the authors finally recommend, because it works best, although it can certainly be criticized on the same grounds (i.e. that it "affronts the definition of $\alpha$").

ii.) Compensation under non-stressed conditions. As you point out, under non-stressed conditions the Jarvis (1989) model does give a different uptake distribution compared with the de Jong van Lier physical model. However, it is wrong to imply that the Jarvis (1989) model does not predict any compensation under non-stressed conditions (page 7, line 4). Under non-stressed conditions, water uptake is increased by a factor of $1/\omega$ in all layers (regardless of the pressure head distribution) to maintain transpiration at the rate demanded by the atmosphere during soil drying. It is also not wrong in principle to link compensation to plant stress (page 7, line 3): the onset of stress certainly does affect the nature of compensation: this is demonstrated in Jarvis (2011) in the text following equations 13 and 14 for the physics-based model of de Jong van Lier (2008).

For the above reasons, I strongly suggest that you delete the text on page 6 line 28 to page 7, line 4 and replace it by a short sentence that simply states that the Jarvis (1989) model departs from complete physicality in some respects and that this is explained in the following section. Then at the end of the next section (i.e. after equation 21) you can briefly summarize how the Jarvis (1989) model departs from physicality, based on the comparison with the physics-based model that is represented by equation 14-21. This will be very much clearer.

8. Page 7, lines 5-12. The parameter $h_3$ does not exist in the Jarvis (1989) model (see lines 10-11 especially). I think this paragraph can be deleted (or perhaps moved to the results and discussion section). At the very least, readers should be reminded that the original Jarvis (1989) model does not use a Feddes-type $\alpha$ function.

9. Page 8, line 22: you should add the limits for $\lambda$ here. If compensation means that water uptake increases from sparsely rooted layers, then $\lambda$ must lie between zero and 1. Also, you should replace "deeper soil layers" by "more sparsely rooted layers" to be strictly correct.

10. Page 9, lines 23 to 26: I wonder what it is about your modification to the Li model (the use of the matric flux potential in a threshold function) that resolves the conceptual difficulties with the original formulation that you described earlier on page 9 at lines 3 to 8. As far as I can

see, the same objections should be equally valid for this modified version as for the original model. This should be clarified and the text modified accordingly.

11. Page 11, lines 13-28: As I understand it from table 3, you only have a maximum of two parameters to calibrate for all the models, while each parameter is constrained within known limits. This means that a "brute force" grid search for optimum parameter values would be preferable to the method you chose, since you could be sure of avoiding risks of finding local minima (although it might be slower). I am sure there is no need to repeat the calibrations, but maybe you could mention this?

12. Page 13, lines 23-24: yes, this may be why a constant value of $\omega_c$ often seems to work quite well. Maybe you could add a comment to this effect, and also refer to your equation 20 and cite Jarvis (2011), where this aspect is discussed in detail.

13. Page 15, line 18: You should replace "either R or M" by "both R and M". But this sensitivity to M is in principle also present in the empirical models that include M. Why is it more important for JMII? Is it because this model is not calibrated? Or is it because of the different type of function? I can believe that predictions of JMII are, in comparison with the empirical models, more affected by the value of $M_{max}$, which must be a very uncertain parameter, not least because the Mualem-van Genuchten model of soil hydraulic properties is known to have an incorrect form close to saturation (since it does not allow for a maximum size of pore in soil). These questions should be clarified.

14. Page 15, line 25: it could also be noted (perhaps by referring to equation 20) that $\omega_c > 1$ is not physically unrealistic.

15. Page 16, line 3: This is misleading. The Feddes function for $\alpha$ is not part of the Jarvis (1989) model.

16. Page 17, lines 1-2: it is confusing that different symbols are apparently used for one of the parameters in the Li-type models. In equation 25, $\lambda$ is used, whereas in the text here and in table 5, $l$ is used, while in table 4 $l_m$ is used. I believe they are all the same parameter?

   If I understood it correctly, I don't see how you can write that the optimal values of $\lambda$ follow a logical relation to R and Tp (line 1). In many cases, and especially for low root densities, values of $l$ (i.e. $\lambda$?) in table 5 are larger than 1, which implies to me that compensation is working incorrectly in these scenarios (it is decreasing uptake in the more sparsely rooted layers). Also, in table 4, it is stated that $l_m$ (i.e. $\lambda$) was constrained to take values less than or equal to 1. If I understood it correctly, the results in table 5 suggest that this was not actually the case in practice.

17. Section 4.2, table 6: can you give the total precipitation and potential transpiration here? It's good to get a rough idea of how much stress occurred in these simulations.

18. Page 18, lines 10-12: you did not test the Jarvis (1989) model (see earlier comments).

19. Page 18, line 12: I did not get a good understanding of why the JMII model does not work so well for high R–low Tp scenarios (i.e. high compensation). I would have thought that, in principle, it should work OK. Please briefly explain what you think the reasons are for this.

20. Lines 16-18: I think this is too optimistic, as this test was not a very tough one. You had the same plants (identical roots) and the same three soils. How would it look if you had simulated different scenarios (soils, plants)? I think you would need to re-calibrate the empirical models. How useful is that?

Presentation

1. Abstract, lines 13-14: "Incorporating a newly proposed reduction …". It is not clear to me what you mean by this sentence.

2. Page 2, line 5: you could replace …"derived from" by …"extensions of".

3. Page 2, line 11: "Accordingly, plant water uptake increases …" would be better.

4. Page 2, line 16: in the reference list, you have cited the HESS discussion paper for Jarvis (2011). This must be replaced by the final published version of the paper. The author and article title are the same, but the volume and page numbers must be changed to volume 15, pages 3431 to 3446.

5. Page 2, line 28: delete "… quite incomprehensible and… ". I am not sure what you mean by this, but it's not needed anyway. It's enough to say the limitations are not well understood.

6. Page 5, lines 17-19: yes, this is important. It is discussed in Jarvis (1989, 2011), which you could cite to support this paragraph.

7. Page 5, line 19: it would be better to replace "achieved" by "maintained"

8. Page 5, line 23: the end of this sentence (starting with … "and that it can be…") is confusing and not necessary. It can be deleted.

9. Page 6, line 27: yes, but this could be written a little bit better as: "Equation 12 describes an analogy to stomata functioning (Jarvis, 1989, 2011), giving this model some physical basis. This is demonstrated in the following section."

10. Page 7, line 14: you could replace "numerically" by "mathematically". As you show, it's the actual equations that can be made identical, not just the results of calculations.

11. Page 8, line 16: add "(Jarvis, 2011)" after "conditions"

12. Page 9, line 2: don't you mean the Jarvis (1989) model? I think so, because Jarvis (2011) focuses almost exclusively on the de Jong van Lier (2008) model.

13. Page 10, line 24: data

14. Page 16, line 32: you could refer the reader to equation 20 to illustrate this. It would also be better to write "smaller" than "less than one".

15. Page 18, line 13: using the word "predicting" is a little misleading here. It would be better to write "matching". You are just calibrating against another model. Prediction is a whole different ball-game!

---

## Referee Comment (RC2) · Anonymous Referee #2 · 15 Mar 2016

This paper compares different root water uptake models. In a theoretical part, the relations between the different models, their parameters and differences between the models are derived. Simulations for a dry out scenario and for a growing season in different soils, for different transpiration rates and for different root densities were carried out. The benchmark model that was used was the de Jong van Lier 2013 model, which describes root water uptake in a mechanistic manner. The root water uptake distributions that were simulated with this model were subsequently used to parameterize the empirical models.

Despite the fact that root water uptake is one of the key processes in land surface models and crop models, it is one of the processes about which there is still a lot of uncertainty on how it should be represented in models. This paper makes an important contribution to this problem by making a detailed comparison between different approaches and by presenting new empirical models that could be used to represent root water uptake. Therefore, I think that this paper will make an important contribution to root water uptake modelling.

In general, the paper is well written and is technically sound. Yet, given that it is a very technical paper, I think that it can still be improved at some points so that it becomes clearer. Many of my detailed comments are questions for clarification. One point that needs to be clarified is the dependence of the model parameters on the transpiration rate. The authors address this topic in the discussion of the results. But, also in the theoretical part, I think that this point should be addressed.

The authors introduced a new empirical model for root water uptake. I would propose to include also some arguments why such an empirical model would be needed or beneficial as compared to the mechanistic model. Is this an issue of computational time?

In the conclusions section, I think that a general discussion on the parameters of the empirical model could be included. The problem with empirical models, which is not addressed, is that the 'root water uptake parameters' also seem to depend on the soil properties and the climatic conditions. The authors addressed the dependence of h3 on the transpiration rate and found that it was opposite to what is expected. I would suggest to address also the variation of the other parameters a bit.

Since the parameters of the empirical models depend on soil properties and boundary conditions, it means that these parameters have to be estimated for each specific case. In the paper, the authors parameterized the models based on simulated root water uptake distributions. The problem is that it is not possible to retrieve root water uptake distributions directly from measurements since there is also considerable water redistribution in the soil. I would propose to include a strategy how to deal with this problem.

Detailed comments.

P4: root length density R. Shouldn't that have dimension $L\,L^{-3}$?

P4: The authors propose a stress function $\alpha$ which is a stepwise linear function of M. Since M is a function of h, the new stress function will be a function of h also. But the shape of the function will have a different shape than a piecewise linear function of h. Furthermore, the relation between the new stress function $\alpha$ and h will depend on the hydraulic soil properties and will therefore be different in soils with a different texture. The original Feddes $\alpha(h)$ function depends on the

transpiration rate as shown in Figure 1. Figure 1 suggests that the new stress function $\alpha(M)$ does not depend on the transpiration rate. I do not understand why the transpiration dependency of the stress function disappears when $\alpha$ is expressed as a function of M since M does not depend on the transpiration rate.

P5: ln 15: 'Because Ta and hl are unknowns, eq. 8 and 10 cannot be solved analytically, but an efficient numerical algorithm is described in De Jong van Lier et al. (2013).' I did not understand this. I thought that either Ta=Tp is known as a boundary condition so that hl can be calculated or hl=hw is known and Ta is calculated. I think that the reason why the hl (or Ta) cannot be derived directly is because the set of equations that needs to be solved (including also all ho,i 's) is non-linear in ho,i.

P5 ln 17 and p 29 Figure 2: There are several things I do not understand about Figure 2. The figure caption says that the plant transpiration was set to 1 mm d$^{-1}$. Shouldn't for a fixed rooting depth the root water uptake or sink term S be constant and independent of the root length density R until a threshold soil water potential is reached? This threshold will depend of course on the leaf water potential and the root length density. Can it be that the curves shown in Figure 2 shown the maximal possible sink term as a function of the soil water potential for different leaf water potentials and root length densities? But, when the root water uptake goes to zero, why doesn't the soil water potential then go to the leaf water potential? Now there seems to be a 10 m difference between them. Second, why doesn't the root water uptake for a certain soil water potential then not increase with decreasing leaf water potential. For sufficiently large (small absolute value) soil water potential, the root water uptake becomes independent of the leaf water potential. I do not understand this since the water potential difference increases with decreasing leaf water potential and therefore the root water uptake should also increase with decreasing leaf water potential.

P7 ln 21: 'where Tpmax is the maximum possible transpiration rate attained when M0 = 0'. This assumes that the minimal water potential at the soil-root interface is hw (wilting point). But, doesn't this minimal water potential depend also on the critical leaf water potential hl?

P7 Eq. 17: Why is M0 constant with z? The soil root interface water potential can depend on the depth, can't it?

P 8 ln 15: 'The Jarvis (1989) model predicts RWU by a weighting factor between $\rho$ and M throughout rooting depth.' This is not very clear to me. What do you mean with a weighting factor 'between $\rho$ and M'? Do you mean a weighting factor that is equal to the product of $\rho$ and M?

An interesting feature of the analogy between the Jarvis model and the De Jong van Lier et al. (2008) model is that the analogy is derived based on the assumption that stress only occurs when everywhere at the soil-root interface limiting conditions are reached. It is assumed that M0 is zero everywhere in the root zone. But, I am wondering whether the De Jong van Lier et al. (2008) only

predicts stress under these conditions. Can it be that stress occurs even though M0(z) is not zero everywhere in the root zone? If this is the case, then the analogy between the Jarvis and the De Jong van Lier et al. (2008) models is not given always when stress occurs.

P 8 ln 22: 'The smaller $\lambda$, the more water is taken up in deeper soil layers' I would reword this to '… the more water is taken up from layers with a low root length density'.

P 9 ln 1: 'RWU is calculated by substituting eq. 23 into eq. 3, following the Feddes approach.' This implies that you multiply Eq. 22 again by a(z). So in the nominator, you get $\alpha(z)^2$?

P9 ln 16: Same comment as above.

P 9 ln 18: 'In drier soil layers, $\Gamma$ is reduced, whereas in wetter soil layers $\Gamma$ is increased, thus increasing RWU in these layers before the onset of transpiration reduction.' I do not understand this. If the soil dries out but faster in the upper layers where the root length density is higher than in the deeper layers, the deeper soil layers will not get wetter so $\Gamma$ will not increase in the deeper soil layers, which are still wetter than the upper soil layers. But, $\zeta(z)$ will increase in the deeper soil layers that remained wetter.

P9: Proposed empirical model. Is in this model also the $\alpha(z)$ factor of the Feddes model used?

General question on the used models: The Feddes stress function $\alpha(z)$ is besides a function of the soil water potential, also a function of the potential transpiration rate. How is this considered in the different models? It should be noted that Eq. (20) suggests that $\omega_c$ in the Jarvis model is a function of the transpiration rate but the $\alpha(z)$ used in the Jarvis model is according to Eq. 18 not a function of the transpiration rate. Furthermore, the modified version of the Feddes model shown in Figure 1b suggests that there is no dependence of the $\alpha_m$ function on the transpiration rate and that $\alpha_m$ depends only on the matric flux potential. When looking at table 4, it seems that there is no transpiration rate dependence of the Feddes parameters.

P11 ln 26: 'For high non-linear problems as the one in eq. 29 GLM depends on the initial values of b.' This needs to be reformulated. The GLM does not depend on the initial values of b but the optimized parameter set may depend on the initial value of b since the GLM is a local optimization algorithm that may converge in a local minimum instead of the global minimum.

P 12: '3.2.1 Growing season simulation' This is not a sub section of the optimization section

P13 ln 8: 'hw(= -200 m)'. I am confused here because at p 10 it is written: 'The value of the parameter h4 was set to -150 m.'

P15 ln 30: 'showed by the presence of an outlier and lower medium.' → 'shown' and 'median'.

P17: Growing season simulations. It would be good to have more background about the potential transpiration and the precipitation during the considered growing season.

---

## Editor Comment (EC1) · N. Romano (Editor) · 15 Mar 2016

Dear Authors, Allowing for the interesting comments that your manuscript received so far, and within the spirit of the discussion step of HESS, I would suggest you should start providing preliminary responses. Kind regards, Nunzio Romano
* * *

---

## Referee Comment (RC3) · Anonymous Referee #3 · 25 Mar 2016

The manuscript presents (i) an evaluation of the performance of some well-known conceptual models for root water uptake (RWU) and some modified versions of them and (ii) the determination of their paramters. As reference they use simulations with the more physical model of de Jonge van Lier et al. (2013).

RWU for a simple drying out scenario and a complete season with daily data for precipitation and potential evapotranspiration is simulated with the reference model. The drying out scenario is used to fit the conceptual models to the data of the reference model. With the fitted parameters the complete season is simulated with the conceptual models. RWU patterns and temporal course of transpiration of the conceptual models is then compared with the RWU patterns and temporal course of transpiration of the reference model.

[Figure]

The authors show that most of the conceptual models cannot reproduce the simulations with the reference model but that the modifications greatly improve the performance.

The paper is partly well written and structured. However, to my point of view, especially the results and discussions as well as the conclusions need improvements.

Generally, the topic of evaluating conceptual RWU models by comparison with more physically based models is important, fits well in the scope of HESS and needs consideration. However, as will be outlined below, to my point of view, there are several points, which should be clarified before.

Major comments

The manuscript compares some well-known conceptual models with the reference model, suggests for each of them (except JMII) a modified version and suggests two new models. This is very ambitious. The authors should consider to use less models and go therefore deeper into the discussion, which seem to me rather sketchy.

RWU models must be able to simulate transpiration and local uptake well. However, the conceptual models are only fitted to local uptake data (Eq. 29) and not to temporal course of transpiration. To me this seems to be problematic since (i) for most applications of simple RWU models a sound prediction of transpiration is more important than uptake distribution, (ii) in reality transpiration is much easier to quantify as local uptake so that fitting RWU models to real data (which is the ultimate model test) will probably use transpiration rates for fitting and (iii) even if local uptake at different depth is fitted badly, transpiration can be met well if too low uptake in one depth is compensated by too high uptake at another depth. Therefore, I suggest to fit the models to local uptake and transpiration simultaneously. I know that this will make the weighing scheme more problematic but a solution could be to fit relative transpiration and relative uptake and use weights in a way that transpiration and uptake are equally weighted, e.g. $w_t = n*w_u$, where $w_t$ and $w_u$ are the weights for the transpiration and uptake data and $n$ is the number of depths for which uptake is fitted.

The comparison of the established conceptual models with the proposed modifications seems a bit "unfair" to me since local stress reduction in the modified versions and in the reference model are based on matric flux potential, thus the modifications are closer to the reference model than the other models. Note, that the reference model is still a model and not a representation of reality. This should at least be discussed.

One of the critical points concerning the Feddes stress response function in combination with the Jarivs (1989) compensation approach, the authors mention, is that the models fail to predict compensation under wet conditions, where alpha is 1 for different matric potentials. The modification using martic flux potential with distinct critical point ($M_c$) will perform alike. This is ok but should be discussed.

Model PM mixes stress reduction described by pressure head and compensation calculation based on matric flux potential. I am not sure whether this is a conceptual reasonable model. Please reconsider using this model.

I wonder why model fitting was only performed for the drying out scenario with constant boundary conditions. Under such simple conditions, the information content of the "measurements" might be too low to find parameters for conceptual models, which shall then be used to simulate (extrapolate) dynamics under variable boundary conditions. I would suggest to use the first half of the time with variable boundary conditions for model calibration and the second part for model performance test. This is the usual way for model test in hydrology.

It should be discussed that other physical models do exist, where local RWU is based on energy status instead of matric flux potential, see e.g. Doussan et al. (2006), Javaux et al. (2008) and the simplified model of Couvreur et al. (2012). In this context it can also be discussed that in the de Jonge van Lier model no other parts of the energy density, like gravimetric potential or osmotic potential, can be accounted for. Additionally, although the Feddes stress response function seems to be "out of fashion", it does account for oxygen deficit, which is important at least for the fine textured soils

in the growing season, whereas the matrix flux potential based stress function cannot account for that. These limitations of the reference model should be discussed in the introduction section. Since none of the models (neither the reference model nor the calibrated models) account for oxygen stress, I can imagine that RWU in the clay under variable boundary conditions is not well described by any of the models. This could also be briefly discussed.

To my point of view, a physical model for RWU, which accounts also for the magnitude of potential transpiration (Tp), should be solved with boundary conditions accounting for the daily course of Tp (as done by e.g. Couvreur et al. (2012)).

The title is misleading. I would suggest to use a title, which shows that the paper deals with a performance test of different simple empirical models for RWU using a more complex physical model accounting explicitly for water flux in the soil-plant-atmosphere continuum.

Minor comments

Page 1:

Lines 7 to 8: "The simulated scenarios give more insight into the behaviour of the physical model, especially under wet soil conditions and high potential transpiration rate." This statement seems not to be important for the abstract and can be omitted.

Lines 10 to 11: "...for the scenarios of low RWU "compensation". Better: "...for the scenarios for which RWU "compensation" is expected to be low." or "...for the scenarios for which the physical model predicts low RWU "compensation.""

Lines 13ff: When the Jarvis model is criticized it should be stated that the modifications are conceptually closer to the reference model.

Lines 13 to 14: "Incorporating a newly proposed reduction in the Jarvis model..." Consider: "Incorporating a newly proposed reduction function in the Jarvis model..."

I did not find a statement about the performance of the Jarvis (2010) model in the abstract.

Page 2:

Lines 17 to 18: Models that do not account for compensation are under some circumstances (not all) less accurate, e.g. for coarse to medium textured soils and high root length density.

Line 24: "non-homogeneous" consider "heterogeneous". "For non-homogeneous conditions, RWU for lower R can be the same for higher R depending on the stress level..." Consider: For heterogeneous conditions, RWU for lower R can be the same as for higher R depending on the stress level..." Maybe I am mistaken but I do not see this in Fig. 2: For a certain leaf pressure head (for example -110 m), the RWU for R=0.01 is always lower than for R=0.1 and RWU for R=0.1 is always lower than for R=1.

Line 3: Consider another word than obscure. Compensation will certainly (and shall) enhance uptake (by the factor alpha_2) in some depth compared to the value given by alpha. To me the specific problematic issue is that in case of homogeneous alpha smaller than 1 and omega_c smaller than 1, these models lead to uptake greater than given by the homogeneous value of alpha or, more generally, that relative transpiration can be higher than given by the highest value for alpha in case of heterogeneous alpha distribution with depth (see e.g. Skaggs et al., 2006, Simunek and Hopmans, 2009, Peters, 2016).

Lines 3 to 5: If I understand it right, this holds only for the combination of the Jarvis model with the Feddes stress function for which alpha is 1 for different pressure heads (i.e. between h_2 and h_3).

Line 14: Consider "conceptually" instead of "numerically"

Lines 14-15: I cannot follow: rho and M as defined here do not occur in the Jarvis (1989) model.

Lines 2 to 14: Consider using subsection header such as "3.1 Applied models"

Lines 19 to 20: A free drainage boundary condition is usually used for the case with very deep groundwater level so that groundwater cannot influence the soil. Then the assumption is that at a reasonably deep layer below the root zone the hydraulic gradients are close to unity. This is certainly not the case at the bottom of the root zone. I would suggest to set this boundary condition at a depth of at least 1 m or 1.5 m.

Line 24: "Soil date. . ." should be "Soil data. . ."

Line 26: "These soils are identified in this text as clay, loam and sand (Table 3)." Consider "These soils are identified in this text as clay, loam and sand."

Line 12ff: Please specify in this section at which depths and which time interval the data for S and S* were taken and used to minimize Phi. Consider to fit also transpiration rates and use a weighted least squares scheme instead.

Line 15: ". . .the objective function to be optimized. . ." Consider ". . .the objective function to be minimized. . ."

Line 25: For a nonlinear problem with a model error, i.e. with models that do not fit the data well, there might be several local minima. Did all fitting runs lead to the same minimum? If not I would try to use more starting points to be sure or even a global minimization scheme.

Lines 1 to 2: "This guaranteed that RWU predictions from SWAP corresponded to the best fit of each empirical models to the De Jong van Lier et al. (2013) model." I do not understand this sentence and how it refers to the statement that parameter fitting was only applied for the drying out scenario.

Lines 19 to 20: "Initial pressure heads were obtained by iteratively running SWAP starting with the final pressure heads of the previous simulation until convergence." I do not understand. What converged to which values? And why was the initial condition optimized?

Line 3: "The patterns for the sand and loam soil (not shown here) show very similar features." This is not immediately clear to me since matrix flux potential (M) for the sand is very different from M of clay. In a sand most of the water is available under very low energy densities and thus I would expect that for sand, transpiration is prolonged much longer at potential rates and the drop of $T_a$ to be much steeper after onset of transpiration reduction. Could you discuss this briefly in 2 or 3 sentences?

Line 14: "...increases the reduction of..." consider "... leads to faster reduction of..."

Line 15: "...assumes a parsimonious relationship..." do you mean "... assumes a direct relationship..."

Line 23ff, Tab. 5 and Fig. 6: For Sand with Tp=1mm/d and R=1cm/cm^3 using the JM: omega_c=1, h_3=0 means that transpiration must be reduced from the beginning, since h >0 from the beginning and compensation cannot take place. I cannot see this in Fig. 6, where transpiration is equal to Tp for a prolonged time: Is it due to a very small reduction of alpha_f, so that $T_a$ is smaller than but still close to $T_p$? Please discuss briefly.

The discussion of Line 23ff makes it clear to me that fitting not only the uptake pattern

but also actual transpiration (see major comments) would increase model performance of the conceptual models. Then compensation would be most likely predicted.

Page 15:

Lines 5 to 6: h_s cannot be lower than h_4 if only transpiration but no evaporation is considered.

Lines 16 to 20 and general: "performs better", "overestimates RWU", . . . Please discuss the performance of the conceptual models always with respect to the VLM since you compare models. A comparison with real data is still the best benchmark.

Lines 21ff: Here fitted models are compared by statistical measures like E and rˆ2. Since the fitted models use different numbers of adjustable parameters such a comparison is not justified: More free parameters mean more flexibility and thus a better "chance" to fit the data. Please consider using other measures, which account for number of fitted parameters, like AIC (Aikaike, 1974).

Line 25: ". . .models (except for JM and JMm by setting omega_c > 1) are. . ." This can be omitted since omega_c > 1 makes conceptually no sense.

Lines 16 to 17: "The optimal h3 and Mc values (Table 5) for FM and FMm, respectively, increase as R or Tp increases, contradicting their conceptual relation to R and Tp levels" I see the contradiction only with respect to increased R but not to increased Tp.

Lines 31ff: I assume that parameters h_3 and omega_c for JM are highly correlated. Can you give information about parameter correlation? Moreover, such parameter correlation might be due to model structure but also due to data used for fitting the model. Therefore, I repeat my suggestion to use not only the drying out scenario for model calibration but the scenario with changing boundary conditions. This might reduce correlations.

Lines 1 to 2: What are l-values? L_m and lambda respectively. Please unify.

Line 4ff: A figure with the cumulative transpiration over time would be interesting to see if there are under-/over-estimations for specific time intervals in the complete season.

Line 23: The statement that JMII is poor in performance should be discussed with more caution since it was not adjusted to the reference model. Thus, this finding can be expected. The same holds to a less extend to the models for which only one parameter was adjusted.

Line 24: This is a very daring conclusion, since the reference model and the proposed models have partly a similar structure (see above).

Conclusions section: I could not find a single conclusion. This is rather a summary and not a conclusion.

Line 32: "...especially under wet soil conditions and high potential transpiration." Why do the simulations yield insight especially under wet soil conditions?

Lines 21 to 22: This paper is certainly not in press

Tables and Figures Table 3: Although the Mualem/van Genuchten model is well known the equations should be stated in the text to make it easier to assign the parameters. What, for example, is lambda? I guess the so-called tortuosity parameter in Mualem's model, but i am not sure. Alternatively, Tab. 3 can be completely omited and the functional relationships of theta(h) and K(h) might be plotted in an extra figure.

Table 4: I cannot find l_m for PM and PMm in the text. Do you mean lambda instead of l_m?

Table 5: In the text root length density is R here it is Rd.

Table 6: For comparison: what was the value for potential transpiration

Fig. 1: a) since h_1 and h_2 are set to zero in all simulations, Fig. 1,a should account for that and start with alpha=1 at h=0. b) since M_c for Tp=1 mm/d is different from M_c for Tp = 5 mm/d, this should be indicated in Fig. 1b using M_c,l and M_c,h , similarily to h_3,l and h_3,h in Fig 1,a.

Fig. 3: Should only contain the three root distributions used in this study.

Literature Akaike, H., 1974. A new look at statistical model identification, IEEE Trans. Autom. Control, AC-19, 716–723.

Couvreur, V., Vanderborght, J., Javaux, M., 2012. A simple three-dimensional macroscopic root water uptake model based on the hydraulic architecture approach. Hydrol. Earth Syst. Sci. 16 (8), 2957–2971.

Doussan, C., Pierret, A., Garrigues, E., Pagès, L., 2006. Water uptake by plant roots: II–modelling of water transfer in the soil root-system with explicit account of flow within the root system–comparison with experiments. Plant Soil 283 (1–2), 99–117.

Jarvis, N., 2010. Comment on macroscopic root water uptake distribution using a matric flux potential approach. Vadose Zone J. 9 (2), 499–502.

Jarvis, N.J., 1989. A simple empirical model of root water uptake. J. Hydrol. 107 (1), 57–72.

Javaux, M., Schröder, T., Vanderborght, J., Vereecken, H., 2008. Use of a threedimensional detailed modeling approach for predicting root water uptake. Vadose Zone J. 7 (3), 1079–1088.

Peters, A., 2016. Modified conceptual model for compensated root water uptake–a simulation study, J. Hydrol., 534, 1–10.

Simunek, J., Hopmans, J.W., 2009. Modeling compensated root water and nutrient uptake. Ecol. Modell. 220 (4), 505–521.

Skaggs, T.H., van Genuchten, M.T., Shouse, P.J., Poss, J.A., 2006. Macroscopic approaches to root water uptake as a function of water and salinity stress. Agric. Water Manag. 86 (1), 140–149.

---

## Author Comment (AC1) · 29 Jun 2016

In response to the comments by N. Jarvis:

We are thankful for you critical reading, constructive comments and suggestions that will help to enhance the paper. In the following we address the main questions and the numbered specific questions are addressed thereafter.

Regarding the conclusion that proposed models are recommend, we can make a more thorough analysis, as you suggested by applying the Akaiki information criteria to support the conclusions. Regarding the values of $\lambda$ (that will be called $l_m$) of the proposed models, they can be greater than 1, as discussed in more detail in points 9 and 16. In applying the evaluated models in blind predictions the JMII may have more advantages over the other models as it is more physically based. This will be shortly addressed

in the final version. One of the reason why the models PMm and JMm perform better than JMll is detailed in the reply of point 16. Another reason, as raised by you, might be the fact that the threshold type function can mimic a constant plant resistance that dominates in the early stages of soil drying. Nevertheless, as commented by the Referee#3, the fact that the proposed and the De Jong van Lier et al. [2013] reduction function are both a function of matric flux potential might be favourable to the proposed models. Yes, the uncertainty of soil hydraulic parameters might affect the outcome of the comparisons and as you agreed it is ok for our modeling exercise. We will discuss this issue according to your suggestion.

We will take your suggested titles into consideration.

Next we respond to specific questions:

1. Page 2, lines 24-25 (and line 3 in the abstract): I am not so convinced of this. I would prefer to use a physics-based model even if it did have two or three more parameters, as long as they were, in principle, measurable. The limiting leaf water potential is quite well known, at least

R.: We also would prefer using a physics-based model, but in practice it appears not to be appealing. Root water uptake (RWU) models are usually embedded in larger hydrological models, for instance the ecohydrological model SWAP ( De Jong van Lier et al., 2008), and most users are unfamiliar with plant hydraulic parameters, making them to prefer the simplicity of empirical models like the Feddes et al. [1978] model, as long as empirical parameters are available. Besides, apart from the well-known limiting leaf water potential, radial root hydraulic conductivity has a strong effect on RWU distribution as shown in the paper and it is not easily available.

2. Page 4, line 5–8: the sentence starting ... "using $h$ seems ...." is wrong, as the authors know well enough. It is immediately contradicted by the text at lines 14-21 on the same page (and by the results shown later in the paper). This sentence should be deleted. The remaining text just says that various forms have been proposed for the $\alpha$

function, but that making $\alpha$ depend on M is physically the most plausible. This is quite sufficient

R.: The sentence is in fact misleading. It was intended to say "Comparing to $\theta$, $h$ seems to be more feasible..." instead of "Using $h$ seems". It will be corrected as such and then it will not be contradictory anymore. In the text below, it only states that the use of $M$ is more plausible than both $h$ and $\theta$. We hope this to become clear with this slight modification.

3. Page 5, equation 8: $h_0(z)$ is not defined, as far as I can see?

R.: Correct, it will be defined.

4. Page 5, lines 20-22: yes, it would be good if you mentioned this phenomenon by its name: hydraulic lift or hydraulic re-distribution. You could also cite Jarvis (2011) here, since he discussed and clarified the relationship between water uptake compensation and hydraulic lift in some detail (see the text in relation to equations 13 to 15 in the final version of this paper, not the HESS discussion paper that you cited: see point 4. under "Presentation")

Agree, it is important to mention the name of the phenomena as well as cite Jarvis (2011). These changes will incorporated in the text.

5. Page 6, lines 1-3: I know what you are trying to say here, but it is not so well expressed. You could replace i.) " .... is only relevant" by ".... it only needs to be explicitly addressed ...." and ii.) "... becomes less important" by "... is not necessary". This would help, but you could also add a sentence at the end of this saying that the effects of compensation can nevertheless be explicitly discriminated and identified in physics-based models. This is demonstrated in Jarvis (2011) in the text related to equations 13 and 14 in that paper (again, in the final version).

The text will be improved accordingly. However, we don't think it is always possible to explicitly discriminate and identify the effects of compensation in physically-based models. Such relation (Jarvis [2011] eq. 13 and 14) was easily found for the De Jong van Lier et al.[2008] model comparison (Jarvis, 2011). Furthermore, adding this comment would be contrary to our general reasoning when we add that "In physical models, discriminating compensation is not necessary since in such models "compensation" follows implicitly from the RWU mechanism".

6. Page 6, lines 11-12: "In principle, any definition of $\alpha$ is applicable...". Yes, perhaps, but it does make a difference to the results of course, as you demonstrate very well later in the paper! But what is definitely not debatable is that Jarvis (1989) used a threshold type function for $\alpha$ based on water content (degree of saturation). The reason for adopting this approach was discussed by Jarvis (1989) in relation to the experimental evidence available at that time and no other type of function was considered. The fact that you adopt a Feddes-type function means that in the rest of the paper you cannot refer to this model as the Jarvis (1989) model. It is a modified Jarvis (1989) model, in exactly the same way that JMm is also a modified Jarvis (1989) model, where the threshold water content function is replaced by a threshold function of matric flux potential: in other words, you investigated two different modified Jarvis models and you should refer to them as such, both in table 1 and throughout the rest of the paper, including the abstract (perhaps you could call them JMm1 and JMm2?)

R. Indeed, any kind of $\alpha$ might provide different predictions. We agree that using the Feddes reduction function in the Jarvis (1989) model is also a modification of the Jarvis (1989) model, and will refer to it in the paper as such (it will be call JMf).

7. Page 6, line 28 to page 7, line 4: this is a little vague. You followed quite closely what Skaggs et al. (2006) wrote in this section, but since they wrote their paper ten years ago, it is now much better established exactly how the original Jarvis (1989) model departs from physicality. This was clarified in the papers by Jarvis (2010, 2011), which you also discuss in the following section. There are two aspects to this:

i.) the choice of function for $\alpha$. The threshold function chosen by Jarvis (1989) doesn't make complete physical sense, as the local resistance to uptake should in principle increase continuously as the soil dries (e.g. like equation 18). Jarvis (1989) discussed this choice in terms of the overall resistance to uptake being dominated by an air gap between soil and root which might only develop after a certain critical water deficit was reached: this choice was strongly influenced by experimental studies which showed such an effect. Also, at high soil water contents, the overall resistance to uptake in the soil-plant system would be dominated by plant resistances, which may be more or less constant. Thus, a threshold function might be a good choice from an empirical point of view. In this respect, it can also be pointed out here that the authors also adopt a threshold $\alpha$ function in the PMm model. This model is the one the authors finally recommend, because it works best, although it can certainly be criticized on the same grounds (i.e. that it "affronts the definition of $\alpha$").

ii.) Compensation under non-stressed conditions. As you point out, under non-stressed conditions the Jarvis (1989) model does give a different uptake distribution compared with the de Jong van Lier physical model. However, it is wrong to imply that the Jarvis (1989) model does not predict any compensation under non-stressed conditions (page 7, line 4). Under non-stressed conditions, water uptake is increased by a factor of $1/\omega$ in all layers (regardless of the pressure head distribution) to maintain transpiration at the rate demanded by the atmosphere during soil drying. It is also not wrong in principle to link compensation to plant stress (page 7, line 3): the onset of stress certainly does affect the nature of compensation: this is demonstrated in Jarvis (2011) in the text following equations 13 and 14 for the physics-based model of de Jong van Lier (2008).

For the above reasons, I strongly suggest that you delete the text on page 6 line 28 to page 7, line 4 and replace it by a short sentence that simply states that the Jarvis (1989) model departs from complete physicality in some respects and that this is explained in the following section. Then at the end of the next section (i.e. after equation 21) you can briefly summarize how the Jarvis (1989) model departs from physicality, based on

the comparison with the physics-based model that is represented by equation 14-21. This will be very much clearer.

i. The proposed models are also based on a threshold-hold type function for $\alpha$. However, the fact that Jarvis [1989] model affronts $\alpha$ definition is related to how the model functions, and not the $\alpha$ definition itself. That is, the model has a reduction function accounting for RWU reduction due to soil water resistance. For a given soil layer $i$, it follows that $S_i < S_{p_i}$ if $\alpha_i < 1$. It means RWU is reduced by soil water resistance, accounted for by $\alpha$. However, a situation occur when $\alpha_{2_i} > 1$, causing $S_i > S_{p_i}$ even if $\alpha_i < 1$. In other words, there should be a reduction due to $\alpha$ ($\alpha < 1$), but in fact an increase in RWU might occur. We therefore state that the Jarvis [2011] model is in conflict with $\alpha$ definition, as first noticed by Skaggs et al. (2006). On this regard, the proposed models function differently: RWU is first partitioned by weighting factor between $M$ and $R^\lambda$, then it is reduced by $\alpha$. Anyhow, we agree that just citing Skaggs et al. (2006) should be enough in this part.
ii. We agree that this needs to be corrected. The Jarvis 1989 model predicts compensation under non-stressed conditions. Regarding the linking of compensation to plant stress, it depends on how compensation is defined or understood. Following the Javaux et al. 2013 definition, compensation is an independent process of plant water stress, driven only by soil hydraulic re-distribution. However, in this paper as discussed in section 2.1 we stress that the term "compensation" is not relevant in physical-based models, since it follows implicitly from RWU mechanism. Furthermore, we stressed that compensation term was used to interpret the difference in results predicted by empirical models. Interpreting such results from a physical model, there is no need in referring to "compensation", as it is an implicit RWU mechanism.
Nevertheless, we agree that text will be become clearer by deleting the mentioned part.

8. Page 7, lines 5-12. The parameter $h_3$ does not exist in the Jarvis (1989) model (see lines 10-11 especially). I think this paragraph can be deleted (or perhaps moved to the results and discussion section). At the very least, readers should be reminded that the

original Jarvis (1989) model does not use a Feddes-type $\alpha$ function.

OK, this will be moved to a proper location in the text

9. Page 8, line 22: you should add the limits for $\lambda$ here. If compensation means that water uptake increases from sparsely rooted layers, then $\lambda$ must lie between zero and 1. Also, you should replace "deeper soil layers" by "more sparsely rooted layers" to be strictly correct.

We agree that an explanation is missing about the limits for $\lambda$ as well as how $\lambda$ values affect RWU and compensation. In fact, $\lambda$ is not restricted to the domain between 0 and 1. The $\lambda$ values were originally based on experimental works, but essentially it changes the shape of RWU over depth by giving more weight to $R$ or $\alpha$ in partitioning RWU. For $\lambda = 1$, RWU is partitioned by a simple weighting factor between $\alpha$ and $R$. For $\lambda > 1$ ($\lambda < 1$), $R$ ($\alpha$) becomes more important on partitioning RWU.

10. Page 9, lines 23 to 26: I wonder what it is about your modification to the Li model (the use of the matric flux potential in a threshold function) that resolves the conceptual difficulties with the original formulation that you described earlier on page 9 at lines 3 to 8. As far as I can see, the same objections should be equally valid for this modified version as for the original model. This should be clarified and the text modified accordingly.

The main objection regarding the Li et al. [2001] model is the use of $\alpha$ in $\zeta$ (eq. 22). Thereby, "compensation" taking place before transpiration reduction (when $\alpha = 1$ for all soil layers) can not be computed: RWU is distributed over depth only by $R^\lambda$. By using $M$ instead of $\alpha$, "compensation" before transpiration reduction can be computed. As $M$ integrates both the effects of $K$ and $h$, it might be a better soil hydraulic function than $K$ or $D$ [Molz and Remson, 1970; Selim and Iskandar, 1978] to account for the effects of soil water in partitioning RWU. Such comments will be added into section 2.2.4.

11. Page 11, lines 13-28: As I understand it from table 3, you only have a maximum

of two parameters to calibrate for all the models, while each parameter is constrained within known limits. This means that a "brute force" grid search for optimum parameter values would be preferable to the method you chose, since you could be sure of avoiding risks of finding local minima (although it might be slower). I am sure there is no need to repeat the calibrations, but maybe you could mention this?

R.: The "brute force" grid search is a very slow method. As there are many scenarios and some models to evaluate, we dot not think it is interesting to mention it since it would not be applicable in practice (a very small grid would also be required to avoid finding relative minimum).

12. Page 13, lines 23-24: yes, this may be why a constant value of $\omega_c$ often seems to work quite well. Maybe you could add a comment to this effect, and also refer to your equation 20 and cite Jarvis (2011), where this aspect is discussed in detail.

R.: As eq. 20 gives an expression for $\omega_c$ derived from the De Jong van Lier et al. [2008] physics-based model [Jarvis, 2011], it indeed helps in accounting for some aspects relating RWU phenomena. A constant $\omega_c$ might be quite robust as can be inferred by eq. 20 and from common field observations. However, adding such a comment in this part (page 13, lines 23-24) might get out of the context of the paragraph.

13. Page 15, line 18: You should replace "either R or M" by "both R and M". But this sensitivity to $M$ is in principle also present in the empirical models that include $M$. Why is it more important for JMII? Is it because this model is not calibrated? Or is it because of the different type of function? I can believe that predictions of JMII are, in comparison with the empirical models, more affected by the value of $M_{max}$, which must be a very uncertain parameter, not least because the Mualem-van Genuchten model of soil hydraulic properties is known to have an incorrect form close to saturation (since it does not allow for a maximum size of pore in soil). These questions should be clarified.

R.: Comparing the models JMf (earlier abbreviated as JM ) and JMII it is clear that Jarvis 1989 model type is affected by the defnittion of $\alpha$. This becomes more evident

analyzing Fig. 1 (see its caption below and figure at the end) which shows $\alpha$ of JMII (eq. 18) as a function of soil pressure head $h$ and $\omega_c$ (eq. 20) for different soil types, expressed by $M_{max}$. Focussing first on the $\alpha$ function, it can be seen that despite the fact the soil resistance should increase continuously as soil dries as you point out in point 7, defining $\alpha$ by eq. 18 does not seem very realistic. In this case $\alpha$ is suddenly reduced even close to saturation. When $h = 1$ m, for instance, $\alpha$ is much lower than 0.5. Such a behaviour does not correspond to the $\alpha$ definition. Another interesting point is the values of $\omega_c$ which are also extremely low. The low $\alpha$ values are, however, balanced by high $\alpha_2$ values (due to low $\omega$ and $\omega_c$ values), leading to suitable values of RWU in a given soil layer. Nevertheless, the magnitude of $\alpha$ and $\omega_c$ are physically questionable. SOme interestinh points that can be drawn from the above: i) the $\omega_c$ value which sets the compensation level depends on the $\alpha$ definition. For instance, Jarvis 1889 stated that $\omega_c = 0.5$ is a moderate level of compensation. Surely, it does not hold if $\alpha$ is defined by eq. 18. ii) Comparing Jarvis 1989 to De Jong van Lier et al. [2008] model led to an unrealistic $\alpha$ function, and its behaviour does not properly represent the $\alpha$ concept. The threshold type functions like the other ones evaluated in this paper seems to be more feasible.

The fact that JMII is more sensitive to both $R$ and $M$ when compared to the other $M-$based models is eventually attributed to the $\alpha$ function and related equations derived to express their parameters (eq. 19 and 20). It can be seen from Fig. 1(c) that $\beta$ defined by eq. 18 ($\beta$ of JMII) tends to be higher when $R$ increases and tends to be lower when $R$ decreases compared to $\beta$ of JMf and JMm. Thereby, for the first days of simulations when the soil hydraulic conditions tend to be rather uniform over depth, JMII overestimates RWU compared to VLM predictions. This becomes more important for the high $R$–low $T_p$ scenarios. For such conditions, the RWU over depth predicted by the VLM tends to be more uniform, which is reasonable since the low transpiration demand can be met by any small root density that can be found in deep soil depths. After some period of time, the discrepancies between VLM and JMII tend to increase, since the higher uptake in the upper layers reduces $h$ and because of $\alpha$ shape of JMII

RWU in the upper layers are suddenly reduced towards zero. These are the main reasons why JMII does not predict well in high $R$–low $T_p$ scenarios. (Part of) this discussion may be included in a future version.

Fig 1 Caption: (a) $\alpha$ of JMII model (eq. 18) as function of soil pressure head $h$, (b) $\omega_c$ parameter (eq. 20) for different soil types, expressed by $M_{max}$ and (c) the normalized root length density $\beta$ computed by the eqs. 4 (JMf) and 19 (JMII) as function of root length density $R$ considering $R$ over depth given by eq 28 with $R_{\mathrm{avg}}$ and $b$ equal to 1.0 cm cm$^{-3}$ and 2, respectively.

14. Page 15, line 25: it could also be noted (perhaps by referring to equation 20) that $\omega_c > 1$ is not physically unrealistic.

R.: Yes, $\omega_c > 1$ is not physically unrealistic and it is implicitly stated in line 25, from which can be inferred that if we set $\omega_c > 1$, both JMII and JMm can predicted $T_a/T_p < 1$ for the low $R$-high $T_p$ scenarios as VLM did.

15. Page 16, line 3: This is misleading. The Feddes function for $\alpha$ is not part of the Jarvis (1989) model.

R.: As it was commented in point 6, it will be referred to as JMf.

16. Page 17, lines 1-2: it is confusing that different symbols are apparently used for one of the parameters in the Li-type models. In equation 25, $\lambda$ is used, whereas in the text here and in table 5, $l$ is used, while in table 4 $l_m$ is used. I believe they are all the same parameter?

If I understood it correctly, I don't see how you can write that the optimal values of $\lambda$ follow a logical relation to $R$ and $T_p$ (line 1). In many cases, and especially for low root densities, values of $l$ (i.e. $\lambda$?) in table 5 are larger than 1, which implies to me that compensation is working incorrectly in these scenarios (it is decreasing uptake in the more sparsely rooted layers). Also, in table 4, it is stated that $l_m$ (i.e. $\lambda$) was constrained to take values less than or equal to 1. If I understood it correctly, the results in table 5

suggest that this was not actually the case in practice.

R.: All symbols refer to the same parameter and we will correct this by changing them to $l_m$. Regarding the $l_m$ parameter limit constraining values, the upper values for $l_m$ were constrained to 3. Conceptually, there is no inconsistency in taking $l_m > 1$. Indeed, $l_m = 1$ means no compensation at all and $l_m < 1$ implies compensation. Values of $l_m > 1$ simply indicates that the upper soil layers are more important for RWU distribution. The limits values for $l_m$ will be corrected as well as such a comment will be added explaining these features at a suitable location in the manuscript.

17. Section 4.2, table 6: can you give the total precipitation and potential transpiration here? It's good to get a rough idea of how much stress occurred in these simulations.

R.: We will try to insert this information into the table or making a plot of cummulative transpiration over time

18. Page 18, lines 10-12: you did not test the Jarvis (1989) model (see earlier comments).

R.: Yes, it will be corrected

19. Page 18, line 12: I did not get a good understanding of why the JMII model does not work so well for high R–low Tp scenarios (i.e. high compensation). I would have thought that, in principle, it should work OK. Please briefly explain what you think the reasons are for this.

R.: This is explained in point 13 and will be added at the end of section 4.1.2.

20. Lines 16-18: I think this is too optimistic, as this test was not a very tough one. You had the same plants (identical roots) and the same three soils. How would it look if you had simulated different scenarios (soils, plants)? I think you would need to re-calibrate the empirical models. How useful is that?

R.: Yes, the results would be different. However, it is useful to show that the methodology used to calibrate the models is robust and can be used to assess empirical models and sensitivity of the empirical parameters in order to provide a full calibration of the empirical models in a next step.

References

Q De Jong van Lier, J C Van Dam, K. Metselaar, R. De Jong, and W H M Duijnisveld. Macroscopic root water uptake distribution using a matric flux potential approach. Vadose Zone Journal, 7(3):1065–1078, 2008.

Q De Jong van Lier, J C van Dam, A Durigon, M. A. Santos, and K Metselaar. Modeling water potentials and flows in the soil-plant system comparing hydraulic resistances and transpiration reduction functions. Vadose Zone Journal, 12(3), 2013.

R.A. Feddes, PJ Kowalik, and H. Zaradny. Simulation of field water use and crop yield. Simulation Monograph Series. Pudoc, Wageningen, The Netherlands., 1978. N J Jarvis. A simple empirical model of root water uptake. Journal of Hydrology, 107(1): 57–72, 1989.

NJ Jarvis. Simple physics-based models of compensatory plant water uptake: concepts and eco-hydrological consequences. Hydrology and Earth System Sciences, 15(11):3431– 3446, 2011.

Mathieu Javaux, Valentin Couvreur, Jan Vanderborght, and Harry Vereecken. Root water uptake: From three-dimensional biophysical processes to macroscopic modeling approaches. Vadose Zone Journal, 12(4), 2013.

K Y Li, R. De, Jong, and J B Boisvert. An exponential root-water-uptake model with water stress compensation. Journal of Hydrology, 252(1):189–204, 2001.

FJ Molz and Irwin Remson. Extraction term models of soil moisture use by transpiring plants. Water Resources Research, 6(5):1346–1356, 1970.

HM Selim and IK Iskandar. Nitrogen behavior in land treatment of wastewater: A

simplified model. State of Knowledge in Land Treatment of Wastewater, 1:171–179, 1978.

T. H. Skaggs, M. T. Van Genuchten, P. J. Shouse, and J. A. Poss. Macroscopic approaches to root water uptake as a function of water and salinity stress. agricultural water management, 86(1):140–149, 2006.

J. C. Van Dam, P. Groenendijk, R. F .A. Hendriks, and J. G. Kroes. Advances of modeling water flow in variably saturated soils with swap. Vadose Zone Journal, 7(2):640–653, 2008.
* * *
[Figure]

[Figure]

**Fig. 1.** Caption in the text

---

## Author Comment (AC2) · 29 Jun 2016

**Reply to "Interactive comment on "Determination of empirical parameters for root water uptake models" " by Referee#2**

In response to the Anonymous Referee #2 :

We are thankful for your critical reading, constructive questions and suggestions that will help to improve the paper. In the following we address the general questions and the numbered specific questions are addressed thereafter.

i) Regarding the dependence of the model parameters on transpiration rate, indeed this topic is explicitly addressed in the results and discussion. We can also shortly address

this topic when introducing and discussing about the Feddes and proposed reduction functions at the end of the section 2.

ii) We introduced some general advantages concerning the use of empirical models as compared to the De Jong van Lier et al. [2013] physical model in page 2, lines 23 to 27.

iii) As suggested we can enhance the conclusion section in order to include the aspects regarding the dependence of empirical model parameters on soil properties and hydraulic conditions. We will address the variation of the other parameters as suggested.

iv) It not possible to directly retrieve root water uptake from measurements. This is one of the advantages of using physical-based models. The main purpose of this paper is to evaluate empirical models that can be sensitive to and follow the variations of root water uptake due to different scenarios of soil and plant properties as well as climatic conditions as predicted by a physical model. By using root water uptake it is possible to strictly capture the root water patterns predicted by the models, whereas for instance if using soil water content the results can be "blinded" by the sensitivity of RWU on soil water content which vary with soil type. Using transpiration may lead to wrong predictions on root water uptake. Some of these aspects can be addressed more specifically and a short examination on using transpiration can be performed in order to show this.

Next we respond to specific questions.

1. P4: root length density $R$. Shouldn't that have dimension L L$^{-3}$ ?

Yes, it will be corrected.

2. P4: The authors propose a stress function $\alpha$ which is a stepwise linear function of $M$. Since $M$ is a function of $h$, the new stress function will be a function of $h$ also. But the shape of the function will have a different shape than a piecewise linear function

of $h$. Furthermore, the relation between the new stress function $\alpha$ and $h$ will depend on the hydraulic soil properties and will therefore be different in soils with a different texture. The original Feddes $\alpha(h)$ function depends on the transpiration rate as shown in Figure 1. Figure 1 suggests that the new stress function $\alpha(M)$ does not depend on the transpiration rate. I do not understand why the transpiration dependency of the stress function disappears when $\alpha$ is expressed as a function of $M$ since $M$ does not depend on the transpiration rate.

This is a very important observation. In fact the new $\alpha$ function will also depend on potential transpiration rate $T_p$. This dependency is implicitly expressed in the critical value $M_c$ of $M$. Therefore, as in the case of the Feddes $\alpha$ function there should be two values for $M_c$: one for low $T_p$ and other for high $T_p$. Fig. 1 will be correct to include this.

3. P5: ln 15: "Because $T_a$ and $h_l$ are unknowns, eq. 8 and 10 cannot be solved analytically, but an efficient numerical algorithm is described in De Jong van Lier et al. (2013)." I did not understand this. I thought that either $T_a = T_p$ is known as a boundary condition so that $h_l$ can be calculated or $h_l = h_w$ is known and $T_a$ is calculated. I think that the reason why the $h_l$ (or $T_a$) cannot be derived directly is because the set of equations that needs to be solved (including also all $h_{0,i}$'s ) is non-linear in $h_{0,i}$.

Thank you for other very important observation. As you noticed well, the sentence is wrong. Indeed the set of equations can be solved analytically (and we in fact used an analytical solution by De Jong van Lier et al. [2013]) for some special cases of Brooks and Corey [1964] soils, but not in direct way. This will be corrected.

4. P5 ln 17 and p 29 Figure 2: There are several things I do not understand about Figure 2. The figure caption says that the plant transpiration was set to 1 mm d$^{-1}$ . Shouldn't for a fixed rooting depth the root water uptake or sink term $S$ be constant and independent of the root length density $R$ until a threshold soil water potential is reached? This threshold will depend of course on the leaf water potential and the root length density. Can it be that the curves shown in Figure 2 shown the maximal possible

sink term as a function of the soil water potential for different leaf water potentials and root length densities? But, when the root water uptake goes to zero, why doesn't the soil water potential then go to the leaf water potential? Now there seems to be a 10 m difference between them. Second, why doesn't the root water uptake for a certain soil water potential then not increase with decreasing leaf water potential. For sufficiently large (small absolute value) soil water potential, the root water uptake becomes independent of the leaf water potential. I do not understand this since the water potential difference increases with decreasing leaf water potential and therefore the root water uptake should also increase with decreasing leaf water potential.

i) For a fixed rooting depth, the root water uptake (RWU) is physically given by the soil water flux at the root surface $q = -K\partial h/\partial r|_{r=r_0}$ integrated over the root surface area divided by the soil volume exploited by the root. In the De Jong van Lier et al. [2013]) model this is represented by eq .6. Therefore, root water uptake depends on root length density $R$, as root surface area increases with $R$. Leaf water potential will also affect $S$ as shown in Fig 2 since it will affect the soil pressure head at the soil root interface.

Then, your question is very pertinent since in the empirical RWU models we assume a reduction curve of the type you mentioned in your question (a threshold-type reduction function), whereas the physical RWU model shows this curve is different: there is a continuous reduction ever since the soil starts to dry. This reduction curve is even more complex since the water uptake in one layer is dependent also on the whole rizosphere uptake, i.e. the uptake in one layer is influenced by the uptake in other layer which is overall controlled by the $h_l$ value. In that case it might happen that $h_s$ in one given layer decreases while total uptake increases. There is no $\alpha$ function that can account for this, but we hope that introducing a compensation factor in such approach this overall RWU distribution can be closely mimicked. Thus, physically the reduction function given by eq. 18 is more suitable (see the discussion in the reply of RC1, topic 13). These aspects and more thorough analysis on this will be included in the revised version.

ii) The RWU is zero when the pressure at the soil-root interface is equal to bulk soil pressure head $h_s$. In that case, the $h_l$ must be lower than $h_s$ because of the water head loss from the root to the leaf.

iii) Fig. 2 does show RWU increases with decreasing water potential for a certain $h_s$ value. However, for less negative $h_s$, RWU becomes less sensitive to high negative $h_l$ values.

5. P7 ln 21: "where $T_{p_{\max}}$ is the maximum possible transpiration rate attained when $M_0 = 0$". This assumes that the minimal water potential at the soil-root interface is $h_w$ (wilting point). But, doesn't this minimal water potential depend also on the critical leaf water potential $h_l$?

Yes, it depends on $h_l$. The limiting pressure head at the soil-root interface (called $h_{ws}$ to avoid confusion) is less negative than the limiting $h_l$ (called $h_{wl}$). Although $h_{ws}$ depends on $h_l$ and on plant and soil hydraulic parameters, for the sake of simplicity we considered it as constant and equal to -150 m. The $h_{ws}$ value was the limiting value used in the empirical models that depends on $M$ and it will be listed in Table 2.

6. P7 Eq. 17: Why is $M_0$ constant with $z$? The soil root interface water potential can depend on the depth, can't it?

We take advantage of your question and correct eq 6 to explicitly make $M_0 = M_0(z)$. However, De Jong van Lier et al. [2008] did assume $M_0$ constant with depth in order to solve the problem of the two unknowns: $T_a$ and $M_0$. They made a justification for that, and we refer to their paper (?) for more detail. With this assumption it was possible later on to Jarvis (2011) make a comparison with the Jarvis (1989) model.

7. P8 ln 15: "The Jarvis (1989) model predicts RWU by a weighting factor between $\rho$ and $M$ throughout rooting depth". This is not very clear to me. What do you mean with a weighting factor "between $\rho$ and $M$"? Do you mean a weighting factor that is equal to the product of $\rho$ and $M$?

Interactive
comment

An interesting feature of the analogy between the Jarvis model and the De Jong van Lier et al. [2008] model is that the analogy is derived based on the assumption that stress only occurs when everywhere at the soil-root interface limiting conditions are reached. It is assumed that $M_0$ is zero everywhere in the root zone. But, I am wondering whether the De Jong van Lier et al. [2008] only predicts stress under these conditions. Can it be that stress occurs even though $M_0(z)$ is not zero everywhere in the root zone? If this is the case, then the analogy between the Jarvis and the De Jong van Lier et al. [2008] models is not given always when stress occurs.

A weighting factor was meant as equal to the product between $\rho$ and $M$ divided by the integral of this product over the rooting zone. As $M_0$ in the De Jong van Lier et al. [2008] model is constant over depth, stress occurs when $M_0$ over the root zone.

8. P 8 ln 22: "The smaller $\lambda$, the more water is taken up in deeper soil layers" I would reword this to "... the more water is taken up from layers with a low root length density".

We will change the sentence accordingly.

9. P 9 ln 1: "RWU is calculated by substituting eq. 23 into eq. 3, following the Feddes approach." This implies that you multiply Eq. 22 again by $a(z)$. So in the nominator, you get $\alpha^2$?

Yes, that will be the case.

10. P9 ln 16: Same comment as above.

No, in this case it will not happen.

11. P 9 ln 18: "In drier soil layers, $\Gamma$ is reduced, whereas in wetter soil layers $\Gamma$ is increased, thus increasing RWU in these layers before the onset of transpiration reduction." I do not understand this. If the soil dries out but faster in the upper layers where the root length density is higher than in the deeper layers, the deeper soil layers will not get wetter so $\Gamma$ will not increase in the deeper soil layers, which are still wetter than the upper soil layers. But, $\zeta(z)$ will increase in the deeper soil layers that remained

wetter.

Indeed this sentence needs to be rephrased. As you put out well, in fact $\Gamma$ in wetter soil layers will not increase, but $\zeta$ will do because $\Gamma$ in these layers will be less reduced compared to $\Gamma$ in the upper dryer layers.

12. P9: Proposed empirical model. Is in this model also the $\alpha(z)$ factor of the Feddes model used?

General question on the used models: The Feddes stress function $\alpha(z)$ is besides a function of the soil water potential, also a function of the potential transpiration rate. How is this considered in the different models? It should be noted that Eq. (20) suggests that $\omega_c$ in the Jarvis model is a function of the transpiration rate but the $\alpha(z)$ used in the Jarvis model is according to Eq. 18 not a function of the transpiration rate. Furthermore, the modified version of the Feddes model shown in Figure 1b suggests that there is no dependence of the $\alpha_m$ function on the transpiration rate and that $\alpha_m$ depends only on the matric flux potential. When looking at table 4, it seems that there is no transpiration rate dependence of the Feddes parameters.

The proposed root water uptake models are obtained by incorporating $\zeta_m$ into eq. 23, then into eq. 3. The PM uses Feddes reduction function whereas PMm uses the proposed reduction function $\alpha_m$ as shown in Table 1. We will add more information in this part in order to it get more clear.
The dependence of the models on potential transpiration are implicitly built-in in the values of their empirical parameters that were optimized. For instance, in the Feddes reduction function there are two values for $h_3$: one for low $T_p$ ($h_{3l}$) and another for high $T_p$ ($h_{3h}$). The dependence of $T_p$ in the other models are accounted for similarly. We then optimized the models for two levels of $T_p$ (1 and 5 mm d$^{-1}$, therefore the optimized parameters are derived for low and high $T_p$.

13. P11 ln 26: "For high non-linear problems as the one in eq. 29 GLM depends on the initial values of b." This needs to be reformulated. The GLM does not depend on the

initial values of b but the optimized parameter set may depend on the initial value of b since the GLM is a local optimization algorithm that may converge in a local minimum instead of the global minimum.

We agree with your observation. We will reformulate this sentence accordingly.

14. P 12: "3.2.1 Growing season simulation". This is not a sub section of the optimization section.

Yes, it will be changed.

15. P13 ln 8: "$h_w(= -200$ m)". I am confused here because at p 10 it is written: "The value of the parameter $h_4$ was set to -150 m.".

We discussed this above in point 5.

16. P15 ln 30: "showed by the presence of an outlier and lower medium. "→" "shown" and "median"

Thanks for noticing. It will be corrected.

17. P17: Growing season simulations. It would be good to have more background about the potential transpiration and the precipitation during the considered growing season.

We will provide these data in Table 6 or add other table or figure.

References Royal Harvard Brooks and A. J. Corey. Hydraulic properties of porous media. Hydrol. Paper, (3), 1964.

Q De Jong van Lier, J C Van Dam, K. Metselaar, R. De Jong, and W H M Duijnisveld. Macroscopic root water uptake distribution using a matric flux potential approach. Vadose Zone Journal, 7(3):1065–1078, 2008.

Q De Jong van Lier, J C van Dam, A Durigon, M. A. Santos, and K Metselaar. Modeling water potentials and flows in the soil-plant system comparing hydraulic resistances and

transpiration reduction functions. Vadose Zone Journal, 12(3), 2013.

N J Jarvis. A simple empirical model of root water uptake. Journal of Hydrology, 107(1): 57–72, 1989.

NJ Jarvis. Simple physics-based models of compensatory plant water uptake: concepts and eco-hydrological consequences. Hydrology and Earth System Sciences, 15(11):3431– 3446, 2011.

Mathieu Javaux, Tom Schr ÌĹoder, Jan Vanderborght, and Harry Vereecken. Use of a three- dimensional detailed modeling approach for predicting root water uptake. Vadose Zone Journal, 7(3):1079–1088, 2008.

Mathieu Javaux, Valentin Couvreur, Jan Vanderborght, and Harry Vereecken. Root water uptake: From three-dimensional biophysical processes to macroscopic modeling approaches. Vadose Zone Journal, 12(4), 2013.
* * *

---

## Author Comment (AC3) · 29 Jun 2016

**Reply to "Interactive comment on "Determination of empirical parameters for root water uptake models" " by Referee#3**

In response to the Anonymous Referee #3 :

We are thankful for your critical reading, constructive questions and suggestions. In the following we address your major comments. The response to the minor comments can be found in the supplement.

i) Regarding the discussion of the empirical models, we hope that the modifications (suggested by the referees) made in the revised manuscript will improve the discussion

and understanding.

ii) For most applications transpiration is more relevant than uptake distribution. For specific applications in which average water content in soil layers is of interest, root water uptake distribution can play a major role. Indeed, fitting the models with local uptake and transpiration simultaneously, and using a proper weighting scheme, might lead to more reliable results. Nevertheless, fitting the models to root water uptake only also provided good predictions of relative transpiration (by the models that showed good performance on predicting RWU) as shown by the statistical indices of Fig. 9, suggesting that using RWU only for such a task is quite sufficient. Conversely, using transpiration for such a task can lead to wrong predictions of RWU uptake distribution. We could use some simulated scenarios to show this.

iii) It is correct that those models that use matric flux potential are mathematically closer to the reference model, an advantage for the comparison, and this should be emphasized. A short discussion about this will be placed in the text.

iv)" One of the critical points concerning the Feddes stress response function in combination with the Jarivs (1989) compensation approach, the authors mention, is that the models fail to predict compensation under wet conditions, where alpha is 1 for different matric potentials. The modification using martic flux potential with distinct critical point ($M_c$) will perform alike. This is ok but should be discussed". A critical comparison between the models is made from line 23, page 14 to line 2, page 15. We will address this fact also.

v) Regarding the fact the "Model PM mixes stress reduction described by pressure head and compensation calculation based on matric flux potential". Conceptually the two models distributes RWU over depth by taking into account root length density and a hydraulic function to account for the effects of soil water in partitioning RWU. Any hydraulic function could be used, however the matric flux potential property seems a good alternative since it integrates both effects of soil hydraulic conductivity and soil

pressure head. This will define $S_p$ in the model. The actual local uptake can then be obtained by applying a stress response function $\alpha$ of any type, and for PM $\alpha = \alpha(h)$ is used. Thus, the fact that PM mingles $M$ and $h$ is not conceptually unreasonable.

vi) Using variable boundary conditions would provide more information content of the "measurements" as you comment as compared to the used constant boundary condition. We will comment this limitation of the work, however the applied scenarios included distinct hydraulic conditions, submitting the models to a wide range of conditions.

vii) Indeed, it is important to discuss about other existing physical models. We will include a discussion about the models you suggested. We will also discuss the limitations of the De Jong van Lier et al. [2013] model and emphasize that this work deals with only reduction of RWU/transpiration due to soil water stress.

viii) Although considering daily variation of $T_p$ during the day would give more detail about the predictions, the simulations performed did provide important features to strictly analyze De Jong van Lier et al. [2013] model as shown in section 4.1. In most applications root water uptake models are performed with no variation of $T_p$.

ix) We will consider your suggestion about the title, as it was suggested by N. Jarvis in RC1 comment.

References

Q De Jong van Lier, J C van Dam, A Durigon, M. A. Santos, and K Metselaar. Modeling water potentials and flows in the soil-plant system comparing hydraulic resistances and transpiration reduction functions. Vadose Zone Journal, 12(3), 2013.

N J Jarvis. A simple empirical model of root water uptake. Journal of Hydrology, 107(1): 57–72, 1989.

Please also note the supplement to this comment:

http://www.hydrol-earth-syst-sci-discuss.net/hess-2016-59/hess-2016-59-AC3-supplement.pdf

[Figure]

**Supplement:**

Next we respond to the minor comments.

1)Page 1, Lines 7 to 8: "The simulated scenarios give more insight into the behaviour of the physical model, especially under wet soil conditions and high potential transpiration rate." This statement seems not to be important for the abstract and can be omitted.

We agree and it will be omitted

2) Page 1, Lines 10 to 11: "...for the scenarios of low RWU "compensation". Better: "...for the scenarios for which RWU "compensation" is expected to be low." or ". . .for the scenarios for which the physical model predicts low RWU "compensation.""

OK, we will consider this

4) Page 1, Lines 13ff: When the Jarvis model is criticized it should be stated that the modifications are conceptually closer to the reference model.

Agreed, this will be discussed.

5) Page 1, Lines 13 to 14: "Incorporating a newly proposed reduction in the Jarvis model..." Consider: "Incorporating a newly proposed reduction function in the Jarvis model..." I did not find a statement about the performance of the Jarvis (2010) model in the abstract.

Ok. We will add a statement about the performance of JMII

6) Page 2, Lines 17 to 18: Models that do not account for compensation are under some circumstances (not all) less accurate, e.g. for coarse to medium textured soils and high root length density.

Agreed, we will correct this sentence.

7) Page 5, Line 24: "non-homogeneous" consider "heterogeneous". "For non-homogeneous conditions, RWU for lower R can be the same for higher R depending on the stress level" Consider: For heterogeneous conditions, RWU for lower R can be the same as for higher R depending on the stress level..." Maybe I am mistaken but I do not see this in Fig. 2: For a certain leaf pressure head (for example -110 m), the RWU for R=0.01 is always lower than for R=0.1 and RWU for R=0.1 is always lower than for R=1.

We will consider your suggestions. Indeed, it does not always happen and depends on $h_l$

value. It will be corrected.

8) Page 7, Line 3: Consider another word than obscure. Compensation will certainly (and shall) enhance uptake (by the factor $\alpha_2$) in some depth compared to the value given by alpha. To me the specific problematic issue is that in case of homogeneous alpha smaller than 1 and $\omega_c$ smaller than 1, these models lead to uptake greater than given by the homogeneous value of alpha or, more generally, that relative transpiration can be higher than given by the highest value for alpha in case of heterogeneous alpha distribution with depth (see e.g. Skaggs et al., 2006, Simunek and Hopmans, 2009, Peters, 2016).

This part will be rewritten as also suggested by N. Jarvis in RC1 comment.

9) Page 7, Lines 3 to 5: If I understand it right, this holds only for the combination of the Jarvis model with the Feddes stress function for which alpha is 1 for different pressure heads (i.e. between $h_2$ and $h_3$).

In that case we agree with Jarvis's comment (see point 7 of RC1 comment) that this sentence is wrong since Jarvis [1989] can predict compensation by $1/\omega_c$ under wet conditions

10) Page 7, Line 14: Consider "conceptually" instead of "numerically"

OK

11) Page 8, Lines 14-15: I cannot follow: $\rho$ and $M$ as defined here do not occur in the Jarvis (1989) model.

This is the result of comparing Jarvis [1989] model to the De Jong van Lier et al. [2008] model. The models can be correlated for stressed conditions if $\alpha$ and $\beta$ are given by eq. 18 and 19, respectively. For stressed conditions, substituting these eqs into the Jarvis [1989] model gives the same eq. for $S$ for both models. For unstressed condition, however, a different equation is found, eq. 21.

12) Page 10, Lines 2 to 14: Consider using subsection header such as "3.1 Applied models"

OK

13) Page 10, Lines 19 to 20: A free drainage boundary condition is usually used for the case with very deep groundwater level so that groundwater cannot influence the soil. Then the assumption is that at a reasonably deep layer below the root zone the hydraulic

gradients are close to unity. This is certainly not the case at the bottom of the root zone. I would suggest to set this boundary condition at a depth of at least 1 m or 1.5 m.

This is an important point and requires a careful justification. We used free outflow close to the bottom of the root zone. Many studies of water flow in soils without roots use the unit hydraulic gradient in the entire profile as a reasonable hypothesis. Extracting roots of course change this scenario dramatically, but simulated root length densities were already very low in the bottom part of the rooted zone. Nevertheless, changing the depth of free drainage will alter the water regime, especially in the lower part of the soil profile. That may, on its turn, change the simulated uptake pattern. Other important scenario changes might also be studied, like more or different soils with distinct soil hydraulic properties. Any alteration of this kind implies in a whole new set of scenarios and simulations and a considerable job in analyzing them and possibly lead to some new discussion or insight, but will it be a crucial factor in the comparison between the RWU models? We think it will not change the conclusions and would rather like to decline from this suggestion. We may, however, include a comment/thought about the issue in the discussion part.

14) Page 10, Line 24: "Soil date..." should be "Soil data.."

OK

15) Page 10, Line 26: "These soils are identified in this text as clay, loam and sand (Table 3)." Consider "These soils are identified in this text as clay, loam and sand."

OK

16) Page 11, Line 12ff: Please specify in this section at which depths and which time interval the data for $S$ and $S*$ were taken and used to minimize $\Phi$. Consider to fit also transpiration rates and use a weighted least squares scheme instead.

OK, this will be specified. We already addressed this point in point ii of major comments.

17) Page 11, Line 15: "...the objective function to be optimized..." Consider "...the objective function to be minimized..."

OK

18) Page 11, Line 25: For a nonlinear problem with a model error, i.e. with models that

do not fit the data well, there might be several local minima. Did all fitting runs lead to the same minimum? If not I would try to use more starting points to be sure or even a global minimization scheme.

For most cases the same minimum was found whatever was the starting point. In some cases we found different values and then used the "the lowest minimum".

19) Page 12, Lines 1 to 2: "This guaranteed that RWU predictions from SWAP corresponded to the best fit of each empirical models to the De Jong van Lier et al. (2013) model." I do not understand this sentence and how it refers to the statement that parameter fitting was only applied for the drying out scenario.

This sentence was meant to emphasize that the optimizations were performed only in the drying-out scenarios and by optimizing the parameters the best fit to De Jong van Lier et al. [2013] model was reached.

20) Page 12, Lines 19 to 20: "Initial pressure heads were obtained by iteratively running SWAP starting with the final pressure heads of the previous simulation until convergence." I do not understand. What converged to which values? And why was the initial condition optimized?

SWAP was run until the initial soil pressure head set values were equal to the end pressure head set values.

21) Page 13, Line 3: "The patterns for the sand and loam soil (not shown here) show very similar features." This is not immediately clear to me since matrix flux potential $(M)$ for the sand is very different from M of clay. In a sand most of the water is available under very low energy densities and thus I would expect that for sand, transpiration is prolonged much longer at potential rates and the drop of $T_a$ to be much steeper after onset of transpiration reduction. Could you discuss this briefly in 2 or 3 sentences?

The RWU predictions for sand soil are very close from what you inferred. We will make a short discussion to make this clear.

22) Page 13, Line 14: "... increases the reduction of. . ." consider "... leads to faster reduction of..."

OK

23) Page 13, Line 15: " assumes a parsimonious relationship..." do you mean "assumes a direct relationship..."

We tried to say a simple relationship when compared to other empirical relationships, ex. Fisher et al. [1981]

24) Page 14, Line 23ff, Tab. 5 and Fig. 6: For Sand with $Tp = 1mm/d$ and $R = 1cm/cm^3$ using the JM: $\omega_c = 1$, $h_3 = 0$ means that transpiration must be reduced from the beginning, since $h > 0$ from the beginning and compensation cannot take place. I cannot see this in Fig. 6, where transpiration is equal to $T_p$ for a prolonged time: Is it due to a very small reduction of $\alpha_f$, so that $T_a$ is smaller than but still close to $T_p$? Please discuss briefly.

The discussion of Line 23ff makes it clear to me that fitting not only the uptake pattern but also actual transpiration (see major comments) would increase model performance of the conceptual models. Then compensation would be most likely predicted.

Indeed, this is due to the small reduction of $\alpha$. In fact, fitting the models to transpiration will improve these two models performance regarding transpiration, but reducing RWU predictions. It might not be worthy forcing these models to mimic transpiration, whereas other models can mimic well both variables by fitting to only RWU. We can fit the models to transpiration in some scenarios to show this.

25) Page 15, Lines 5 to 6: $h_s$ cannot be lower than $h_4$ if only transpiration but no evaporation is considered.

Agreed. We will correct this. In fact, $h_s$ becomes close or equal to $h_4$.

26) Page 15, Lines 16 to 20 and general: "performs better", "overestimates RWU", . . . Please discuss the performance of the conceptual models always with respect to the VLM since you compare models. A comparison with real data is still the best benchmark.

We will add this.

27) Page 15, Lines 21ff: Here fitted models are compared by statistical measures like $E$ and $r^2$. Since the fitted models use different numbers of adjustable parameters such a comparison is not justified: More free parameters mean more flexibility and thus a better "chance" to fit the data. Please consider using other measures, which account for number

of fitted parameters, like AIC (Aikaike, 1974).

We will consider this as also discussed in the reply of RC1 comment.

28) Page 15, Line 25: ". . .models (except for JM and JMm by setting $\omega_c > 1$) are..." This can be omitted since $\omega_c > 1$ makes conceptually no sense.

We think it does make sense. See point 14 of RC1 comment.

29) Page 16, Lines 16 to 17: "The optimal $h_3$ and $M_c$ values (Table 5) for FM and FMm, respectively, increase as R or Tp increases, contradicting their conceptual relation to $R$ and $T_p$ levels" I see the contradiction only with respect to increased $R$ but not to increased $T_p$.

Yes, we will correct this sentence.

30) Page 16, Lines 31ff: I assume that parameters $h_3$ and $\omega_c$ for JM are highly correlated. Can you give information about parameter correlation? Moreover, such parameter correlation might be due to model structure but also due to data used for fitting the model. Therefore, I repeat my suggestion to use not only the drying out scenario for model calibration but the scenario with changing boundary conditions. This might reduce correlations.

We will analyse this correlation and discuss it.

31) Page 17, Lines 1 to 2: What are $l$-values? $L_m$ and lambda respectively. Please unify.

They all will be referred as to $l_m$.

32) Page 17, Line 4ff: A figure with the cumulative transpiration over time would be interesting to see if there are under-/over-estimations for specific time intervals in the complete season.

We will add transpiration information.

33) Page 17, Line 23: The statement that JMII is poor in performance should be discussed with more caution since it was not adjusted to the reference model. Thus, this finding can be expected. The same holds to a less extend to the models for which only one parameter was adjusted.

We will analyse this more carefully. The Akaike information criteria measure, to be include, will help in this analysis.

34) Page 17, Line 24: This is a very daring conclusion, since the reference model and the proposed models have partly a similar structure (see above).

As commented above, we will discuss this more.

35) Conclusions section: I could not find a single conclusion. This is rather a summary and not a conclusion.

If required, we can alter the writing stIf required, we can alter the writing style of this section.

36) Page 17, Line 32: ". . .especially under wet soil conditions and high potential transpiration." Why do the simulations yield insight especially under wet soil conditions?

For high $T_p$ and low $R$ under wet conditions it was shown that $T_p$ can not be achieve and also how plant hydraulic parameters relate to this.

37) Page 19, Lines 21 to 22: This paper is certainly not in press.

It will be corrected

38) and Figures Table 3: Although the Mualem/van Genuchten model is well known the equations should be stated in the text to make it easier to assign the parameters. What, for example, is lambda? I guess the so-called tortuosity parameter in Mualem's model, but I am not sure. Alternatively, Tab. 3 can be completely omited and the functional relationships of $\theta(h)$ and $K(h)$ might be plotted in an extra figure.

We will follow your suggestion

39) Table 4: I cannot find $l_m$ for PM and PMm in the text. Do you mean lambda instead of $l_m$?

Yes, it will be corrected.

40) Table 5: In the text root length density is $R$ here it is $R_d$.

It will be corrected.

41) Table 6: For comparison: what was the value for potential transpiration

42) Fig. 1: a) since $h_1$ and $h_2$ are set to zero in all simulations, Fig. 1,a should account for that and start with $\alpha = 1$ at $h = 0$. b) since $M_c$ for $T_p$=1 mm/d is different from $M_c$ for $T_p = 5$ mm/d, this should be indicated in Fig. 1b using $M_{c,l}$ and $M_{c,h}$ , similarily to $h_{3,l}$ and $h_{3,h}$ in Fig 1,a.

We will improve this accordingly.

43) Fig. 3: Should only contain the three root distributions used in this study.

Because we chose to set $b = 2$, it would be good to graphically see how $b$ affects the curve.

**References**

Q De Jong van Lier, J C Van Dam, K. Metselaar, R. De Jong, and W H M Duijnisveld. Macroscopic root water uptake distribution using a matric flux potential approach. *Vadose Zone Journal*, 7(3):1065–1078, 2008.

Q De Jong van Lier, J C van Dam, A Durigon, M. A. Santos, and K Metselaar. Modeling water potentials and flows in the soil-plant system comparing hydraulic resistances and transpiration reduction functions. *Vadose Zone Journal*, 12(3), 2013.

M J Fisher, D A Charles-Edwards, and M M Ludlow. An analysis of the effects of repeated short-term soil water deficits on stomatal conductance to carbon dioxide and leaf photosynthesis by the legume macroptilium atropurpureum cv. siratro. *Functional Plant Biology*, 8(3):347–357, 1981.

N J Jarvis. A simple empirical model of root water uptake. *Journal of Hydrology*, 107(1): 57–72, 1989.

---

## Author Response (AR1)

**Author's Response**

We are thankful for the critical reading, constructive comments and suggestion made by all referees. Their comments were very important to enhance our paper. Below we address all questions and comments made by each referee.

**N. Jarvis: major comments**

Regarding the conclusion that proposed models are recommend, it was slightly changed after taking into consideration the Akaiki information criteria. Regarding the values of $\lambda$ (changed to $l_m$) of the proposed models, they can be greater than 1, as discussed in more detail in points 9 and 16. In applying the evaluated models in blind predictions the JMII may have more advantages over the other models as it is more physically based. This was pointed out in the conclusion. One of the reason why the models PMm and JMm perform better than JMII can be seen now in the discussion, at the end of Section 4.2.

We considered your suggestion for the title.

Specific questions:

1. Page 2, lines 24-25 (and line 3 in the abstract): I am not so convinced of this. I would prefer to use a physics-based model even if it did have two or three more parameters, as long as they were, in principle, measurable. The limiting leaf water potential is quite well known, at least

R.: We also would prefer using a physics-based model, but in practice it appears not to be appealing. Root water uptake (RWU) models are usually embedded in larger hydrological models, for instance the ecohydrological model SWAP (Van Dam et al., 2008), and most users are unfamiliar with plant hydraulic parameters, making them to prefer the simplicity of empirical models like the Feddes et al. (1978) model, as long as empirical parameters are available. Besides, apart from the well-known limiting leaf water potential, radial root hydraulic conductivity has a strong effect on RWU distribution as shown in the paper and it is not easily available.

2. Page 4, line 5–8: the sentence starting ... "using $h$ seems ...." is wrong, as the authors know well enough. It is immediately contradicted by the text at lines 14-21 on the same page (and by the results shown later in the paper). This sentence should be deleted. The remaining text just says that various forms have been proposed for the $\alpha$ function, but that making $\alpha$ depend on M is physically the most plausible. This is quite sufficient

R.: The sentence is in fact misleading. It was intended to say "Comparing to $\theta$, $h$ seems to be more feasible..." instead of "Using $h$ seems". It was corrected as such.

3. Page 5, equation 8: $h_0(z)$ is not defined, as far as I can see?

R.: It is defined now

4. Page 5, lines 20-22: yes, it would be good if you mentioned this phenomenon by its name: hydraulic lift or hydraulic re-distribution. You could also cite Jarvis (2011) here, since he discussed and clarified the relationship between water uptake compensation and hydraulic lift in some detail (see the text in relation to equations 13 to 15 in the final version of this paper, not the HESS discussion paper that you cited: see point 4. under "Presentation")

Agree, it is important to mention the name of the phenomena as well as cite Jarvis (2011). These changes are incorporated in the text.

5. Page 6, lines 1-3: I know what you are trying to say here, but it is not so well expressed. You could replace i.) " .... is only relevant" by ".... it only needs to be explicitly addressed ...." and ii.) "... becomes less important" by "... is not necessary". This would help, but you could also add a sentence at the end of this saying that the effects of compensation can nevertheless be explicitly discriminated and identified in physics-based models. This is demonstrated in Jarvis (2011) in the text related to equations 13 and 14 in that paper (again, in the final version).

The text is improved accordingly. However, we don't think it is always possible to explicitly discriminate and identify the effects of compensation in physically-based models. Such relation (Jarvis (2011) eq. 13 and 14) was easily found for the De Jong van Lier et al. (2008) model comparison (Jarvis, 2011). Furthermore, adding this comment would be contrary to our general reasoning when we add that "In physical models, discriminating compensation is not necessary since in such models "compensation" follows implicitly from the RWU mechanism".

6. Page 6, lines 11-12: "In principle, any definition of $\alpha$ is applicable...". Yes, perhaps, but it does make a difference to the results of course, as you demonstrate very well later in the paper! But what is definitely not debatable is that Jarvis (1989) used a threshold type function for $\alpha$ based on water content (degree of saturation). The reason for adopting this approach was discussed by Jarvis (1989) in relation to the experimental evidence available at that time and no other type of function was considered. The fact that you adopt a Feddes-type function means that in the rest of the paper you cannot refer to this model as the Jarvis (1989) model. It is a modified Jarvis (1989) model, in exactly the same way that JMm is also a modified Jarvis (1989) model, where the threshold water content function is replaced by a threshold function of matric flux potential: in other words, you investigated two different modified Jarvis models and you should refer to them as such, both in table 1 and throughout the rest of the paper, including the abstract (perhaps you could call them JMm1 and JMm2?)

R. Indeed, any kind of $\alpha$ might provide different predictions. We agree that using the Feddes reduction function in the Jarvis (1989) model is also a modification of the Jarvis (1989) model. Thus, we renamed the Jarvis (1989) model to JMf. We also moved the statement "In principle, any kind of $\alpha$ is applicable..." to the end of the section, and then introduced the modified version JMf.

7. Page 6, line 28 to page 7, line 4: this is a little vague. You followed quite closely what Skaggs et al. (2006) wrote in this section, but since they wrote their paper ten years ago, it is now much better established exactly how the original Jarvis (1989) model departs from physicality. This was clarified in the papers by Jarvis (2010, 2011), which you also discuss in the following section. There are two aspects to this:

i.) the choice of function for $\alpha$. The threshold function chosen by Jarvis (1989) doesn't make complete physical sense, as the local resistance to uptake should in principle increase continuously as the soil dries (e.g. like equation 18). Jarvis (1989) discussed this choice in terms of the overall resistance to uptake being dominated by an air gap between soil and root which might only develop after a certain critical water deficit was reached: this choice was strongly influenced by experimental studies which showed such an effect. Also, at high soil water contents, the overall resistance to uptake in the soil-plant system would be dominated by plant resistances, which may be more or less constant. Thus, a threshold function might be a good choice from an empirical point of view. In this respect, it can also be pointed out here that the authors also adopt a threshold $\alpha$ function in the PMm model.This model is the one the authors finally recommend, because it works best, although it can certainly be criticized on the same grounds (i.e. that it "affronts the definition of $\alpha$").

ii.) Compensation under non-stressed conditions. As you point out, under non-stressed conditions the Jarvis (1989) model does give a different uptake distribution compared with the de Jong van Lier physical model. However, it is wrong to imply that the Jarvis (1989) model does not predict any compensation under non-stressed conditions (page 7, line 4). Under non-stressed conditions, water uptake is increased by a factor of $1/\omega$ in all layers (regardless of the pressure head distribution) to maintain transpiration at the rate demanded by the atmosphere during soil drying. It is also not wrong in principle to link compensation to plant stress (page 7, line 3): the onset of stress certainly does affect the nature of compensation: this is demonstrated in Jarvis (2011) in the text following equations 13 and 14 for the physics-based model of de Jong van Lier (2008).

For the above reasons, I strongly suggest that you delete the text on page 6 line 28 to page 7, line 4 and replace it by a short sentence that simply states that the Jarvis (1989) model departs from complete physicality in some respects and that this is explained in the following section. Then at the end of the next section (i.e. after equation 21) you can briefly summarize how the Jarvis (1989) model departs from physicality, based on the comparison with the physics-based model that is represented by equation 14-21. This will be very much clearer.

This as interesting discussion regarding the expected physical behaviour of reduction functions. We agree with the arguments. Therefore, we replaced the mentioned parts by shortly referencing to Skaggs et al. (2006) and Javaux et al. (2013). Finally we discussed a bit more about Jarvis (1989) model before showing how it departs from physicality.

8. Page 7, lines 5-12. The parameter $h_3$ does not exist in the Jarvis (1989) model (see lines 10-11 especially). I think this paragraph can be deleted (or perhaps moved to the results and discussion section). At the very least, readers should be reminded that the original Jarvis (1989) model does not use a Feddes-type $\alpha$ function.

We rearranged this paragraph by first defining JMf, then kept this discussion regarding to JMf.

9. Page 8, line 22: you should add the limits for $\lambda$ here. If compensation means that water uptake increases from sparsely rooted layers, then $\lambda$ must lie between zero and 1. Also, you should replace "deeper soil layers" by "more sparsely rooted layers" to be strictly correct.

We added the limits for $\lambda$ in the text. In fact, $\lambda$ is not restricted to the domain between 0 and 1. This is described in the text

10. Page 9, lines 23 to 26: I wonder what it is about your modification to the Li model (the use of the matric flux potential in a threshold function) that resolves the conceptual difficulties with the original formulation that you described earlier on page 9 at lines 3 to 8. As far as I can see, the same objections should be equally valid for this modified version as for the original model. This should be clarified and the text modified accordingly.

The main objection regarding the Li et al. (2001) model is the use of $\alpha$ in $\zeta$ (eq. 22). Thereby, "compensation" taking place before transpiration reduction (when $\alpha = 1$ for all soil layers) can not be computed: RWU is distributed over depth only by $R^\lambda$. By using $M$ instead of $\alpha$, "compensation" before transpiration reduction can be computed. As $M$ integrates both the effects of $K$ and $h$, it might be a better soil hydraulic function than $K$ or $D$ (Molz and Remson, 1970; Selim and Iskandar, 1978) to account for the effects of soil water in partitioning RWU. Such comments are added into section 2.2.4.

11. Page 11, lines 13-28: As I understand it from table 3, you only have a maximum of two parameters to calibrate for all the models, while each parameter is constrained within known limits. This means that a "brute force" grid search for optimum parameter values would be preferable to the method you chose, since you could be sure of avoiding risks of finding local minima (although it might be slower). I am sure there is no need to repeat the calibrations, but maybe you could mention this?

R.: The "brute force" grid search is a very slow method. As there are many scenarios and some models to evaluate, we do not think it is interesting to mention it since it would not be applicable in practice (a very small grid would also be required to avoid finding relative minimum).

12. Page 13, lines 23-24: yes, this may be why a constant value of $\omega_c$ often seems to work quite well. Maybe you could add a comment to this effect, and also refer to your equation 20 and cite Jarvis (2011), where this aspect is discussed in detail.

R.: As eq. 20 gives an expression for $\omega_c$ derived from the De Jong van Lier et al. (2008) physics-based model (Jarvis, 2011), it indeed helps in accounting for some aspects relating RWU phenomena. A constant $\omega_c$ might be quite robust as can be inferred by eq. 20 and from common field observations. However, adding such a comment in this part (page 13, lines 23-24) might get out of the context of the paragraph

13. Page 15, line 18: You should replace "either R or M" by "both R and M". But this sensitivity to $M$ is in principle also present in the empirical models that include $M$. Why is it more important for JMII? Is it because this model is not calibrated? Or is it because of the different type of function? I can believe that predictions of JMII are, in comparison with the empirical models, more affected by the value of $M_{max}$, which must be a very uncertain parameter, not least because the Mualem-van Genuchten model of soil hydraulic properties is known to have an incorrect form close to saturation (since it does not allow for a maximum size of pore in soil). These questions should be clarified.

R.: This is explained in the text, at the end of Section 4.1.2. A new graph is inserted to support the discussion.

14. Page 15, line 25: it could also be noted (perhaps by referring to equation 20) that $\omega_c > 1$ is not physically unrealistic.

R.: Yes, $\omega_c > 1$ is not physically unrealistic and it is implicitly stated in line 25: "by setting $\omega_c > 1$". It means JMII and JMm can predict $T_a/T_p < 1$ for the low $R$-high $T_p$ scenarios as VLM did, but we decided to not assess these results since it is not possible to compare to other empirical models.

15. Page 16, line 3: This is misleading. The Feddes function for $\alpha$ is not part of the Jarvis (1989) model.

R.: The text was corrected accordingly.

16. Page 17, lines 1-2: it is confusing that different symbols are apparently used for one of the parameters in the Li-type models. In equation 25, $\lambda$ is used, whereas in the text here and in table 5, $l$ is used, while in table 4 $l_m$ is used. I believe they are all the same parameter?

If I understood it correctly, I don't see how you can write that the optimal values of $\lambda$ follow a logical relation to $R$ and $T_p$ (line 1). In many cases, and especially for low root densities, values of $l$ (i.e. $\lambda$?) in table 5 are larger than 1, which implies to me that compensation is working incorrectly in these scenarios (it is decreasing uptake in the more sparsely rooted layers). Also, in table 4, it is stated that $l_m$ (i.e. $\lambda$) was constrained to take values less than or equal to 1. If I understood it correctly, the results in table 5 suggest that this was not actually the case in practice.

R.: All symbols refer to the same parameter and we corrected this by changing them to $l_m$. Regarding the $l_m$ parameter limit values, the upper values for $l_m$ were constrained to 3. Conceptually, there is no inconsistency in taking $l_m > 1$. Indeed, $l_m = 1$ means no compensation at all and $l_m < 1$ implies compensation. Values of $l_m > 1$ simply indicates that the upper soil layers are more important for RWU distribution. This is now explained in section 2

17. Section 4.2, table 6: can you give the total precipitation and potential transpiration here? It's good to get a rough idea of how much stress occurred in these simulations.

The old table 6 was substituted by a figure of the time course of cumulative transpiration, precipitation and $T_p$. The values at the of the period are also given in the figure.

18. Page 18, lines 10-12: you did not test the Jarvis (1989) model (see earlier comments).

R.: It is corrected

19. Page 18, line 12: I did not get a good understanding of why the JMII model does not work so well for high R–low Tp scenarios (i.e. high compensation). I would have thought that, in principle, it should work OK. Please briefly explain what you think the reasons are for this.

R.: This is explained at the end of section 4.1.2.

20. Lines 16-18: I think this is too optimistic, as this test was not a very tough one. You had the same plants (identical roots) and the same three soils. How would it look if you had simulated different scenarios (soils, plants)? I think you would need to re-calibrate the empirical models. How useful is that?

R.: Yes, the results would be different. However, it is useful to show that the methodology used to calibrate the models is robust and can be used to assess empirical models and sensitivity of the empirical parameters in order to provide a full calibration of the empirical models in a next step.

**Anonymous Referee #2: major comments**

i) Regarding the dependence of the model parameters on transpiration rate, the stress reduction function parameters are already shortly discussed on how they depend on potential transpiration. Adding a discussion about this dependence in the review section would be rather repetitive. Thus, we think is better to discuss it only in the results.

ii) We think the general advantages concerning the use of empirical models as compared to the De Jong van Lier et al. (2013) physical model can already be found page 2, lines 23 to 27.

iii) We included transpiration prediction in the conclusion.

iv) It not possible to directly retrieve root water uptake from measurements. This is one of the advantage of using physically-based models. The main purpose of this paper is to evaluate empirical models that can be sensitive to the variations of root water uptake due to different scenarios of soil and plant properties as well as climatic conditions as predicted by a physical model. By using root water uptake it is possible to strictly capture the root water patterns predicted by the models, whereas for instance if using soil water content the results can be "blinded" by the sensitivity of RWU on soil water content which vary with soil type. Using transpiration may lead to wrong predictions on root water uptake. This is addressed now in the paper, in section 4.1.5.

Specific questions.

1. P4: root length density $R$. Shouldn't that have dimension $L\,L^{-3}$ ?

Yes, it is corrected.

2. P4: The authors propose a stress function $\alpha$ which is a stepwise linear function of $M$. Since $M$ is a function of $h$, the new stress function will be a function of $h$ also. But the shape of the function will have a different shape than a piecewise linear function of $h$. Furthermore, the relation between the new stress function $\alpha$ and $h$ will depend on the hydraulic soil properties and will therefore be different in soils with a different texture. The original Feddes $\alpha(h)$ function depends on the transpiration rate as shown in Figure 1. Figure 1 suggests that the new stress function $\alpha(M)$ does not depend on the transpiration rate. I do not understand why the transpiration dependency of the stress function disappears when $\alpha$ is expressed as a function of $M$ since $M$ does not depend on the transpiration rate.

This is a very important observation. In fact the new $\alpha$ function also depends on potential transpiration rate $T_p$. This dependency is implicitly expressed in the critical value $M_c$ of $M$. Therefore, as in the case of the Feddes $\alpha$ function there should be two values for $M_c$: one for low $T_p$ and other for high $T_p$. Fig. 1 is now corrected.

3. P5: ln 15: "Because $T_a$ and $h_l$ are unknowns, eq. 8 and 10 cannot be solved analytically, but an efficient numerical algorithm is described in De Jong van Lier et al. (2013)." I did not understand this. I thought that either $T_a = T_p$ is known as a boundary condition so that $h_l$ can be calculated or $h_l = h_w$ is known and $T_a$ is calculated. I think that the reason why the $h_l$ (or $T_a$) cannot be derived directly is because the set of equations that needs to be solved (including also all $h_{0,i}$'s ) is non-linear in $h_{0,i}$.

As commented in the interactive discussion, the sentence is wrong. The set of equations can be solved analytically, but not in a direct way, for some special cases of Brooks and Corey (1964) soils. The sentence is now corrected.

4. P5 ln 17 and p 29 Figure 2: There are several things I do not understand about Figure 2. The figure caption says that the plant transpiration was set to 1 mm d$^{-1}$ . Shouldn't for a fixed rooting depth the root water uptake or sink term $S$ be constant and independent of the root length density $R$ until a threshold soil water potential is reached? This threshold will depend of course on the leaf water potential and the root length density. Can it be that the curves shown in Figure 2 shown the maximal possible sink term as a function of the soil water potential for different leaf water potentials and root length densities? But, when the root water uptake goes to zero, why doesn't the soil water potential then go to the leaf water potential? Now there seems to be

a 10 m difference between them. Second, why doesn't the root water uptake for a certain soil water potential then not increase with decreasing leaf water potential. For sufficiently large (small absolute value) soil water potential, the root water uptake becomes independent of the leaf water potential. I do not understand this since the water potential difference increases with decreasing leaf water potential and therefore the root water uptake should also increase with decreasing leaf water potential.

5    These question were address in the interactive comment

i

5. P7 ln 21: "where $T_{p_{\max}}$ is the maximum possible transpiration rate attained when $M_0 = 0$". This assumes that the minimal water potential at the soil-root interface is $h_w$ (wilting point). But, doesn't this minimal water potential depend also on the critical leaf water potential $h_l$?

10    Yes, it depends on $h_l$. The limiting pressure head at the soil-root interface (called $h_{ws}$ to avoid confusion) is less negative than the limiting $h_l$ (called $h_{wl}$). Although $h_{ws}$ depends on $h_l$ and on plant and soil hydraulic parameters, for the sake of simplicity we considered it as constant and equal to -150 m. The $h_{ws}$ value was the limiting value used in the empirical models that depends on $M$ and is listed in Table 2.

6. P7 Eq. 17: Why is $M_0$ constant with $z$? The soil root interface water potential can depend on the depth, can't it?

15    We take advantage of your question and correct eq 6 to explicitly make $M_0 = M_0(z)$. However, De Jong van Lier et al. (2008) did assume $M_0$ constant with depth in order to solve the problem of the two unknowns: $T_a$ and $M_0$. They made a justification for that, and we refer to their paper (De Jong van Lier et al., 2008) for more detail. With this assumption it was possible later on to Jarvis (2011) make a comparison with the Jarvis (1989) model.

7. P8 ln 15: "The Jarvis (1989) model predicts RWU by a weighting factor between $\rho$ and $M$ throughout rooting depth". This
20    is not very clear to me. What do you mean with a weighting factor "between $\rho$ and $M$"? Do you mean a weighting factor that is equal to the product of $\rho$ and $M$?
An interesting feature of the analogy between the Jarvis model and the De Jong van Lier et al. (2008) model is that the analogy is derived based on the assumption that stress only occurs when everywhere at the soil-root interface limiting conditions are reached. It is assumed that $M_0$ is zero everywhere in the root zone. But, I am wondering whether the De Jong van Lier et al.
25    (2008) only predicts stress under these conditions. Can it be that stress occurs even though $M_0(z)$ is not zero everywhere in the root zone? If this is the case, then the analogy between the Jarvis and the De Jong van Lier et al. (2008) models is not given always when stress occurs.

A weighting factor was meant as equal to the product between $\rho$ and $M$ divided by the integral of this product over the rooting zone. As $M_0$ in the De Jong van Lier et al. (2008) model is constant over depth, stress is assumed to occur when $M_0 = 0$ over
30    the root zone.

8. P 8 ln 22: "The smaller $\lambda$, the more water is taken up in deeper soil layers" I would reword this to "... the more water is taken up from layers with a low root length density".

We improved this part and we took note of your suggestion.

9. P 9 ln 1: "RWU is calculated by substituting eq. 23 into eq. 3, following the Feddes approach." This implies that you multiply
35    Eq. 22 again by $a(z)$. So in the nominator, you get $\alpha^2$?

Yes, and it is an alternative case to write the equation.

10. P9 ln 16: Same comment as above.

No, in this case it will not happen since $D$ or $K$ is used to account for water availability.

11. P 9 ln 18: "In drier soil layers, $\Gamma$ is reduced, whereas in wetter soil layers $\Gamma$ is increased, thus increasing RWU in these layers before the onset of transpiration reduction." I do not understand this. If the soil dries out but faster in the upper layers where the root length density is higher than in the deeper layers, the deeper soil layers will not get wetter so $\Gamma$ will not increase in the deeper soil layers, which are still wetter than the upper soil layers. But, $\zeta(z)$ will increase in the deeper soil layers that remained wetter.

The sentence was rephrased. As you put out well, in fact $\Gamma$ in wetter soil layers will not increase, but $\zeta$ will do because $\Gamma$ in these layers will be less reduced compared to $\Gamma$ in the upper dryer layers.

12. P9: Proposed empirical model. Is in this model also the $\alpha(z)$ factor of the Feddes model used?

General question on the used models: The Feddes stress function $\alpha(z)$ is besides a function of the soil water potential, also a function of the potential transpiration rate. How is this considered in the different models? It should be noted that Eq. (20) suggests that $\omega_c$ in the Jarvis model is a function of the transpiration rate but the $\alpha(z)$ used in the Jarvis model is according to Eq. 18 not a function of the transpiration rate. Furthermore, the modified version of the Feddes model shown in Figure 1b suggests that there is no dependence of the $\alpha_m$ function on the transpiration rate and that $\alpha_m$ depends only on the matric flux potential. When looking at table 4, it seems that there is no transpiration rate dependence of the Feddes parameters.

The proposed root water uptake models are obtained by incorporating $\zeta_m$ into eq. 23, then into eq. 3. The PM uses Feddes reduction function whereas PMm uses the proposed reduction function $\alpha_m$ as shown in Table 1. This is described now in the text.
The dependence of the models on potential transpiration are implicitly built-in in the values of their empirical parameters that were optimized. For instance, in the Feddes reduction function there are two values for $h_3$: one for low $T_p$ ($h_{3l}$) and another for high $T_p$ ($h_{3h}$). The dependence of $T_p$ in the other models are accounted for similarly. We then optimized the models for two levels of $T_p$ (1 and 5 mm d$^{-1}$), therefore the optimized parameters are derived for low and high $T_p$.

13. P11 ln 26: "For high non-linear problems as the one in eq. 29 GLM depends on the initial values of b." This needs to be reformulated. The GLM does not depend on the initial values of b but the optimized parameter set may depend on the initial value of b since the GLM is a local optimization algorithm that may converge in a local minimum instead of the global minimum.

We agree with your observation. It was reformulated accordingly.

14. P 12: "3.2.1 Growing season simulation". This is not a sub section of the optimization section.

Yes, it is corrected.

15. P13 ln 8: "$h_w (= -200$ m)". I am confused here because at p 10 it is written: "The value of the parameter $h_4$ was set to -150 m.".

As discussed above in point 5, we used different abbreviations for them.

16. P15 ln 30: "showed by the presence of an outlier and lower medium. "$\rightarrow$" "shown" and "median"

Thanks for noticing. It is corrected.

17. P17: Growing season simulations. It would be good to have more background about the potential transpiration and the precipitation during the considered growing season.

A new plot was inserted and Table 6 was deleted.

**Anonymous Referee #3: major comments**

i) Regarding the discussion of the empirical models, we hope it is improved with the modifications made.

ii) Regarding fitting the models to temporal course of transpiration. We added a new subsection in which we discuss this. Instead of using a more problematic scheme, we show (for some models and scenarios) that fitting the models only to RWU can provide suitable relative transpiration predictions for the models that account for "compensation". Therefore, it seems unnecessary to use a more problematic weigh scheme for the paper purpose. Conversely, it is shown that fitting the models to $T_r$ leads to wrong predictions of RWU.

iii) It is correct that those models that use matric flux potential are mathematically closer to the reference model, an advantage for the comparison. We added a comment about this at the end of Section 4.2.

iv)" One of the critical points concerning the Feddes stress response function in combination with the Jarivs (1989) compensation approach, the authors mention, is that the models fail to predict compensation under wet conditions, where alpha is 1 for different matric potentials. The modification using martic flux potential with distinct critical point ($M_c$) will perform alike. It seems it is already discussed at the end of the this paragraph: "... Conversely, the JMm was able to reproduce considerably well the VLM pattern for these scenarios due to the shape of $\alpha_m$ as discussed above. As soon as $M > M_c$ in the upper layers, RWU decreased at a higher rate, compensated by increasing uptake from the wetter, deeper layers".

v) Regarding the fact the "Model PM mixes stress reduction described by pressure head and compensation calculation based on matric flux potential". Conceptually the two models distributes RWU over depth by taking into account root length density and a hydraulic function to account for the effects of soil water in partitioning RWU. Any hydraulic function could be used, however the matric flux potential seems to be a good alternative since it integrates both effects of soil hydraulic conductivity and soil pressure head. This will define $S_p$ in the model. The actual local uptake can then be obtained by applying a stress response function $\alpha$ of any type, and for PM $\alpha = \alpha(h)$ is used. Thus, the fact that PM mingles $M$ and $h$ is not conceptually unreasonable.

vi) Using variable boundary conditions would provide more information content of the "measurements", as you comment, as compared to the used constant boundary condition. The applied scenarios included distinct hydraulic conditions, submitting the models to a wide range of conditions. This is also discussed now at the of section 4.1.4.

vii) Indeed, it is important to discuss about other existing physical models. We briefly discussed this in the introduction.

viii) Although considering daily variation of $T_p$ during the day would give more detail about the predictions, the simulations performed did provide important features to strictly analyse De Jong van Lier et al. (2013) model as shown in section 4.1. In most applications root water uptake models are performed with no variation of $T_p$.

ix) The title was changed.

Specific questions.

1)Page 1, Lines 7 to 8: "The simulated scenarios give more insight into the behaviour of the physical model, especially under wet soil conditions and high potential transpiration rate." This statement seems not to be important for the abstract and can be omitted.

OK, it is omitted

2) Page 1, Lines 10 to 11: "...for the scenarios of low RWU "compensation". Better: "...for the scenarios for which RWU "compensation" is expected to be low." or ". . .for the scenarios for which the physical model predicts low RWU "compensation.""

OK, it was will considered

4) Page 1, Lines 13ff: When the Jarvis model is criticized it should be stated that the modifications are conceptually closer to the reference model.

It is discussed in the results.

5) Page 1, Lines 13 to 14: "Incorporating a newly proposed reduction in the Jarvis model..." Consider: "Incorporating a newly proposed reduction function in the Jarvis model..." I did not find a statement about the performance of the Jarvis (2010) model in the abstract.

Considered. A statement was added

6) Page 2, Lines 17 to 18: Models that do not account for compensation are under some circumstances (not all) less accurate, e.g. for coarse to medium textured soils and high root length density.

Agree, the sentence is corrected.

7) Page 5, Line 24: "non-homogeneous" consider "heterogeneous". "For non-homogeneous conditions, RWU for lower R can be the same for higher R depending on the stress level" Consider: For heterogeneous conditions, RWU for lower R can be the same as for higher R depending on the stress level..." Maybe I am mistaken but I do not see this in Fig. 2: For a certain leaf pressure head (for example -110 m), the RWU for R=0.01 is always lower than for R=0.1 and RWU for R=0.1 is always lower than for R=1.

The sentences were corrected. It seems Fig 2 does show that for a specific $h_l$ RWU decreases as $R$ decreases.

8) Page 7, Line 3: Consider another word than obscure. Compensation will certainly (and shall) enhance uptake (by the factor $\alpha_2$) in some depth compared to the value given by alpha. To me the specific problematic issue is that in case of homogeneous alpha smaller than 1 and $\omega_c$ smaller than 1, these models lead to uptake greater than given by the homogeneous value of alpha or, more generally, that relative transpiration can be higher than given by the highest value for alpha in case of heterogeneous alpha distribution with depth (see e.g. Skaggs et al., 2006, Simunek and Hopmans, 2009, Peters, 2016).

This is rewritten, following also the suggestions made by N. Jarvis in RC1 comment.

9) Page 7, Lines 3 to 5: If I understand it right, this holds only for the combination of the Jarvis model with the Feddes stress function for which alpha is 1 for different pressure heads (i.e. between $h_2$ and $h_3$).

This was rewritten. However, it seems any type of stress reduction function can be used, as for instance Jarvis (2010) used a different reduction function. The model essentially changes how/when transpiration is reduced: a new reduction function is introduced. Locally, any stress function can be considered for RWU.

10) Page 7, Line 14: Consider "conceptually" instead of "numerically"

OK

11) Page 8, Lines 14-15: I cannot follow: $\rho$ and $M$ as defined here do not occur in the Jarvis (1989) model.

This is the result of comparing Jarvis (1989) model to De Jong van Lier et al. (2008) model. The models can be correlated for stressed conditions if $\alpha$ and $\beta$ are given by eq. 18 and 19, respectively. For stressed conditions, substituting these eqs into the Jarvis (1989) model leads to the same equation for $S$ of De Jong van Lier et al. (2008) model. However, the same does not happen for unstressed condition, which leads to a different equation for $S$, eq. 21.

12) Page 10, Lines 2 to 14: Consider using subsection header such as "3.1 Applied models"

OK

13) Page 10, Lines 19 to 20: A free drainage boundary condition is usually used for the case with very deep groundwater level so that groundwater cannot influence the soil. Then the assumption is that at a reasonably deep layer below the root zone the hydraulic gradients are close to unity. This is certainly not the case at the bottom of the root zone. I would suggest to set this boundary condition at a depth of at least 1 m or 1.5 m.

This is an important point and requires a careful justification. We used free outflow close to the bottom of the root zone. Many studies of water flow in soils without roots use the unit hydraulic gradient in the entire profile as a reasonable hypothesis. Extracting roots of course change this scenario dramatically, but simulated root length densities were already very low in the bottom part of the rooted zone. Nevertheless, changing the depth of free drainage will alter the water regime, especially in the lower part of the soil profile. That may, on its turn, change the simulated uptake pattern. Other important scenario changes might also be studied, like more or different soils with distinct soil hydraulic properties. Any alteration of this kind implies in a whole new set of scenarios and simulations and a considerable job in analyzing them and possibly lead to some new discussion or insight, but will it be a crucial factor in the comparison between the RWU models? We think it will not change the conclusions and we did not make changes in this respect.

14) Page 10, Line 24: "Soil date..." should be "Soil data.."

OK

15) Page 10, Line 26: "These soils are identified in this text as clay, loam and sand (Table 3)." Consider "These soils are identified in this text as clay, loam and sand."

OK

16) Page 11, Line 12ff: Please specify in this section at which depths and which time interval the data for $S$ and $S*$ were taken and used to minimize $\Phi$. Consider to fit also transpiration rates and use a weighted least squares scheme instead.

OK, this was specified. We added a section in which it was discussed.

17) Page 11, Line 15: "...the objective function to be optimized..." Consider "...the objective function to be minimized..."

OK

18) Page 11, Line 25: For a nonlinear problem with a model error, i.e. with models that do not fit the data well, there might be several local minima. Did all fitting runs lead to the same minimum? If not I would try to use more starting points to be sure or even a global minimization scheme.

Mostly they led to the same minimum. In the case it did not happen, we compared the minimum and made the fitting runs again. It is now also added in the text.

19) Page 12, Lines 1 to 2: "This guaranteed that RWU predictions from SWAP corresponded to the best fit of each empirical models to the De Jong van Lier et al. (2013) model." I do not understand this sentence and how it refers to the statement that parameter fitting was only applied for the drying out scenario.

This sentence is just to emphasize that the optimizations were performed only in the drying-out scenarios and by optimizing the parameters the best fit to De Jong van Lier et al. (2013) model was reached.

20) Page 12, Lines 19 to 20: "Initial pressure heads were obtained by iteratively running SWAP starting with the final pressure heads of the previous simulation until convergence." I do not understand. What converged to which values? And why was the initial condition optimized?

The swap was run until the initial soil pressure head set values were equal to the end pressure head set values.

21) Page 13, Line 3: "The patterns for the sand and loam soil (not shown here) show very similar features." This is not immediately clear to me since matrix flux potential ($M$) for the sand is very different from M of clay. In a sand most of the water is available under very low energy densities and thus I would expect that for sand, transpiration is prolonged much longer at potential rates and the drop of $T_a$ to be much steeper after onset of transpiration reduction. Could you discuss this briefly in 2 or 3 sentences?

The RWU predictions for sand soil are very close from what you inferred. A short discussion was added.

22) Page 13, Line 14: "... increases the reduction of. . ." consider "... leads to faster reduction of..."

OK

23) Page 13, Line 15: " assumes a parsimonious relationship..." do you mean "assumes a direct relationship..."

We mean a simple relationship when compared to other empirical relationships, ex. Fisher et al. (1981)

24) Page 14, Line 23ff, Tab. 5 and Fig. 6: For Sand with $Tp = 1mm/d$ and $R = 1cm/cm^3$ using the JM: $\omega_c = 1$, $h_3 = 0$ means that transpiration must be reduced from the beginning, since $h > 0$ from the beginning and compensation cannot take place. I cannot see this in Fig. 6, where transpiration is equal to $T_p$ for a prolonged time: Is it due to a very small reduction of $\alpha_f$, so that $T_a$ is smaller than but still close to $T_p$? Please discuss briefly.

The discussion of Line 23ff makes it clear to me that fitting not only the uptake pattern but also actual transpiration (see major comments) would increase model performance of the conceptual models. Then compensation would be most likely predicted.

Indeed, this is due to the small reduction of $\alpha$. We added a subsection regarding fitting the models to $T_r$.

25) Page 15, Lines 5 to 6: $h_s$ cannot be lower than $h_4$ if only transpiration but no evaporation is considered.

Agreed. It was corrected. In fact, $h_s$ becomes close or equal to $h_4$.

26) Page 15, Lines 16 to 20 and general: "performs better", "overestimates RWU", . . . Please discuss the performance of the conceptual models always with respect to the VLM since you compare models. A comparison with real data is still the best benchmark.

ok

27) Page 15, Lines 21ff: Here fitted models are compared by statistical measures like $E$ and $r^2$. Since the fitted models use different numbers of adjustable parameters such a comparison is not justified: More free parameters mean more flexibility and thus a better "chance" to fit the data. Please consider using other measures, which account for number of fitted parameters, like AIC (Aikaike, 1974).

AIC measure was included.

28) Page 15, Line 25: ". . .models (except for JM and JMm by setting $\omega_c > 1$) are..." This can be omitted since $\omega_c > 1$ makes conceptually no sense.

In fact it does make sense. See point 14 of RC1 comment.

29) Page 16, Lines 16 to 17: "The optimal $h_3$ and $M_c$ values (Table 5) for FM and FMm, respectively, increase as R or Tp increases, contradicting their conceptual relation to $R$ and $T_p$ levels" I see the contradiction only with respect to increased $R$ but not to increased $T_p$.

It was corrected.

30) Page 16, Lines 31ff: I assume that parameters $h_3$ and $\omega_c$ for JM are highly correlated. Can you give information about parameter correlation? Moreover, such parameter correlation might be due to model structure but also due to data used for fitting the model. Therefore, I repeat my suggestion to use not only the drying out scenario for model calibration but the scenario with changing boundary conditions. This might reduce correlations.

We made a brief discussion about this.

31) Page 17, Lines 1 to 2: What are $l$-values? $L_m$ and lambda respectively. Please unify.

It was corrected.

32) Page 17, Line 4ff: A figure with the cumulative transpiration over time would be interesting to see if there are under-/over-estimations for specific time intervals in the complete season.

Figure of cumulative transpiration was added and Table 6 was deleted.

33) Page 17, Line 23: The statement that JMII is poor in performance should be discussed with more caution since it was not adjusted to the reference model. Thus, this finding can be expected. The same holds to a less extend to the models for which only one parameter was adjusted.

The use of Akaike information helped the discussion about this.

34) Page 17, Line 24: This is a very daring conclusion, since the reference model and the proposed models have partly a similar structure (see above).

Although the proposed models are close to the reference model, it was not guaranteed that these simple modifications would result in considerable improvements in their predictions.

35) Conclusions section: I could not find a single conclusion. This is rather a summary and not a conclusion.

We added more information into the conclusion, but kept the writing style

36) Page 17, Line 32: ". . .especially under wet soil conditions and high potential transpiration." Why do the simulations yield insight especially under wet soil conditions?

For high $T_p$ and low $R$ under wet conditions it was shown that $T_p$ can not be achieve and also how plant hydraulic parameters relate to this.

37) Page 19, Lines 21 to 22: This paper is certainly not in press.

It was corrected

38) and Figures Table 3: Although the Mualem/van Genuchten model is well known the equations should be stated in the text to make it easier to assign the parameters. What, for example, is lambda? I guess the so-called tortuosity parameter in Mualem's model, but I am not sure. Alternatively, Tab. 3 can be completely omited and the functional relationships of $\theta(h)$ and $K(h)$ might be plotted in an extra figure.

We added the equations, but we kept the table.

39) Table 4: I cannot find $l_m$ for PM and PMm in the text. Do you mean lambda instead of $l_m$?

It was corrected.

40) Table 5: In the text root length density is $R$ here it is $R_d$.

It is corrected.

41) Table 6: For comparison: what was the value for potential transpiration

The old table 6 was substituted by a figure showing the time course of cumulative transpiration, precipitation and $T_p$. The values at the of the period was also given in figure.

42) Fig. 1: a) since $h_1$ and $h_2$ are set to zero in all simulations, Fig. 1,a should account for that and start with $\alpha = 1$ at $h = 0$. b) since $M_c$ for $T_p$=1 mm/d is different from $M_c$ for $T_p$ = 5 mm/d, this should be indicated in Fig. 1b using $M_{c,l}$ and $M_{c,h}$ , similarily to $h_{3,l}$ and $h_{3,h}$ in Fig 1,a.

It was corrected.

43) Fig. 3: Should only contain the three root distributions used in this study.

Because we chose to set $b = 2$, it would be good the graphically see how $b$ affects the curve.

[revised manuscript text omitted]

---

## Referee Report (RR1)

I think that the authors have greatly improved their paper. It must be noted though that the paper is very technical. This implies that the methods and the definition of all the parameters need to be very clearly explained so that the reader can understand the paper. To my opinion, there is still some work to improve the clarity of the paper.

Furthermore, I think that the conclusion and discussion section of the paper can be strengthened further by adding some discussion about the dependence of root water uptake parameters on soil properties and climatic conditions. This is an important issue since root water uptake parameters are normally linked to the vegetation type and tables are provided that provide root water uptake parameters for a certain vegetation type. This paper actually shows that the root water uptake parameters of the empirical models also depend on the soil type and the climate. The question therefore arises whether the empirical models can be really considered to be parsimonious compared to the full physically based model. I think that this deserves some discussion in the final part of the paper.

General comments

Abstract: If you briefly mention what is behind the 'alternative empirical models' that are proposed, I think that the abstract could be improved. How do they differ from the Feddes and Jarvis models?

P6 ln 2-3: Maybe before going to De Jong van Lier et al. (2013) where the reader can find information about the algorithm, I think it would be helpful to explain how an equation with three unknowns $T_a$, $h_l$ and $h_0$ can be solved in general. First, I think you need to write that in equation 8, there are only two unknowns: either $h_0$ and $h_l$ or $h_0$ and $T_a$. In order to solve the equation for the two unknowns, additional equations are required. After formulating equations for $h_0$'s at different depths, Eq 1, making use of Eq. 6, can be used to solve for $T_a$ (or $h_l$) and the distribution of $h_0$ with depth in the soil profile.

P6 Figure 2: I am afraid that I still do not understand figure 2 and its caption. What I suppose that is shown in Figure 2 is the sink term for the case that the root length density and the soil water potential are uniform in the root zone (i.e. they do not change with depth in the root zone). As a consequence, also the sink term is uniform in the root zone and the transpiration rate is simply the sink term multiplied by the root zone thickness. So I do not understand that the plant transpiration was set to 1 mm d$^{-1}$.

P7 and P8: Comparison of the Jarvis model and the De Jong van Lier model and Figure 5. In my previous comments on the paper, I posed the question whether the analogy between the two models relies on the assumption that when stress occurs the water potential the root surface is every the same in the root zone. I also asked whether the model of De Jong van Lier only predicts stress when the water potential at the root surface is everywhere equal to the wilting point. If the De Jong van Lier model can predict that stress may also occur even when in parts of the root zone the surface water potential at the root surface is still above the wilting point, then the De Jong van Lier and Jarvis models may also deviate under stress conditions. I think that the main problem in the description here is that the authors are not fully consistent in defining the stress conditions: Eq. 14 is not the same as Eq. 10. In Eq. 14, it is assumed that the main loss of pressure head between the bulk soil and the leaves is in the soil when stress occurs. Under this assumption, it can be stated that the pressure

head at the soil root interface is everywhere in the root zone equal to the wilting point. However, when pressure head losses in the root system become important, the pressure head at the soil root interface can be in some parts of the root system well above the wilting point. To check this, another Tpmax can be defined which is the maximal uptake when the leaf water potential is equal to the critical leaf water potential, hwl and the water potentials at the soil root interface, h0, are everywhere in the root system equal to the bulk soil water potential hs. When this Tpmax is smaller than the Tpmax that is obtained assuming that the water potentials at the soil root interface are everywhere equal to the wilting point, then pressure head losses in the root system are more important.

In Eq. 14, there are no plant conductivities. Therefore, Tpmax as defined in Eq. 14 must be different from Tpmax in Figure 5. In figure 5, the effect of Kroot and Ll on Tpmax is evaluated. But it is not clear to me what exactly the boundary conditions were to calculate Tpmax. I think the authors should explain how they defined Tpmax and try to use a consistent definition.

P10 section 2.2.5: I made the comment that besides the pressure head, also the potential transpiration rate plays a role in the definition of the stress function. Instead of making an extra section about it, I would suggest to include how the different model concepts deal with the dependence of the stress function on the potential transpiration. I would propose to include in Eq. 3 also Tp in the stress function: $\alpha(h(z),Tp)$. In the physical model, the maximal uptake rate is calculated and this maximal uptake rate depends on the soil water potential. That means that if a stress function would be defined for the physical model as Ta/Tp(h), it would also depend on Tp since for a lower Tp, the pressure head at which the maximal uptake is equal to Tp is lower. The authors already mention for the Jarvis model, that the dependence of the stress function on Tp should be different from the dependence that is derived for the Feddes model. It is interesting that in the comparison between the Jarvis model and the physical model, they determine a stress function (Eq. 18) that is independent of the potential transpiration rate (neither M nor Mmax depend on Tp).

P 18 Eq. 32: The authors included now the Aikake's information criterion. I think this is a very suitable parameter but in this context, I am wondering whether it would not be better to investigate the performance of a model that uses the same vegetation parameters for different soils and transpiration conditions could be used. The problem now is that for the same properties of the vegetation (root density and root hydraulic conductivities) different 'root water uptake parameters' must be used depending on the soil and potential ET conditions. Therefore, although the empirical models do not have that many parameters, they need to be adjusted for different soil conditions and climate (potential transpiration). My question is therefore whether the de Jong van Lier model requires that many more parameters than the empirical models. Root length density can be measured and the root conductivities could be fitted as well. In the end, this might result in less parameters that need to be determined when the model is to be used in different soils and for different transpiration rates. This problem may be even more relevant when considering that soils often have layers with different hydraulic properties. I do not suggest that the authors refit now the models to the different cases they considered using only one parameter set for the different cases (soils and potential transpiration) but I would propose including this in the conclusion and discussion section.

P20: The authors concluded that one drying experiment would be sufficient to parameterize the model and use it to run root water uptake during a growing season. I am just wondering whether one

drying experiment would be enough. How can the dependence of the stress parameters on the transpiration rate be defined then? Maybe, this dependence is not so important for simulations over an entire growing season as long as a drying period with a relevant transpiration rate for the entire growing season is chosen. But this could maybe be taken up in the discussion section. Furthermore, the authors used daily averaged transpiration rates and made the stress function dependent on the daily transpiration rates. But this means that in the model, the transpiration rates should not be resolved within one day since otherwise, different stress functions will have to be used that consider the peak transpiration rates during midday (which are about a factor 8 higher than the daily average transpiration rate).

P21: The authors concluded that for cases with low root water uptake compensation, which correspond with cases of low root length density, the Feddes models perform pretty well. Can this be explained by the fact that for low root density, the resistance to water water flow from the soil to the leaves is mainly dominated by the resistance to flow in the soil? When the soil dries out at one depth, the resistance to the flow at another depth and therefore flow will not change since this resistance is dominated by the soil conditions and is hardly influenced by changing conditions in the roots.

Detailed comments:

P1 ln 13 ‚all models that accounts' → that account

P1 ln 14-15: From reading only the abstract, the reader will not understand this sentence since it is not yet clear what the JMII model is.

P3 ln 29: Dimension of Ta should be $L\,T^{-1}$.

P17: Definition of coefficient of determination and model efficiency. In fact, the model efficiency is the same as the formal definition of the coefficient of determination. However, the squared correlation coefficient is sometimes called the coefficient of determination (which is very confusing of course when coefficient of determination mostly refers to a metric that is equal to model efficiency).

P17: omega_c > 1. Isn't the upper boundary of omega_c equal to 1?

---

## Author Response (AR2)

**Author's Response**

We are thankful for the critical reading, comments and suggestion made by N. Jarvis and Referee#2. Their suggestions were implemented in the manuscript. We also replied the questions and comments made by referee#2 in the following.

**Anonymous Referee #2**

Abstract: If you briefly mention what is behind the 'alternative empirical models' that are proposed, I think that the abstract could be improved. How do they differ from the Feddes and Jarvis models?

We added how, in general, all empirical models are defined.

P6 ln 2-3: Maybe before going to De Jong van Lier et al. (2013) where the reader can find information about the algorithm, I think it would be helpful to explain how an equation with three unknowns Ta, hl and h0 can be solved in general. First, I think you need to write that in equation 8, there are only two unknowns: either h0 and hl or h0 and Ta. In order to solve the equation for the two unknowns, additional equations are required. After formulating equations for h0's at different depths, Eq 1, making use of Eq. 6, can be used to solve for Ta (or hl) and the distribution of h0 with depth in the soil profile.

We rewrote this part of the manuscript according to your suggestions

P6 Figure 2: I am afraid that I still do not understand figure 2 and its caption. What I suppose that is shown in Figure 2 is the sink term for the case that the root length density and the soil water potential are uniform in the root zone (i.e. they do not change with depth in the root zone). As a consequence, also the sink term is uniform in the root zone and the transpiration rate is simply the sink term multiplied by the root zone thickness. So I do not understand that the plant transpiration was set to 1 mm d−1.

The caption of Fig. 2 was corrected. The root water uptake (RWU) is not calculated considering a homogeneous root length density (RLD) down to a depth of 50 cm. It corresponds to the RWU for any soil layer having a given RLD (three values are shown in Fig. 2) belonging to a certain RLD distribution. Thus, Fig. 2 shows the RWU in a given soil layer (with a certain RLD) when a plant adjusts its $h_l$ to achieve a transpiration rate of 1 mm d$^{-1}$ for different soil water conditions. The true value of $h_l$ will depend on the RLD distribution and soil moisture distribution over the root zone.

With the corrected caption, we think it is now what we intend to show in Fig. 2.

P7 and P8: Comparison of the Jarvis model and the De Jong van Lier model and Figure 5. In my previous comments on the paper, I posed the question whether the analogy between the two models relies on the assumption that when stress occurs the water potential the root surface is every the same in the root zone. I also asked whether the model of De Jong van Lier only predicts stress when the water potential at the root surface is everywhere equal to the wilting point. If the De Jong van Lier model can predict that stress may also occur even when in parts of the root zone the surface water potential at the root surface is still above the wilting point, then the De Jong van Lier and Jarvis models may also deviate under stress conditions. I think that the main problem in the description here is that the authors are not fully consistent in defining the stress conditions: Eq. 14 is not the same as Eq. 10. In Eq. 14, it is assumed that the main loss of pressure head between the bulk soil and the leaves is in the soil when stress occurs. Under this assumption, it can be stated that the pressure head at the soil root interface is everywhere in the root zone equal to the wilting point. However, when pressure head losses in the root system become important, the pressure head at the soil root interface can be in some parts of the root system well above the wilting point. To check this, another Tpmax can be defined which is the maximal uptake when the leaf water potential is equal to the critical leaf water potential, hwl and the water potentials at the soil root interface, h0, are everywhere in the root system equal to the bulk soil water potential hs. When this Tpmax is smaller than the Tpmax that is obtained assuming that the water potentials at the soil root interface are everywhere equal to the wilting point, then pressure head losses in the root system are more important.

In Eq. 14, there are no plant conductivities. Therefore, Tpmax as defined in Eq. 14 must be different from Tpmax in Figure 5. In figure 5, the effect of Kroot and Ll on Tpmax is evaluated. But it is not clear to me what exactly the boundary conditions were to calculate Tpmax. I think the authors should explain how they defined Tpmax and try to use a consistent definition.

**The analogy between the models relies on the assumption of uniform $h_0$ at the onset of drought stress**

5     The De Jong van Lier et al. (2008) model indeed assumes that $h_0 = h_w$ over the root zone at stress onset. The comparison of the Jarvis (1989) model with the De Jong van Lier et al. (2008) model (Jarvis, 2011) is also based on the assumption of $h_0 = h_w$ over the root zone. It is also shown that the models differ in unstressed conditions. This information can be found in the last version of the manuscript. However, possibly referee#2 did not clearly distinguish between the De Jong van Lier et al. (2008) and De Jong van Lier et al. (2013) models. As shown in section 2.2.1, we are comparing the Jarvis (1989) model to the

10    De Jong van Lier et al. (2008) model — a physical model that does not account for plant hydraulic resistances. We mention in section 2.1 that this model was extended to take this into account, leading to the De Jong van Lier et al. (2013) model. Thus, when stating that " If the De Jong van Lier model can predict that stress may also occur even when in parts of the root zone the surface water potential at the root surface is still above the wilting point, then the De Jong van Lier and Jarvis models may also deviate under stress conditions", he is referring to the De Jong van Lier et al. (2013) model, and not to the De Jong van

15    Lier et al. (2008) model, which we are discussing and making the comparison in the current section.

**Defining the stress condition: eq. 14 versus eq. 10**

The stress conditions are defined differently by the De Jong van Lier et al. (2008) and De Jong van Lier et al. (2013) models — eq. 14 and eq. 10, respectively. In the De Jong van Lier et al. (2008) model at the stress onset, $h_0 = h_w$ over the root zone and it is correlated to the Jarvis (1989) model (Jarvis, 2011). Whereas, in the De Jong van Lier et al. (2013) model, when

20    stress occurs, $h_l = h_{wl}$. Notice that $h_{wl} < h_w$, and means that when $h_l = h_{wl}$, $h_0$ over the root zone must approach $h_w$ due to pressure head loss, but it is not necessarily equal to $h_w$, as it is assumed in the De Jong van Lier et al. (2008) model.

We tried to make all this clearer by including in section 2.1 how $T_{p,max}$ is defined in the De Jong van Lier et al. (2013) model and stressing the difference with $T_{p,max}$ for the De Jong van Lier et al. (2008) model. We think it is now clearer how both models define stress, .i.e. plant transpiration reduction: the De Jong van Lier et al. (2013) defines transpiration reduction in

25    terms of plant leaf potential (eq. 10), which is an improvement of the De Jong van Lier et al. (2008) model that defines it in terms of the pressure at the root surface, considered constant with depth (eq. 14).

P10 section 2.2.5: I made the comment that besides the pressure head, also the potential transpiration rate plays a role in the definition of the stress function. Instead of making an extra section about it, I would suggest to include how the different model concepts deal with the dependence of the stress function on the potential transpiration. I would propose to include in Eq. 3 also

30    Tp in the stress function: $\alpha(z),Tp$. In the physical model, the maximal uptake rate is calculated and this maximal uptake rate depends on the soil water potential. That means that if a stress function would be defined for the physical model as Ta/Tp(h), it would also depend on Tp since for a lower Tp, the pressure head at which the maximal uptake is equal to Tp is lower. The authors already mention for the Jarvis model, that the dependence of the stress function on Tp should be different from the dependence that is derived for the Feddes model. It is interesting that in the comparison between the Jarvis model and the

35    physical model, they determine a stress function (Eq. 18) that is independent of the potential transpiration rate (neither M nor Mmax depend on Tp).

In the case of the dependence of the stress function on $T_p$, it should be noticed that $\alpha$ in eq. 3 already takes $T_p$ in to account by setting $h_3$ for two levels of $T_p$: low (1 mm d$^{-1}$) and high (5 mm d$^{-1}$) level. In the manuscript we proposed to make the optimizations for these two levels of $T_p$ aiming to provide $h_3$ for the respective $T_p$ levels. In that way, the dependence of $h_3$ on

40    $T_p$ can be taken into account.

As commented by referee#2, we did mention the dependence of $\alpha$ on $T_p$ is different in Jarvis and Feddes model. It can be seen that we made this statement on the grounds of the $h_3$ values in both models.

All the evaluated empirical models make use of the Feddes reduction function or a modified version, at which the dependence of $\alpha$ on $T_p$ is taken into account by the value of $h_3$.

P 18 Eq. 32: The authors included now the Aikake's information criterion. I think this is a very suitable parameter but in this context, I am wondering whether it would not be better to investigate the performance of a model that uses the same vegetation parameters for different soils and transpiration conditions could be used. The problem now is that for the same properties of the vegetation (root density and root hydraulic conductivities) different 'root water uptake parameters' must be used depending on the soil and potential ET conditions. Therefore, although the empirical models do not have that many parameters, they need to be adjusted for different soil conditions and climate (potential transpiration). My question is therefore whether the de Jong van Lier model requires that many more parameters than the empirical models. Root length density can be measured and the root conductivities could be fitted as well. In the end, this might result in less parameters that need to be determined when the model is to be used in different soils and for different transpiration rates. This problem may be even more relevant when considering that soils often have layers with different hydraulic properties. I do not suggest that the authors refit now the models to the different cases they considered using only one parameter set for the different cases (soils and potential transpiration) but I would propose including this in the conclusion and discussion section.

To address this, as comment was added to the conclusion.

P20: The authors concluded that one drying experiment would be sufficient to parameterize the model and use it to run root water uptake during a growing season. I am just wondering whether one drying experiment would be enough. How can the dependence of the stress parameters on the transpiration rate be defined then? Maybe, this dependence is not so important for simulations over an entire growing season as long as a drying period with a relevant transpiration rate for the entire growing season is chosen. But this could maybe be taken up in the discussion section. Furthermore, the authors used daily averaged transpiration rates and made the stress function dependent on the daily transpiration rates. But this means that in the model, the transpiration rates should not be resolved within one day since otherwise, different stress functions will have to be used that consider the peak transpiration rates during midday (which are about a factor 8 higher than the daily average transpiration rate).

This part must be rephrased. The dependence of $\alpha$ on $T_p$ is accounted for by the parameter values derived for the two levels of $T_p$ in the drying-out experiment. By setting a constant $T_p$ over the drying-out period, the optimized parameters are therefore defined for this $T_p$ level. Parameter values in between can be obtained by linear interpolation as we did in section 4.2, and explained in section 3.4.

P21: The authors concluded that for cases with low root water uptake compensation, which correspond with cases of low root length density, the Feddes models perform pretty well. Can this be explained by the fact that for low root density, the resistance to water water flow from the soil to the leaves is mainly dominated by the resistance to flow in the soil? When the soil dries out at one depth, the resistance to the flow at another depth and therefore flow will not change since this resistance is dominated by the soil conditions and is hardly influenced by changing conditions in the roots.

Higher root hydraulic resistances will increase root water uptake "compensation". As this mechanism is enhanced, the Feddes models will not perform well. However, in our simulations we kept plant hydraulic resistances constant and varied only the root length density. Thus, the performance of the Feddes models when varying $R$ is not related to the increase/decrease of plant hydraulic resistances. Increasing $K_{root}$, for instance, as $R$ increases will enhance the deviations.

[revised manuscript text omitted]